# Initialization Matters: Unraveling the Impact of Pre-Training on Federated Learning

**Divyansh Jhunjhunwala**                                                   *djhunjhu@andrew.cmu.edu*
*Carnegie Mellon University*

**Pranay Sharma**                                                          *pranaysh@iitb.ac.in*
*IIT Bombay*

**Zheng Xu**                                                               *xuzheng@google.com*
*Google*

**Gauri Joshi**                                                           *gaurij@andrew.cmu.edu*
*Carnegie Mellon University*

**Reviewed on OpenReview:** *https: // openreview. net/ forum? id= wW4Cvhkxcx*

## Abstract

Initializing with pre-trained models when learning on downstream tasks is becoming standard practice in machine learning. Several recent works explore the benefits of pre-trained initialization in a federated learning (FL) setting, where the downstream training is performed at the edge clients with heterogeneous data distribution. These works show that starting from a pre-trained model can substantially reduce the adverse impact of data heterogeneity on the test performance of a model trained in a federated setting, with no changes to the standard FedAvg training algorithm. In this work, we provide a deeper theoretical understanding of this phenomenon. To do so, we study the class of two-layer convolutional neural networks (CNNs) and provide bounds on the training error convergence and test error of such a network trained with FedAvg. We introduce the notion of *aligned* and *misaligned* filters at initialization and show that the data heterogeneity only affects learning on misaligned filters. Starting with a pre-trained model typically results in fewer misaligned filters at initialization, thus producing a lower test error even when the model is trained in a federated setting with data heterogeneity. Experiments in synthetic settings and practical FL training on CNNs verify our theoretical findings.

## 1 Introduction

Federated Learning (FL) (McMahan et al., 2017) has emerged as the de-facto paradigm for training a Machine Learning (ML) model over data distributed across multiple clients with privacy protection due to its no data-sharing philosophy. Ever since its inception, it has been observed that heterogeneity in client data can severely slow down FL training and lead to a model that has poorer generalization performance than a model trained on Independent and Identically Distributed (IID) data (Kairouz et al., 2021; Li et al., 2020; Yang et al., 2021a). This has led works to propose several *algorithmic* modifications to the popular Federated Averaging (`FedAvg`) algorithm such as variance-reduction (Acar et al., 2021; Karimireddy et al., 2020), contrastive learning (Li et al., 2021; Tan et al., 2022) and sophisticated model-aggregation techniques (Lin et al., 2020; Wang et al., 2020), to combat the challenge of data heterogeneity.

A recent line of work (Chen et al., 2022; Nguyen et al., 2022) has sought to understand the benefits of starting from *pre-trained* models instead of randomly initializing the global model when doing FL. This idea has been popularized by results in the centralized setting (Devlin et al., 2019; Radford et al., 2019; He et al.,

2019; Dosovitskiy et al., 2021), which show that starting from a pre-trained model can lead to state-of-the-art accuracy and faster convergence on downstream tasks. Pre-training is usually done on internet-scale public data (Schuhmann et al., 2022; Thomee et al., 2016; Raffel et al., 2020; Gao et al., 2020) in order for the model to learn fundamental data representations (Sun et al., 2017; Mahajan et al., 2018; Radford et al., 2019), that can be easily applied for downstream tasks. Thus, while it would not be unexpected to see some gains of using pre-trained models even in FL, what is surprising is the sheer scale of improvement. In many cases Nguyen et al. (2022); Chen et al. (2022) show that just starting from a pre-trained model can significantly reduce the gap between the performance of a model trained in a federated setting with non-IID versus IID data partitioning with *no algorithmic modifications*. Figure 1 shows our own replication of this phenomenon, where starting from a pre-trained model can lead to almost 14% improvement in accuracy for FL with non-IID data (i.e., high data heterogeneity) compared to 4% for FL with IID data and 2% in the centralized setting. This observation leads us to ask the question:

*Why can pre-trained initialization drastically improve model performance in FL?*

One reason suggested by Nguyen et al. (2022) is a lower value of the training loss at initialization when starting from pre-trained models. However, this observation can only explain improvement in training convergence speed (see Theorem V in Karimireddy et al. (2021)) and not the significantly improved generalization performance of the trained model. Also, a pre-trained initialization can have larger loss than random initialization while continuing to have faster convergence and better generalization (see Table 1 in Nguyen et al. (2022)). Chen et al. (2022); Nguyen et al. (2022), also observe some optimization-related factors when starting from a pre-trained model including smaller distance to optimum, better conditioned loss surface (smaller value of the largest eigen value of Hessian) and more stable global aggregation. However, it has not been formally proven that these factors can reduce the adverse effect of non-IID data. Thus, there is still a lack of fundamental understanding of why pre-trained initialization benefits generalization for non-IID FL.

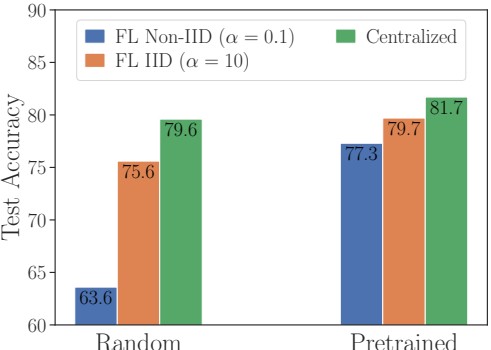

Figure 1: Test accuracy (%) on CIFAR10 with SqueezeNet model Iandola et al. (2016) under random and pretrained initializations for FL and centralized training. Pre-training benefits FL more than centralized setting and significantly reduces the gap between IID and non-IID FL model performance.

**Our contributions.** In this work we provide a deeper theoretical understanding of the importance of initialization for `FedAvg` by studying two-layer ReLU Convolutional Neural Networks (CNNs) for binary classification. This class of neural networks lends itself to tractable analysis while providing valuable insights that empirically extend to training deeper CNNs as shown by several recent works (Cao et al., 2022; Du et al., 2018; Kou et al., 2023; Zou et al., 2023; Jelassi & Li, 2022; Bao et al., 2024; Oh & Yun, 2024). Our data generation model, also studied in Cao et al. (2022); Kou et al. (2023), allows us to utilize a *signal-noise decomposition* result (see Proposition 1) to perform a fine-grained analysis of the CNN filter weight updates than can be done with general non-convex optimization. Some highlights of our results are as follows:

1. We introduce the notion of *aligned* and *misaligned* filters at initialization (Lemma 1) and show that data heterogeneity affects signal learning only on misaligned filters while noise memorization is unaffected by data heterogeneity (see Lemma 2). A pre-trained model is expected to have fewer misaligned filters, which can explain the reduced effect of non-IID data.

2. We provide a test error upper bound for `FedAvg` that depends on the number of misaligned filters at initialization and data heterogeneity. The effect of data heterogeneity on misaligned filters is exacerbated as clients perform more local steps, which explains why FL benefits more from pre-trained initialization than centralized training. To our knowledge, this is the first result where the test error for `FedAvg` explicitly depends on initialization conditions (Theorem 2).

3. We prove the training error convergence of `FedAvg` by adopting a two-stage analysis: a first stage where the local loss derivatives are lower bounded by a constant and second stage where the model is in the

neighborhood of a global minimizer with nearly convex loss landscape. Our analysis shows a provable benefit of using local steps in the first stage to reduce communication cost.

4. We experimentally verify our upper bound on the test error in a synthetic data setting (see Section 3 as well as conduct experiments on practical FL tasks which show that our insights extend to deeper CNNs (see Section 4).

**Related Work.** The two-layer CNN model that we study in this work was originally introduced in Zou et al. (2023) for the purpose of analyzing the generalization error of the `Adam` optimizer in the centralized setting. Later Cao et al. (2022) study the same model to analyze the phenomenon of *benign overfitting* in two-layer CNN, i.e., give precise conditions under which the CNN can perfectly fit the data while also achieving small population loss. Oh & Yun (2024) use this model to prove the benefit of patch-level data augmentation techniques such as Cutout and CutMix. Kou et al. (2023) relaxes the the polynomial ReLU activation in Cao et al. (2022) to the standard ReLU activation and also introduces label-flipping noise when analyzing benign overfitting in the centralized setting. We do not consider label-flipping in our work for simplicity; however this can be easily incorporated as future work. To the best of our knowledge, we are only aware of two other works (Huang et al., 2023; Bao et al., 2024) that analyze the two-layer CNN in a FL setting. The focus in Huang et al. (2023) is on showing the benefit of collaboration in FL by considering signal heterogeneity across the data in clients while Bao et al. (2024) considers signal heterogeneity to show the benefit of local steps. Both Huang et al. (2023) and Bao et al. (2024) do not consider any label heterogeneity and there is no emphasis on the importance of initialization, making their analysis quite different from ours. We defer more discussion on other related works to the Appendix.

## 2 Problem Setup

We begin by introducing the data generation model and the two-layer convolutional neural network, followed by our FL objective and a brief primer on the `FedAvg` algorithm. We note that given integers $a, b$, we denote by $[a : b]$ the set of integers $\{a, a + 1, \ldots, b\}$. Also, $[n]$ denotes $\{1, 2, \ldots, n\}$.

**Data-Generation Model.** Let $\mathcal{D}$ be the global data distribution. A datapoint $(\mathbf{x}, y) \sim \mathcal{D}$ contains feature vector $\mathbf{x} = [\mathbf{x}(1)^\top, \mathbf{x}(2)^\top]^\top \in \mathbb{R}^{2d}$ with two components $\mathbf{x}(1), \mathbf{x}(2) \in \mathbb{R}^d$ and label $y \in \{+1, -1\}$, that are generated as follows:

1. Label $y \in \{-1, 1\}$ is generated as $\mathbb{P}[y = 1] = \mathbb{P}[y = -1] = 1/2$.

2. One of $\mathbf{x}(1), \mathbf{x}(2)$ is chosen at random and assigned as $y\boldsymbol{\mu}$, where $\boldsymbol{\mu} \in \mathbb{R}^d$ is the signal vector that we are interested in learning. The other of $\mathbf{x}(1), \mathbf{x}(2)$ is set to be the noise vector $\boldsymbol{\xi} \in \mathbb{R}^d$, which is generated from the Gaussian distribution $\mathcal{N}(\mathbf{0}, \sigma_p^2 \cdot (\mathbf{I} - \boldsymbol{\mu}\boldsymbol{\mu}^\top \cdot \|\boldsymbol{\mu}\|_2^{-2}))$.

This data generation model is inspired by image classification tasks Cao et al. (2022) where it has been observed that only some of the image patches (for example, the foreground) contain information (i.e. the signal) about the label. We would like the model to predict the label by focusing on such informative image patches and ignoring background patches that act as noise and are irrelevant to the classification. Note that by definition, the noise vector $\boldsymbol{\xi}$ is orthogonal to the signal $\boldsymbol{\mu}$, i.e., $\boldsymbol{\xi}^\top \boldsymbol{\mu} = 0$. We assume orthogonality just for simplicity of analysis and can be easily relaxed as done in Kou et al. (2023) by assuming a stronger condition on the dimensionality $d$ (see Condition (C2) defined as part of Main Condition 1). The key idea is that we can show that $\frac{\langle \xi, \mu \rangle}{\|\xi\|^2} = \mathcal{O}(1/d) \approx 0$ which implies that $\xi$ and $\mu$ are nearly orthogonal with high probability when $d$ is sufficiently large. Consequently, terms involving $\langle \xi, \mu \rangle$ can be treated as negligible noise, absorbed into existing noise terms, and do not significantly affect the analysis.

**Measure of Data Heterogeneity.** We consider $n$ datapoints drawn from the distribution $\mathcal{D}$, and partitioned across $K$ clients such that each client has $N = n/K$ datapoints. The assumption of equal-sized client datasets is made for simplicity of analysis and can be easily relaxed. The data partitioning determines the level of heterogeneity across clients.

Let $D_{+,k}$ and $D_{-,k}$ denote the set of samples at client $k$ with positive $(y = +1)$ and negative $(y = -1)$ labels respectively. Define

$$h := \frac{\sum_{k=1}^{K} \min\left(\left|D_{+,k}\right|, \left|D_{-,k}\right|\right)}{n} \in [0, 1/2]. \tag{1}$$

Note that a smaller $h$ implies a higher data heterogeneity on average. In the IID setting, with uniform partitioning across clients, we expect $\min(\left|D_{+,k}\right|, \left|D_{-,k}\right|) \approx n/2K$ for all $k \in [K]$, and therefore $h \approx 1/2$. In the extreme non-IID setting where each client only has samples from one class, $h = 0$.

**Two-Layer CNN.** We now describe our two-layer CNN model. The first layer in our model consists of $2m$ filters $\{\mathbf{w}_{j,r}\}_{r=1}^{m}, j \in \{\pm 1\}$, where each $\mathbf{w}_{j,r} \in \mathbb{R}^d$ performs a 1-D convolution on the feature $\mathbf{x}$ with stride $d$ followed by ReLU activation and average pooling Lin et al. (2013); Yu et al. (2014). The weights in the second layer then aggregate the outputs produced after pooling to get the final output and are fixed as $2/m$ for $j = +1$ filters and $-2/m$ for $j = -1$ filters. Formally, we have,

$$f(\mathbf{W}, \mathbf{x}) = \underbrace{\frac{1}{m} \sum_{r=1}^{m} \left[\sigma\left(\langle \mathbf{w}_{+1,r}, y\boldsymbol{\mu}\rangle\right) + \sigma\left(\langle \mathbf{w}_{+1,r}, \boldsymbol{\xi}\rangle\right)\right]}_{:=F_{+1}(\mathbf{W}_{+1}, \mathbf{x})} - \underbrace{\frac{1}{m} \sum_{r=1}^{m} \left[\sigma\left(\langle \mathbf{w}_{-1,r}, y\boldsymbol{\mu}\rangle\right) + \sigma\left(\langle \mathbf{w}_{-1,r}, \boldsymbol{\xi}\rangle\right)\right]}_{:=F_{-1}(\mathbf{W}_{-1}, \mathbf{x})}. \tag{2}$$

Here $\mathbf{W} \in \mathbb{R}^{2md}$ parameterizes all the weights of our neural network, $\mathbf{W}_{+1}, \mathbf{W}_{-1} \in \mathbb{R}^{md}$ parameterize the weights of the $j = +1$ filters and $j = -1$ filters respectively, and $\sigma(z) = \max(0, z)$ is the ReLU activation. Intuitively $F_j(\mathbf{W}_j, \mathbf{x})$ represents the 'logit score' that the model assigns to label $j$.

**FL Training and Test Objectives.** Let $\{(\mathbf{x}_{k,i}, y_{k,i})\}_{i=1}^{N}$ be the local dataset at client $k$. Then the global FL objective can be written as follows:

$$\min_{\mathbf{W} \in \mathbb{R}^{2d}} \left\{L(\mathbf{W}) = \frac{1}{K} \sum_{k=1}^{K} L_k(\mathbf{W})\right\},$$
$$L_k(\mathbf{W}) = \frac{1}{N} \sum_{i=1}^{N} \ell(y_{k,i}, f(\mathbf{W}, \mathbf{x}_{k,i})), \tag{3}$$

where $L_k(\mathbf{W})$ is the local objective at client $k$ and $\ell(z, \hat{z}) = \log(1 + \exp(-z \cdot \hat{z}))$ is the cross-entropy loss. We also define the test-error $L_{\mathcal{D}}^{0-1}(\mathbf{W})$ as the probability that $\mathbf{W}$ will misclassify a point $(\mathbf{x}, y) \sim \mathcal{D}$:

$$L_{\mathcal{D}}^{0-1}(\mathbf{W}) := \mathbb{P}_{(\mathbf{x},y)\sim\mathcal{D}}\left(y \neq \text{sign}(f(\mathbf{W}, \mathbf{x}))\right). \tag{4}$$

**The FedAvg Algorithm.** The standard approach to minimizing objectives of the form in Equation (3) is the `FedAvg` algorithm. In each round $t$ of the algorithm, the central server sends the current global model $\mathbf{W}^{(t)}$ to the clients. Clients initialize their local models to the current global model by setting $\mathbf{W}_k^{(t,0)} = \mathbf{W}^{(t)}$, for all $k \in [K]$, and run $\tau$ local steps of gradient descent (GD) as follows

$$\text{Local GD:} \quad \mathbf{W}_k^{(t,s+1)} = \mathbf{W}_k^{(t,s)} - \eta\nabla L_k(\mathbf{W}_k^{(t,s)}) \tag{5}$$

for all $s \in [0 : \tau - 1]$ and for all $k \in [K]$. After $\tau$ steps of Local GD, the clients send their local models $\{\mathbf{W}_k^{(t,\tau)}\}$ to the server, which aggregates them to get the global model for the next round: $\mathbf{W}^{(t+1)} = \sum_{k=1}^{K} \mathbf{W}_k^{(t,\tau)}/K$. While we focus on FedAvg with local GD in this work, we note that several modifications such as stochastic gradients instead of full-batch GD, partial client participation Yang et al. (2021b) and server momentum Reddi et al. (2021) are considered in both theory and practice. Studying these modifications is an interesting future research direction.

## 3  Main Results

In this section we first introduce our definition of filter alignment at initialization and a fundamental result regarding the signal-noise decomposition of the CNN filter weights. We then state our main result regarding the convergence of `FedAvg` with random initialization for the problem setup described in Section 2 and the impact of data heterogeneity and filter alignment at initialization on the test-error. Later we discuss why starting from a pre-trained model can improve the test accuracy of `FedAvg`.

### 3.1 Filter Alignment at Initialization

Given datapoint $(\mathbf{x}, y)$, for the CNN to correctly predict the label $y$ and minimize the loss $\ell(y, f(\mathbf{W}, \mathbf{x}))$, from Equation (2)-Equation (3), we want $yf(\mathbf{W}, \mathbf{x}) = F_y(\mathbf{W}_y, \mathbf{x}) - F_{-y}(\mathbf{W}_{-y}, \mathbf{x})) \gg 0$. At an individual filter $r \in [m]$, this can happen either with $\langle \mathbf{w}_{y,r}, y\boldsymbol{\mu} \rangle \gg 0$ or $\langle \mathbf{w}_{y,r}, \boldsymbol{\xi} \rangle \gg 0$. However, we want the model to focus on the signal $y\boldsymbol{\mu}$ in $\mathbf{x}$ while making the prediction. Therefore, for filter $(j, r)$ we want $\langle \mathbf{w}_{j,r}, y\boldsymbol{\mu} \rangle \gg 0$ if $j = y$ and $\langle \mathbf{w}_{j,r}, y\boldsymbol{\mu} \rangle \ll 0$ if $j = -y$. Depending on the initialization of our CNN, we have the following definition of *aligned* and *misaligned* filters.

**Definition 1.** *The $(j, r)$-th filter (with $j \in \{\pm 1\}, r \in [m]$) is said to be aligned (with signal) at initialization if $\langle \mathbf{w}_{j,r}^{(0)}, j\boldsymbol{\mu} \rangle \geq 0$ and misaligned otherwise.*

We shall see in Section 3.4 that the alignment of a filter at initialization plays a crucial role in how well it learns the signal and also the overall generalization performance of the CNN in Theorem 2.

**Extension to multi-class settings and other architectures.** At a high level, the notion of alignment captures whether a filter consistently responds to the underlying signal in the data. Specifically, once a filter is aligned, the sign of its inner product with the signal remains unchanged over time, i.e., if $\langle \mathbf{w}^{(t)}, \boldsymbol{\mu} \rangle > 0$ then for all $t' \geq T$, $\text{sign}(\langle \mathbf{w}^{(t')}, \boldsymbol{\mu} \rangle) = \text{sign}(\langle \mathbf{w}, \boldsymbol{\mu} \rangle)$ (see Lemma 27). Importantly, this definition is not restricted to the binary setup. The key property - the preservation of sign consistency with respect to the signal - provides a principled way to quantify misalignment (see Equation (8)). While our current analysis formalizes this in the binary case, we believe the same notion can be naturally extended to multi-class classification. In particular, one could define alignment with respect to multiple class-specific signals, and then study whether filters consistently align with the relevant signal subspaces. We also discuss how alignment can be extended to deeper CNNs and Transformer architectures in Section 5.

### 3.2 Signal Noise Decomposition of CNN Filter Weights

One of the key insights in Cao et al. (2022) is that when training the two-layer CNN with GD, the filter weights at each iteration can be expressed as a linear combination of the initial filter weights, signal vector and noise vectors. Our first result below shows that this is true for `FedAvg` as well.

**Proposition 1.** *Let $\{\mathbf{w}_{j,r}^{(t)}\}$, for $j \in \{\pm 1\}$ and $r \in [m]$, be the global CNN filter weights in round $t$. Then there exist unique coefficients $\Gamma_{j,r}^{(t)} \geq 0$ and $\{P_{j,r,k,i}^{(t)}\}_{k,i}$ such that*

$$\mathbf{w}_{j,r}^{(t)} = \mathbf{w}_{j,r}^{(0)} + \underbrace{j\Gamma_{j,r}^{(t)} \cdot \|\boldsymbol{\mu}\|_2^{-2} \cdot \boldsymbol{\mu}}_{Signal\ Term} + \underbrace{\sum_{k=1}^K \sum_{i=1}^N P_{j,r,k,i}^{(t)} \cdot \|\boldsymbol{\xi}_{k,i}\|_2^{-2} \cdot \boldsymbol{\xi}_{k,i}}_{Noise\ Term}, \tag{6}$$

*where $k \in [K]$ and $i \in [N]$ denote the client and sample index respectively.*

This decomposition allows us to decouple the effect of the signal and noise components on the CNN filter weights, and analyze them separately throughout training.

As we run more communication rounds (denoted by $t$), we expect the weights to learn the signal $y\boldsymbol{\mu}$, hence it is desirable for $\Gamma_{j,r}^{(t)}$ to increase with $t$. In addition, the filter weights also inevitably memorize noise $\boldsymbol{\xi}$ and overfit to it, therefore the noise coefficients $\{P_{j,r,k,i}^{(t)}\}$ will also grow with $t$. We are primarily interested in the growth of positive noise coefficients $\overline{P}_{j,r,k,i}^{(t)} = P_{j,r,k,i}^{(t)} \mathbb{1}(P_{j,r,k,i}^{(t)} \geq 0)$ since the negative noise-coefficients $\underline{P}_{j,r,k,i}^{(t)} := P_{j,r,k,i}^{(t)} \mathbb{1}(P_{j,r,k,i}^{(t)} \leq 0)$ remain bounded (see Theorem 3 in Appendix C) and we can show that $\sum_{k,i} P_{j,r,k,i}^{(t)} = \Theta(\sum_{k,i} \overline{P}_{j,r,k,i}^{(t)})$. Henceforth, we refer to $\Gamma_{j,r}^{(t)}$ and $\sum_{k,i} \overline{P}_{j,r,k,i}^{(t)}$, as the *signal learning* and *noise memorization* coefficients of filter $(j, r)$ respectively. As we see later in Theorem 2, the ratio of signal learning to noise memorization $\Gamma_{j,r}^{(t)} / \sum_{k,i} \overline{P}_{j,r,k,i}^{(t)}$ is fundamental to the generalization performance of the CNN.

### 3.3 Training Loss Convergence and Test Error Guarantee

Next, we state our main result regarding the convergence of `FedAvg` with random initialization. We assume the CNN weights are initialized as $\mathbf{w}_{j,r}^{(0)} \sim \mathcal{N}(\mathbf{0}, \sigma_0^2 \mathbf{I}_d)$ for all filters, where $\mathbf{I}_d$ is the $(d \times d)$ identity matrix. We first state the following standard conditions used in our analysis.

**Condition 1** (Main Condition). *Let $\epsilon$ be a desired training error threshold and $\delta \in (0,1)$ be some failure probability.*[1]

*(C1) The allowed number of communication rounds $t$ is bounded by $T^* = \frac{1}{\eta}\mathrm{poly}(\epsilon^{-1}, m, n, d)$.*

*(C2) Dimension $d$ is sufficiently large: $d \gtrsim \max\left\{\frac{n\|\boldsymbol{\mu}\|_2^2}{\sigma_p^2}, n^2\right\}$.*

*(C3) Training set size $n$ and neural network width $m$ satisfy: $m \gtrsim \log(n/\delta), n \gtrsim \log(m/\delta)$.*

*(C4) Standard deviation of Gaussian initialization is sufficiently small: $\sigma_0 \lesssim \min\left\{\frac{\sqrt{n}}{\sigma_p d\tau}, \frac{1}{\|\boldsymbol{\mu}\|_2}\right\}$.*

*(C5) The norm of the signal satisfies: $\|\boldsymbol{\mu}\|_2^2 \gtrsim \sigma_p^2$.*

*(C6) Learning rate is sufficiently small: $\eta \lesssim \min\left\{\frac{nm}{\sigma_p^2 d}, \frac{1}{\|\boldsymbol{\mu}\|_2^2}, \frac{1}{\sigma_p^2 d}\right\}$.*

The above conditions are standard and have also been made in Cao et al. (2022); Kou et al. (2023) for the purpose of theoretical analysis. (C1) is a mild condition needed to ensure that the signal and noise coefficients remain bounded throughout the duration of training. Furthermore, we see in Theorem 1 that we only need $T = \mathcal{O}\left(mn\eta^{-1}\epsilon^{-1}d^{-1}\log(\tau/\epsilon)\right)$ rounds to reach a training error of $\epsilon$, which is well within the admissible number of rounds. (C2) is used to bound the correlation between the noise vectors and also the correlation of the initial filter weights with the signal and noise. (C3) is needed to ensure that a sufficient number of filters have non-zero activations at initialization so that the initial gradient is non-zero. (C4) is needed to ensure that the initial weights of the CNN are not too large and that it has bounded loss for all datapoints. (C5) is needed to ensure that signal learning is not too slow compared to noise memorization. Finally, a small enough learning rate in (C6) ensures that Local GD does not diverge. Additional discussion on these assumptions is provided in Appendix C. With this assumption we are now state our main results.

**Theorem 1** (Training Loss Convergence). *For any $\epsilon > 0$ under Condition 1, there exists a $T = \mathcal{O}\left(\frac{mn}{\eta\sigma_p^2 d\tau}\right) +$*

*$\mathcal{O}\left(\frac{mn\log(\tau/\epsilon)}{\eta\sigma_p^2 d\epsilon}\right)$ such that `FedAvg` satisfies $L(\mathbf{W}^{(T)}) \leq \epsilon$ with probability $\geq 1 - \delta$.*

**Proof Sketch.** The proof is divided into 3 parts. In the first part (Appendix C.2), we show that the magnitude of the signal and noise memorization coefficients for the global model is bounded for the entire duration of training (see Theorem 3), where $|\Gamma_{j,r}^{(t)}| \leq 4\log(T^*\tau)$ and $|P_{j,r,k,i}^{(t)}| \leq 4\log(T^*\tau)$ for all $0 \leq t \leq T^*-1$. Next, we divide our training into two stages. In the first stage (Appendix C.3), we show (see Lemma 21) that the noise (and also signal) memorization coefficients grow fast and are lower bounded by some constant after $T_1$ rounds i.e., $|\overline{P}_{j,r,k,i}^{(T_1)}| = \Omega(1)$. In the second stage (Appendix C.4), the growth of the noise and signal coefficients becomes relatively slower and the model reaches a neighborhood of a global minimizer where the loss landscape is nearly convex (see Lemma 25). Using this we can show that our objective is monotonically decreasing in every round (see Lemma 26), which establishes convergence (in Appendix C.5).

Note that our analysis does not require the condition $\eta \propto 1/\tau$ as is common in many works analyzing `FedAvg`. Therefore, by setting $\tau$ large enough we can make the number of rounds in the first stage as small as $\mathcal{O}(1)$, thereby reducing the communication cost of FL. However, in the second stage we do not see any continued benefit of local steps; in fact the number of rounds required grows as $\log(\tau)$. This suggests an optimal strategy would be to adapt $\tau$ throughout training: start with large $\tau$ and decrease $\tau$ after some rounds, which has also been found to work well empirically Wang & Joshi (2019).

**Theorem 2** (Test Error Bound). *Define signal-to-noise ratio $\mathrm{SNR} := \frac{\|\boldsymbol{\mu}\|_2}{\sigma_p\sqrt{d}}$ and $A_j := \{r \in [m] : \langle \mathbf{w}_{j,r}^{(0)}, j\boldsymbol{\mu}\rangle \geq 0\}$ to be the set of aligned filters (Definition 1) corresponding to label $j$. Then under the same conditions as Theorem 1, our trained CNN achieves*

---

[1] We use $\lesssim$ and $\gtrsim$ to denote inequalities that hide constants and logarithmic factors. See Appendix for exact conditions.

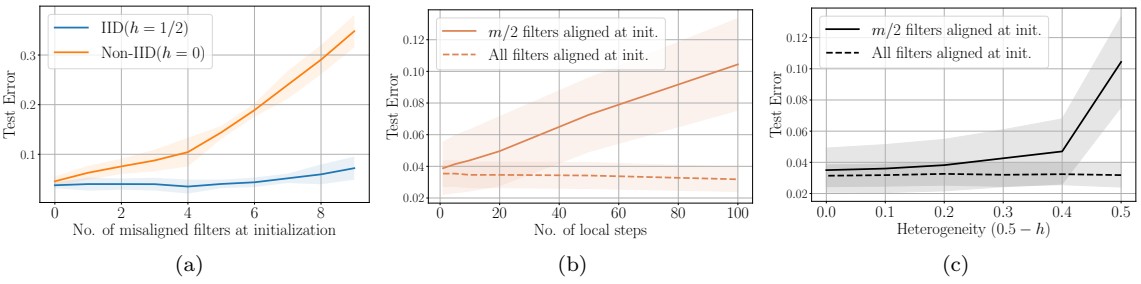

Figure 2: Empirical results on synthetic dataset to verify the upper bound on test error in Theorem 2. We fix the training error $\epsilon = 0.1$. Figure 2a: Test error increases as we increase the number of misaligned filters, with much larger rate of increase in the non-IID setting. Figures 2b and 2c: Test error increases with local steps and heterogeneity when $m/2$ filters are misaligned at initialization, remains constant when all the filters are aligned.

1. When $\text{SNR}^2 \lesssim 1/\sqrt{nd}$, test error $L_{\mathcal{D}}^{0-1}(\mathbf{W}^{(T)}) \geq 0.1$.

2. When $\text{SNR}^2 \gtrsim 1/\sqrt{nd}$, test error

$$L_{\mathcal{D}}^{0-1}(\mathbf{W}^{(T)}) \leq \frac{1}{2} \sum_{j \in \{\pm 1\}} \exp\Big( -\frac{n}{d}\Big[\frac{|A_j|}{m}\text{SNR}^2 + \big(1 - \frac{|A_j|}{m}\big)\text{SNR}^2\Big(h + \frac{1}{\tau}(1-h)\Big)\Big]^2\Big).$$

**Proof Sketch.** The upper bound on the test error in Theorem 2 relies on upper bounding the probability that $\Pr(y \cdot f(\mathbf{W}^{(t)}, \mathbf{x}) \leq 0)$ for a randomly sampled test data point $(\mathbf{x}, y)$. Following the Gaussian concentration of Lipschitz function (see Lemma 35), we can show that this comes down to lower bounding the ratio $\sum_{r=1}^{m} \sigma(\langle \mathbf{w}_{y,r}^{(t)}, y\boldsymbol{\mu}\rangle)/\sum_{r,k,i}\overline{P}_{-y,r,k,i}^{(t)}$ which can be intuitively interpreted as bounding the sum of signal learning across the $j = y$ filters to the sum of noise memorization across the $j = -y$ filters. As outlined later in Section 3.4, we see that noise memorization remains unaffected by initialization and data heterogeneity, i.e., we get $\sum_{r,k,i}\overline{P}_{-y,r,k,i}^{(t)} = \mathcal{O}(\sigma_p^2 md)$ (see Lemma 30), whereas signal learning depends on both the initial alignment and data heterogeneity , i.e., $\sum_{r=1}^{m}\sigma(\langle \mathbf{w}_{y,r}^{(t)}, y\boldsymbol{\mu}\rangle) = \Omega(\eta\|\boldsymbol{\mu}\|_2^2(|A_y| + (m - |A_y|)(h + \frac{1}{\tau}(1-h))))$ (see Lemma 33). Substituting these quantities in our upper bound in Appendix D.1 completes the proof.

For our lower bound, we use a similar argument as done in Kou et al. (2023). The first step involves showing that the $\Pr(y \cdot f(\mathbf{W}^{(t)}, \mathbf{x}) \leq 0) \geq 0.5\Pr(\Omega)$ where $\Omega = \Big\{\boldsymbol{\xi} : \Big|\sum_{j,r} j\sigma(\langle w_{j,r}^{(t)}, \boldsymbol{\xi}\rangle)\Big| \geq C\max_j\Big\{\sum_r \Gamma_{j,r}^{(T)}\Big\}\Big\}$ where $C$ is some positive constant. Now given $\text{SNR}^2 \lesssim 1/\sqrt{nd}$, we can show for any given $\boldsymbol{\xi}$ there exists a vector $\mathbf{v}$ such that one of $\boldsymbol{\xi}, \boldsymbol{\xi} + \mathbf{v}, -\boldsymbol{\xi}, \boldsymbol{\xi} + \mathbf{v}$ must belong to $\Omega$ (see Lemma 5.8 in Kou et al. (2023)). By union bound it follows that $\min\{\Pr(\Omega), \Pr(-\Omega), \Pr(\Omega - \mathbf{v}), \Pr(-\Omega - \mathbf{v})\} \geq 0.25$. Now using arguments based on symmetry and TV distance we can show that $\Pr(\Omega) \geq 0.22$, which completes the bound.

**Impact of SNR on harmful/benign overfitting.** Intuitively, if the SNR is too low ($\text{SNR}^2 \lesssim 1/\sqrt{nd}$), then there is simply not enough signal strength for the model to learn compared to the noise. Hence, we cannot expect the model to generalize well no matter how we train it. This generalizes the centralized training result in (Kou et al., 2023, Theorem 4.2) (with $p = 0$), which corresponds to $\tau = 1$ in `FedAvg`. In this case, the model is in the regime of *harmful overfitting*. However, if the SNR is sufficiently large ($\text{SNR}^2 \gtrsim 1/\sqrt{nd}$), we enter the regime of *benign overfitting*, where the model can fit the data and generalize well with the test error reducing exponentially with the global dataset size $n$.

**Empirical Verification.** We now provide empirical verification of the upper bound on the test error in Theorem 2 in the benign overfitting regime. We simulate a synthetic dataset following our data-generation model in Section 2, with $n = 20$ datapoints, $K = 2$ clients and $m = 10$ filters. Additional experimental details can be found in Appendix F. We fix a training error threshold of $\epsilon = 0.1$ and then measure the test error of our CNN under various settings in Figure 2. Figure 2a shows the test error as a function of the number of misaligned filters ($m - |A_j|$ in Theorem 2) under different data partitionings with the number of local steps

fixed at $\tau = 100$. While the test error grows with the number of misaligned filters in both data settings, the rate of growth is much larger in the non-IID setting. Figure 2b shows the test error as a function of local steps $\tau$ under different initializations for fixed $h = 0$ while Figure 2c shows the test error as a function of heterogeneity under different initializations for fixed $\tau = 100$. As predicted by our theory, heterogeneity and the number of local steps do not affect test error when all the filters are aligned at initialization. On the other hand, the test error grows with $\tau$ and heterogeneity when the number of misaligned filters is non-zero ($m/2 = 5$) for each $j \in \{\pm 1\}$. Therefore, our empirical results strongly validate our theoretical results showing the effect of heterogeneity, number of local steps and number of misaligned filters on the test error.

### 3.4 Impact of Filter Alignment and Data Heterogeneity on Signal Learning and Noise Memorization.

The key results in our analysis are the following lemmas which bound the growth of the signal learning and noise coefficient during the first stage of training, that is $0 \le t \le T_1$ (see discussion under Theorem 1). Using our definition of $A_j := \{r \in [m] : \langle \mathbf{w}_{j,r}^{(0)}, j\boldsymbol{\mu} \rangle \ge 0\}$ as the set of aligned filters, we have the following lemma for growth of the signal learning coefficient in the first stage.

**Lemma 1.** *Under Condition 1, for all $0 \le t \le T_1$, we have* $\Gamma_{j,r}^{(t)} = \Omega \left( \dfrac{t\eta \|\boldsymbol{\mu}\|_2^2 \tau}{m} \right)$ *if $r \in A_j$ and* $\Gamma_{j,r}^{(t)} = \Omega \left( \dfrac{t\eta \|\boldsymbol{\mu}\|_2^2 (1 + h(\tau - 1))}{m} \right)$ *if $r \notin A_j$.*

This lemma shows that *for aligned filters ($r \in A_j$), $\Gamma_{j,r}^{(t)}$ does not depend on heterogeneity and grows linearly with the number of local steps $\tau$.* On the other hand, *for misaligned filters ($r \notin A_j$), the growth depends on the heterogeneity parameter $h$.* Furthermore, under extreme data heterogeneity ($h = 0$), for misaligned filters $\Gamma_{j,r}^{(t)}$ *does not scale with the number of local steps $\tau$.* For the growth of noise coefficients we have the following corresponding lemma,

**Lemma 2.** *Under Condition 1, for all $0 \le t \le T_1$ we have* $\sum_{k,i} \overline{P}_{j,r,k,i}^{(t)} = \Theta \left( \dfrac{t\eta\tau\sigma_p^2 d}{m} \right)$.

This lemma shows that *noise memorization does not depend on data-heterogeneity or filter alignment and always scales linearly with the number of local steps $\tau$.* Intuitively, this can be expected because the noise vectors are independent of the label information $y$ in a datapoint following our data generation model in Section 2 and for any given filter we can show there are $\Omega(N)$ noise vectors that are aligned with the filter at initialization for every client with high probability (see Lemma 7).

Using the above two lemmas, we have the following bound on the ratio of signal learning to noise memorization for filter $(j, r)$ at the end of the first stage of training

$$\frac{\Gamma_{j,r}^{(T_1)}}{\sum_{k,i} \overline{P}_{j,r,k,i}^{(T_1)}} \ge \begin{cases} \mathrm{SNR}^2, & \text{if } r \in A_j, \\ \mathrm{SNR}^2(h + \frac{1}{\tau}(1 - h)), & \text{if } r \in [m] \setminus A_j. \end{cases} \tag{7}$$

This ratio is key to bounding the generalization performance of the CNN model as we show later in the proof of Theorem 2 in Appendix D.1. For aligned filters ($r \in A_j$), the ratio is unaffected by data heterogeneity $h$ and the number of local steps $\tau$. However, for misaligned filters ($r \in [m] \setminus A_j$), the ratio becomes smaller as heterogeneity increases ($h$ becomes smaller) or $\tau$ increases. Thus, for misaligned filters we see a corresponding dependence on heterogeneity and local steps in our upper bound on test error in Theorem 2. Note that in centralized training with $\tau = 1$, we have $(h + \frac{1}{\tau}(1 - h)) = 1$ and thus we do not see any impact of heterogeneity at misaligned filters. Therefore, we recover the bound $L_{\mathcal{D}}^{0-1}(\mathbf{W}^{(T)}) \le \exp(-n\mathrm{SNR}^2/d)$ in (Kou et al., 2023, Theorem 4.2). *It is only in FL training with $\tau > 1$ local steps that we encounter the adverse effect of data heterogeneity at the misaligned filters.*

**Empirical Verification.** We empirically verify the results above in the IID ($h = 1/2$) and non-IID ($h = 0$) setting following the same simulation setup as done in Figure 2. Figure 4 shows the alignment of filters at initialization (note that only filter $r = 3$ is aligned). Figure 3a shows that in the IID setting signal learning coefficients are similar for all the filters regardless of alignment and increases with the number of local step.

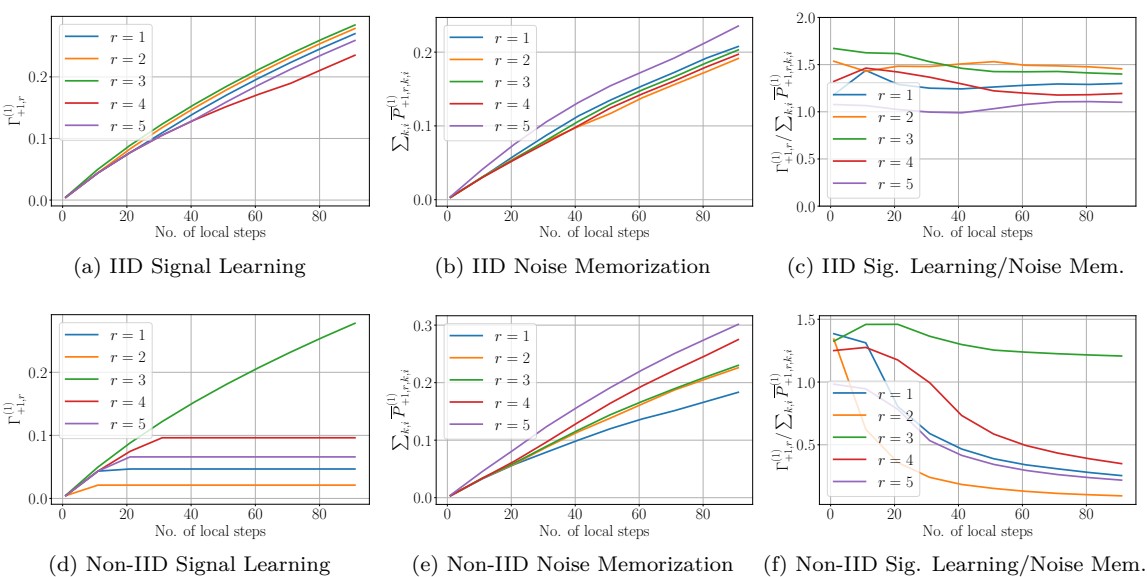

(a) IID Signal Learning     (b) IID Noise Memorization     (c) IID Sig. Learning/Noise Mem.

(d) Non-IID Signal Learning     (e) Non-IID Noise Memorization     (f) Non-IID Sig. Learning/Noise Mem.

Figure 3: Signal learning and noise memorization for our CNN model in the IID ($h = 1/2$) and non-IID ($h = 0$) setting after 1 round. Figures 3a, 3d: In the IID setting signal learning coefficients are similar for all the filters and increase with the number of local steps $\tau$ but in the non-IID setting they saturate (Lemma 1) for misaligned filters ($r = 1, 2, 4, 5$). Figures 3b, 3e: Noise memorization is similar for all filters in both settings and grows with $\tau$ Lemma 2. Figures 3c, 3f: in the IID setting, the ratio of signal learning to noise memorization remains independent of $\tau$. But in the non-IID setting, the ratio decreases to zero as $\tau$ increases for misaligned filters ($r = 1, 2, 4, 5$).

However, as shown by Figure 3d, in the non-IID setting signal learning saturates for misaligned filters. Figures 3b and 3e show that the growth of noise coefficients for all the filters is similar in the IID and non-IID case.

In Figure 3c we see that ratio of signal learning to noise memorization is lower bounded by a constant for all filters in the IID setting whereas in the non-IID setting it decays as $\tau$ increases for misaligned filters (Figure 3f), thus verifying our theoretical analysis.

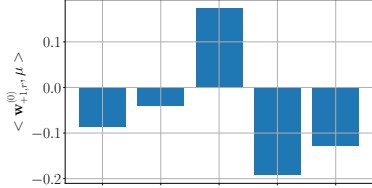

Figure 4: Initial alignment of the filters in Figure 3. Only filter $r = 3$ is aligned.

## 3.5 Impact of Pre-Training on Federated Learning

Given the result in Theorem 2, we return to our question in Section 1, about the *effect of pre-trained initialization on improving generalization performance in FL*. We focus on centralized pre-training but our discussion here can be extended to federated pre-training as well (see Lemma 31 which states a federated counterpart of the lemma below).

Suppose we pre-train a CNN model in a centralized manner on a dataset with signal $\boldsymbol{\mu}^{(\text{pre})}$ generated according to the data model described in Section 2. Now if we train for sufficient number of iterations, then we can show that *all* filters will be correctly aligned with the pre-training signal.

**Lemma 3** (All Filters Aligned After Sufficient Training). *There exists $T_1 = \mathcal{O}\left(\dfrac{mn}{\eta\sigma_p^2 d}\right)$ such that for all $t \geq T_1, j \in \{\pm 1\}, r \in [m]$ we have $\langle \mathbf{w}_{j,r}^{(pre,t)}, j\boldsymbol{\mu}^{(pre)}\rangle \geq 0$.*

Now suppose we pre-train for $t \geq T_1$ iterations to get a model $\mathbf{W}^{(\text{pre},*)}$ and use this model to initialize for downstream federated training (i.e., $\mathbf{W}^{(0)} = \mathbf{W}^{(\text{pre},\,*)}$) with signal vector $\boldsymbol{\mu}$. Then for all $j, r$ filters, we have $\langle \mathbf{w}_{j,r}^{(0)}, j\boldsymbol{\mu}\rangle = \langle \mathbf{w}_{j,r}^{(\text{pre},*)}, j\boldsymbol{\mu}^{(\text{pre})}\rangle + \langle \mathbf{w}_{j,r}^{(\text{pre},*)}, j(\boldsymbol{\mu} - \boldsymbol{\mu}^{(\text{pre})})\rangle$. We also know that $\langle \mathbf{w}_{j,r}^{(\text{pre},*)}, j\boldsymbol{\mu}^{(\text{pre})}\rangle \geq 0$ using Lemma 3. Therefore, if $\|\boldsymbol{\mu} - \boldsymbol{\mu}^{(\text{pre})}\|_2$ is small, all the filters $\{\mathbf{w}_{j,r}^{(0)}\}$ are correctly aligned with the signal $j\boldsymbol{\mu}$.

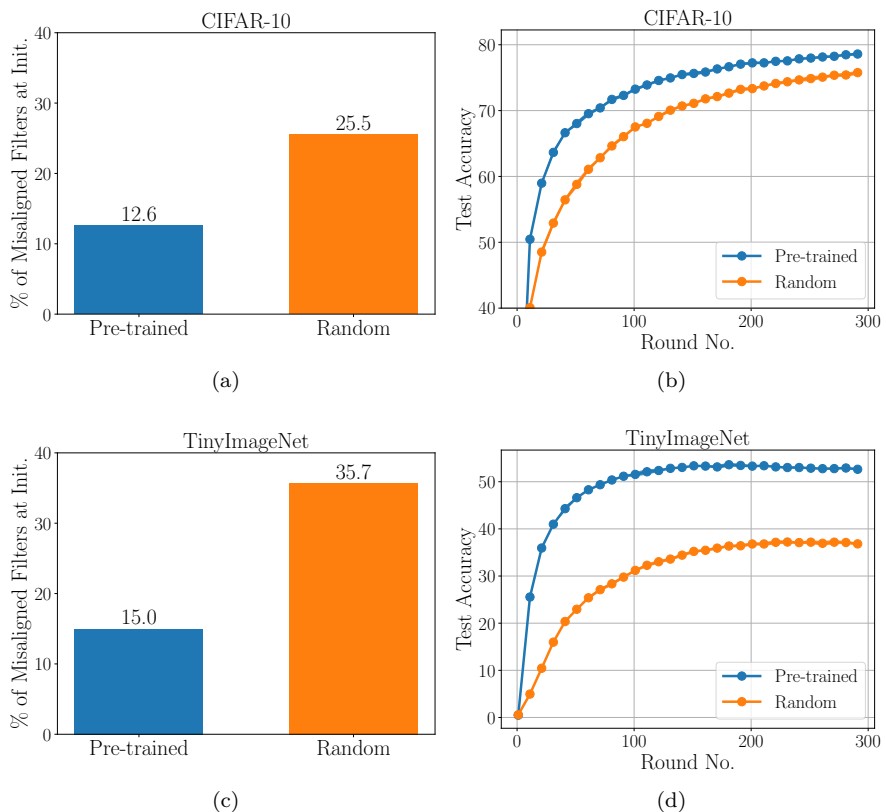

Figure 5: The percentage of misaligned filters (see Equation (8) and test accuracy for different initializations on CIFAR-10 (Figure 5a and Figure 5b) and TinyImageNet (Figure 5c and Figure 5d). As the complexity of the signal information in the data grows from CIFAR-10 to TinyImageNet, we see a sharp increase in the ratio of misaligned filters for random initialization, explaining why pre-trained initialization offers larger improvements for TinyImageNet.

As a result, in Theorem 2 $A_j = [m]$ for $j \in \{\pm 1\}$ and in the benign overfitting regime ($\text{SNR}^2 \gtrsim 1/\sqrt{nd}$), we recover the centralized result $L_{\mathcal{D}}^{0-1}(\mathbf{W}^{(T)}) \leq \exp(-n\text{SNR}^2/d)$ (Kou et al., 2023, Theorem 4.2). Hence, the adverse effects of cross-client heterogeneity are mitigated with pre-trained initialization.

## 4  Experiments

In this section we provide empirical results showing how our insights from Section 3 extend to practical FL training on real world datasets with deep CNN models. Unless specified otherwise, we use the ResNet18 model He et al. (2016) in all our experiments and split the data across 20 clients using the Dirichlet sampling scheme Hsu et al. (2019) with non-iid parameter $\alpha = 0.3$. For pre-training, we use a ResNet18 pre-trained on ImageNet Russakovsky et al. (2015), available in PyTorch Paszke et al. (2019). We emphasize that these results are primarily qualitative and indicative, as several simplifying assumptions from our theoretical analysis such as binary classification and a two-layer CNN do not strictly hold in this setting. Nevertheless, we observe that the key insights derived from theory continue to hold empirically. For completeness, we also perform experiments on a small binary-class subset that directly mirrors the theoretical setup and provide additional details in Appendix F.

**Empirical Measure of Misalignment.**   Measuring filter alignment for deep CNNs is challenging since we cannot explicitly characterize the signal information present in real world datasets and furthermore different layers will learn the signal at different levels of granularity. Nonetheless, our theoretical findings suggest that given sufficient number of training rounds, filters will be aligned with the signal (see Section 3) and once a filter is aligned, the sign of the output produced by the filter with respect to the signal does not

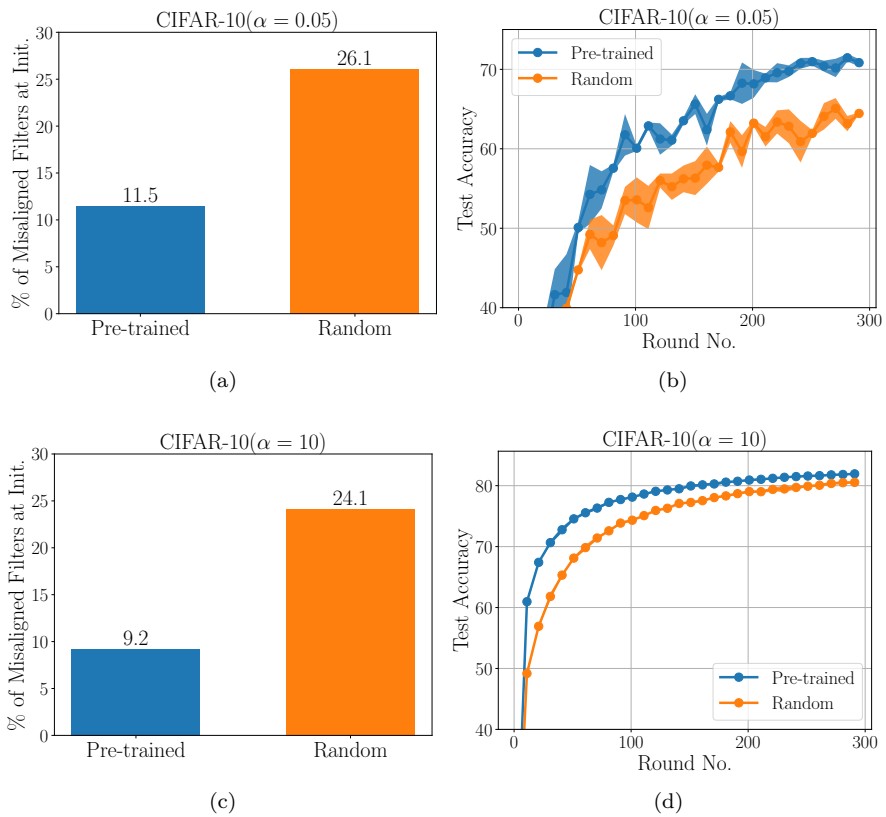

Figure 6: The percentage of misaligned filters (see Equation (8)) and test accuracy for different initializations on CIFAR-10 with $\alpha = 0.05$ heterogeneity (Figure 6a and Figure 6b) and $\alpha = 10$ heterogeneity (Figure 6c and Figure 6d). Although the percentage of misaligned filters does not vary significantly across the two settings for both initializations (signal information is the same in both settings), pre-training offers more improvement in the higher heterogeneity setting ($\alpha = 0.05$), as suggested by our theoretical analysis.

change, i.e, if $\langle \mathbf{w}_{j,r}^{(t)}, j\boldsymbol{\mu} \rangle > 0$ then $\text{sign}(\langle \mathbf{w}_{j,r}^{(t')}, \boldsymbol{\mu} \rangle) = \text{sign}(\langle \mathbf{w}_{j,r}^{(t)}, \boldsymbol{\mu} \rangle)$, for all $t' \geq t$. Therefore, we propose to use the sign of the output produced by a filter at the end of training as a reference for alignment at any given round. Formally, let $\mathbf{W}^{(0)}, \mathbf{W}^{(1)} \cdots \mathbf{W}^{(T)}$ be the sequence of iterates produced by federated training and let $\mathcal{F}(\mathbf{w}, \mathbf{x}) = [\langle \mathbf{w}, \mathbf{x}(1) \rangle, \langle \mathbf{w}, \mathbf{x}(2) \rangle, \dots \langle \mathbf{w}, \mathbf{x}(p) \rangle] \in \mathbb{R}^p$ be the feature map vector generated by filter $\mathbf{w}$ for input $\mathbf{x}$. For a given batch of data $\mathcal{B}$, we define the empirical measure of alignment of filter $\mathbf{w}^{(t)}$ relative to $\mathbf{w}^{(T)}$ as follows:

$$\mathcal{A}(\mathbf{w}^{(t)}) := \sum_{x \in \mathcal{B}, l \in [p]} \text{sign}(\mathcal{F}_l(\mathbf{w}^{(t)}, \mathbf{x}))\text{sign}(\mathcal{F}_l(\mathbf{w}^{(T)}, \mathbf{x})). \tag{8}$$

We say that the weight $\mathbf{w}^{(t)}$ at round $t$ is misaligned if $\mathcal{A}(\mathbf{w}^{(t)}) < 0$, because this implies that the sign of the output produced by the filter $\mathbf{w}$ at round $t$ eventually changed for a majority of the inputs, hence indicating that the filter was misaligned at round $t$. We compute this measure over a batch of data to account for signal information coming from different classes of data as well as reduce the impact of noise in the data.

**Measuring Misalignment on Real World Datasets with Varying Signal Information.** In this experiment our goal is to empirically demonstrate that (a) pre-trained initialization leads to much fewer number of misaligned filters than random initialization and (b) the number of misaligned filters for random initialization increases as we increase the complexity of the signal. To demonstrate this, we consider federated training on the **1. CIFAR-10** Krizhevsky (2009) and **2. TinyImageNet** Le & Yang (2015) datasets. Figure 5 shows the test accuracy and percentage of misaligned filter across training rounds for both datasets with pre-trained and random initialization. Firstly, we see that the percentage of misaligned filters is $2 - 3\times$

smaller when starting from a pre-trained initialization compared to a random initialization. Furthermore, as the complexity of the signal information in the dataset increases (CIFAR-10 < TinyImageNet), we see a sharp increase in the percentage of misaligned filters (25% to 40%) for random initialization. In contrast, with pre-trained initialization, the percentage of misaligned filters remains less than 15% across datasets leading to a larger improvement in test accuracy for TinyImageNet. These results align with our theoretical findings: as the ratio of misaligned filters increases, the benefits of pre-training become more pronounced.

**Measuring Misalignment with Varying Heterogeneity Levels.** We extend the experiment in Figure 5 conducted on CIFAR-10 with $\alpha = 0.3$ Dirichlet heterogeneity to other levels of heterogeneity **1.** $\alpha = 0.05$ which is an extreme non-IID split and **2.** $\alpha = 10$ which can be thought of as close to IID split. Figure 6 shows the test accuracy and percentage of misaligned filters plots for these two heterogeneity levels with pre-trained and random initialization. We observe that in both cases the percentage of misaligned filters remains approximately 25% with random initialization and 10% with pre-trained initialization, regardless of the level of heterogeneity. However, as heterogeneity increases, the improvement in test accuracy provided by pre-trained initialization becomes more pronounced. This trend is consistent with our theoretical analysis in Theorem 2, which suggests that the percentage of misaligned filters will have a greater impact on test performance as data heterogeneity increases.

## 5 Conclusion and Future Work

In this work we provide a deeper theoretical explanation for why pre-training can drastically reduce the adverse effects of non-IID data in FL by studying the class of two layer CNN models under a signal-noise data model. Our analysis shows that the reduction in test accuracy seen in non-IID FL compared to IID FL is only caused by filters that are misaligned at initialization. When starting from a pre-trained model we expect most of the filters to be already aligned with the signal thereby reducing the effect of heterogeneity and leading to a higher ratio of signal learning to noise memorization. This is corroborated by experiments on synthetic setup as well as more practical FL training tasks. Our work also opens up several avenues for future work as we discuss below.

**Extension to Deeper CNNs and Transformer Architectures.** For deeper CNNs, we anticipate that we will need a hierarchical feature model (Li & Li, 2024; Allen-Zhu & Li, 2023b; Wang et al., 2023b) to capture alignment in deeper layers. Intuitively, while deeper architectures provide the capacity to learn more complex hierarchical signals, aligning a filter with the signal will also progressively become harder across layers with the alignment of the $L$-th layer potentially depending on the alignment of the $(L - 1)$-th layer. In non-IID FL, this suggests that feature learning deteriorates as we move from the base layer toward the final layer. This intuition is supported by empirical evidence (see experiment in Table 1 in Yu et al. (2022b)) and explains why pre-trained initialization offers greater benefits for deeper networks compared to shallower ones.

For extending to Transformers, recall that alignment of a convolutional filter is defined via the sign of its inner product with an input patch containing the signal (see Definition 1). For Transformers, the convolution operation is replaced by the self-attention mechanism. However, the core principle remains: intermediate representations are computed as inner products between the previous-layer embeddings and the current-layer attention weights. Specifically, parameterizing a self-attention layer by $(Q, K, V) \in \mathbb{R}^d$ and input $z \in \mathbb{R}^{N \times d}$, we have:

$$q = zQ, \quad k = zK, \quad v = zV \quad \text{(intermediate representations)} \tag{9}$$

$$z' = \text{softmax}\left(\frac{qk^\top}{\sqrt{d}}\right)v \quad \text{(self-attention aggregation).} \tag{10}$$

Notice that each column of $Q$, $K$, and $V$ is involved in an inner product with the input $z$, analogous to filters in CNNs. Thus, our notion of filter alignment and its empirical measurement (Equation (8)) can naturally be extended to these matrices. This connection provides a promising foundation for generalizing alignment analysis to Transformer-based architectures. Lastly, we note that recent work has also begun to explain benign overfitting in Transformer models (Jiang et al., 2024; Frei & Vardi, 2024), which could serve as a natural starting point for extending our analysis.

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

# Appendix

# A   Additional Related Work

**Use of Pre-Trained Models in Federated Learning.**   Tan et al. (2022) explore the benefit of using pre-trained models in FL by proposing to use multiple fixed pre-trained backbones as the encoder model at each client and using contrastive learning to extract useful shared representations. Zhuang et al. (2023) discuss the opportunities and challenges of using large foundation models for FL including the high communication and computation cost.  One solution to this as proposed by Legate et al. (2024) is that instead of full fine-tuning as done in Chen et al. (2022); Nguyen et al. (2022), we can just fine-tune the last layer. Specifically Legate et al. (2024) proposes a two-stage approach to federated fine-tuning by first fine-tuning the head and then doing a full-finetuning. This approach is inspired by results in the centralized setting Kumar et al. (2022) which show that in some case fine-tuning can distort the pre-trained features. Fanì et al. (2023) also study the problem of fine-tuning just the last layer in a federated setting by replacing the softmax classifier with a ridge-regression classifier which enables them to compute a closed form expression for the last layer weights.

There has also been some recent work on exploring the benefit of pre-training for federated natural language processing tasks including the use of Large Language Models (LLMs). Wang et al. (2023a) discuss how to leverage the power of pre-trained LLMs for private on-device fine-tuning of language models. Specifically, Wang et al. (2023a) proposes a distribution matching approach to select public data that is closest to private data and then use this selected public data to train the on-device language model. Zhang et al. (2023) propose to first pre-train on synthetic data to construct the initialization point followed by federated fine-tuning. Hou et al. (2024) propose that clients send DP information to the server which then uses this information to generate synthetic data and fine-tune centrally on this synthetic data. Liu & Miller (2020) discuss the challenges of pre-training and fine-tuning BERT in federated manner using clinical notes from multiple silos without data transfer. Tian et al. (2022) propose to pre-train a BERT model in a federated manner in a more general setting and show that their pre-trained model can retain accuracy on the GLUE (Wang et al., 2018) dataset without sacrificing client privacy. Xu et al. (2023b) pretrain production on-device language models on public web data before fine-tuning in federated learning with differential privacy, and Wu et al. (2024) later replace the pretraining data with data synthesized by LLMs. Gupta et al. (2022) propose a defense using pre-trained models to prevent an attacker from recovering multiple sentences from gradients in the federated training of the language modeling task.

**Importance of Initialization for Private Optimization.**   We note that an orthogonal line of work has explored the benefits of starting from a pre-trained model when doing differentially private optimization Dwork et al. (2006) and seen similar striking improvement in accuracy De et al. (2022); Li et al. (2022b); Yu et al. (2022a); Xu et al. (2023a), as we see in the heterogeneous FL setting. Ganesh et al. (2023) study this phenomenon for a stylized mean estimation problem and show that public pre-training can help the model start from a good loss basin which is otherwise hard to achieve with private noisy optimization. Li et al. (2022a) study differentially private convex optimization and show that starting from a pre-trained model can leads to dimension independent convergence guarantees. Specifically Li et al. (2022a) define the notion of restricted Lipschitz continuity and show that when gradients are low rank most of the restricted Lispchitz coefficients will be zero. Ye et al. (2023) studies the impact of different random initializations on the privacy bound when training overparameterized neural networks and shows that for some initializations (LeCun LeCun et al. (2012), Xavier Glorot & Bengio (2010)) the privacy bound improves with increasing depth while for other initializations (He He et al. (2015), NTK Allen-Zhu & Li (2023a)) it degrades with increasing depth.

**Generalization performance in Federated Learning.**   Several existing works have studied the generalization performance of FL in different settings Cheng et al. (2021); Gholami & Seferoglu (2024); Huang et al. (2023); Yuan et al. (2021). Some of the initial works either provide results independent of the algorithm being used Mohri et al. (2019); Hu et al. (2022); Sun & Wei (2022), or only study convex losses Chen et al. (2021); Fallah et al. (2021). Barnes et al. (2022); Sefidgaran et al. (2022) derive information-theoretic bounds, but these bounds require specific forms of loss functions and cannot capture effects of heterogeneity. Huang et al. (2021) study the generalization of FedAvg on wide two-layer ReLU networks with homogeneous data. Collins et al. (2022) studies FedAvg under multi-task linear representation learning setting. In Sun et al. (2024),

the authors have demonstrated the impact of data heterogeneity on the generalization performance of some popular FL algorithms.

## B  Theory Notation and Preliminaries

We follow a similar notation as Kou et al. (2023) in most of the analysis.

Table 1: Summary of notation

| Symbol | Description |
|---|---|
| $j \in \{-1, 1\}$ | Layer index |
| $m$ | Number of filters |
| $d$ | Dimension of filter |
| $r \in [m]$ | Filter Index |
| $K$ | Number of clients |
| $k \in [K]$ | Client index |
| $N$ | Number of datapoints at each client |
| $i \in [N]$ | Datapoint index |
| $n = KN$ | Global dataset size |
| $y_{k,i} \in \{1, -1\}$ | Label of $i$-th datapoint at $k$-th client |
| $\boldsymbol{\mu}$ | Signal vector |
| $\sigma_p^2$ | Variance of Gaussian noise |
| $\boldsymbol{\xi}_{k,i}$ | Noise vector for $k$-th client and $i$-th datapoint |
| $\eta$ | Local learning rate |
| $\tau$ | Number of local steps |
| $\ell(z, \hat{z}) = \log(1 + \exp(-z \cdot \hat{z}))$ | Cross-entropy loss function |
| $\sigma(z) = \max(0, z)$ | ReLU function |
| $\sigma'(z) = \mathbb{1}(z \geq 0)$ | Derivative of ReLU function |
| $t$ | Round index |
| $s$ | Iteration index |
| $h$ | Heterogeneity parameter |
| $\mathrm{SNR} := \|\boldsymbol{\mu}\|_2 / \sigma_p \sqrt{d}$ | Signal to Noise Ratio |
| $\mathbf{W}_k^{(\cdot,\cdot)}$ | Parameterized weights of the $k$-th client |
| $\mathbf{w}_{j,r,k}^{(\cdot,\cdot)}$ | $(j, r)$-th filter weight of the $k$-th client |
| $\gamma_{j,r,k}^{(\cdot,\cdot)}$ | Local signal co-efficient for $k$-th client |
| $\rho_{j,r,k,i}^{(\cdot,\cdot)}$ | Local noise coefficient for $k$-th client and $i$-th datapoint |
| $\overline{\rho}_{j,r,k,i}^{(\cdot,\cdot)}$ | Positive local noise coefficient for $k$-th client and $i$-th datapoint |
| $\underline{\rho}_{j,r,k,i}^{(t,s)}$ | Negative local noise coefficient for $k$-th client and $i$-th datapoint |
| $\ell'^{(\cdot,\cdot)}_{k,i}$ | Shorthand for $-1/\left(1 + \exp(y_{k,i} f(\mathbf{W}_k^{(\cdot,\cdot)}, \mathbf{x}_{k,i}))\right)$ which is the derivative of cross-entropy loss for $i$-th datapoint at $k$-th client |
| $\mathbf{W}^{(\cdot)}$ | Parameterized weight vector of the global model |
| $\mathbf{w}_{j,r}^{(\cdot)}$ | $j, r$-th filter weight of the global model |
| $\Gamma_{j,r}^{(\cdot)}$ | Global signal co-efficient |
| $P_{j,r,k,i}^{(\cdot)}$ | Global noise coefficient for $(k, i)$-th datapoint |
| $\overline{P}_{j,r,k,i}^{(\cdot)}$ | Positive global noise coefficient for $(k, i)$-th datapoint |
| $\underline{P}_{j,r,k,i}^{(\cdot)}$ | Negative global noise coefficient for $(k, i)$-th client datapoint |

### B.1 Local Model Update

Using local GD updates in equation 5 to minimize the local loss function in equation 3, the local model update for the $(j, r)$ filter at client $k$ in round $t$ can be written as,

$$
\begin{aligned}
\mathbf{w}_{j,r,k}^{(t,\tau)} &= \mathbf{w}_{j,r}^{(t)} - \frac{\eta}{Nm} \sum_{s=0}^{\tau-1} \sum_{i \in [N]} \ell_{k,i}'^{(t,s)} \cdot \sigma'\left(\langle \mathbf{w}_{j,r,k}^{(t,s)}, \boldsymbol{\xi}_{k,i} \rangle\right) \cdot j y_{k,i} \boldsymbol{\xi}_{k,i} \\
&\quad - \frac{\eta}{Nm} \sum_{s=0}^{\tau-1} \sum_{i \in [N]} \ell_{k,i}'^{(t,s)} \cdot \sigma'\left(\langle \mathbf{w}_{j,r,k}^{(t,s)}, y_{k,i}\boldsymbol{\mu} \rangle\right) \cdot j\boldsymbol{\mu} \\
&= \mathbf{w}_{j,r}^{(t)} + j\gamma_{j,r,k}^{(t,\tau)} \cdot \|\boldsymbol{\mu}\|_2^{-2} \cdot \boldsymbol{\mu} + \sum_{i \in [N]} \rho_{j,r,k,i}^{(t,\tau)} \cdot \|\boldsymbol{\xi}_{k,i}\|_2^{-2} \cdot \boldsymbol{\xi}_{k,i}
\end{aligned}
\tag{11}
$$

where, we use $\mathbf{w}_{j,r,k}^{(t,0)} \triangleq \mathbf{w}_{j,r}^{(t)}$. Further, we define

$$
\gamma_{j,r,k}^{(t,\tau)} \triangleq -\frac{\eta}{Nm} \sum_{s=0}^{\tau-1} \sum_{i \in [N]} \ell_{k,i}'^{(t,s)} \cdot \sigma'\left(\langle \mathbf{w}_{j,r,k}^{(t,s)}, y_{k,i}\boldsymbol{\mu} \rangle\right) \cdot \|\boldsymbol{\mu}\|_2^2,
\tag{12}
$$

$$
\rho_{j,r,k,i}^{(t,\tau)} \triangleq -\frac{\eta}{Nm} \sum_{s=0}^{\tau-1} \ell_{k,i}'^{(t,s)} \cdot \sigma'\left(\langle \mathbf{w}_{j,r,k}^{(t,s)}, \boldsymbol{\xi}_{k,i} \rangle\right) \cdot \|\boldsymbol{\xi}_{k,i}\|_2^2 \cdot j y_{k,i}.
\tag{13}
$$

which respectively, denote the local signal $(\gamma_{j,r,k}^{(t,\tau)})$ and local noise $(\{\rho_{j,r,k,i}^{(t,\tau)}\}_i)$ components of $\mathbf{w}_{j,r,k}^{(t,\tau)}$. We also define $\overline{\rho}_{j,r,k,i}^{(t,\tau)} = \rho_{j,r,k,i}^{(t,\tau)} \mathbb{1}\left(\rho_{j,r,k,i}^{(t,\tau)} \geq 0\right)$ and $\underline{\rho}_{j,r,k,i}^{(t,\tau)} = \rho_{j,r,k,i}^{(t,\tau)} \mathbb{1}\left(\rho_{j,r,k,i}^{(t,\tau)} < 0\right)$, where $\mathbb{1}(\,\cdot\,)$ denotes the indicator function, and which can alternatively be written as

$$
\overline{\rho}_{j,r,k,i}^{(t,\tau)} = -\frac{\eta}{Nm} \sum_{s=0}^{\tau-1} \ell_{k,i}'^{(t,s)} \cdot \sigma'\left(\langle \mathbf{w}_{j,r,k}^{(t,s)}, \boldsymbol{\xi}_{k,i} \rangle\right) \cdot \|\boldsymbol{\xi}_{k,i}\|_2^2 \cdot \mathbb{1}\left(y_{k,i} = j\right),
\tag{14}
$$

$$
\underline{\rho}_{j,r,k,i}^{(t,\tau)} = \frac{\eta}{Nm} \sum_{s=0}^{\tau-1} \ell_{k,i}'^{(t,s)} \cdot \sigma'\left(\langle \mathbf{w}_{j,r,k}^{(t,s)}, \boldsymbol{\xi}_{k,i} \rangle\right) \cdot \|\boldsymbol{\xi}_{k,i}\|_2^2 \cdot \mathbb{1}\left(y_{k,i} = -j\right).
\tag{15}
$$

### B.2 Proof of Proposition 1

The global model update at round $t+1$ can be written as

$$
\begin{aligned}
\mathbf{w}_{j,r}^{(t+1)} &= \sum_{k=1}^{K} \frac{1}{K} \mathbf{w}_{j,r,k}^{(t,\tau)} \\
&= \mathbf{w}_{j,r}^{(t)} + \frac{j}{K} \sum_{k=1}^{K} \gamma_{j,r,k}^{(t,\tau)} \cdot \|\boldsymbol{\mu}\|_2^{-2} \cdot \boldsymbol{\mu} + \sum_{k=1}^{K} \sum_{i \in [N]} \frac{1}{K} \rho_{j,r,k,i}^{(t,\tau)} \cdot \|\boldsymbol{\xi}_{k,i}\|_2^{-2} \cdot \boldsymbol{\xi}_{k,i}.
\end{aligned}
\tag{16}
$$

Mimicking the signal-noise decomposition in equation 11, we can define a similar decomposition for the global model as follows.

$$
\mathbf{w}_{j,r}^{(t)} = \mathbf{w}_{j,r}^{(0)} + j\Gamma_{j,r}^{(t)} \cdot \|\boldsymbol{\mu}\|_2^{-2} \cdot \boldsymbol{\mu} + \sum_{k=1}^{K} \sum_{i \in [N]} P_{j,r,k,i}^{(t)} \cdot \|\boldsymbol{\xi}_{k,i}\|_2^{-2} \cdot \boldsymbol{\xi}_{k,i}.
\tag{17}
$$

### B.3 Co-efficient Update Equations

Comparing with equation 16, we have the following recursive update for the global signal and noise coefficients using $n = KN$.

$$
\begin{aligned}
\Gamma_{j,r}^{(t+1)} &= \Gamma_{j,r}^{(t)} + \sum_{k=1}^{K} \frac{1}{K} \gamma_{j,r,k}^{(t,\tau)} \\
&= \Gamma_{j,r}^{(t)} - \frac{\eta}{nm} \sum_{k=1}^{K} \sum_{i \in [N]} \sum_{s=0}^{\tau-1} \ell_{k,i}^{\prime (t,s)} \cdot \sigma' \left( \langle \mathbf{w}_{j,r,k}^{(t,s)}, y_{k,i} \boldsymbol{\mu} \rangle \right) \cdot \|\boldsymbol{\mu}\|_2^2
\end{aligned}
\tag{18}
$$

$$
\begin{aligned}
P_{j,r,k,i}^{(t+1)} &= P_{j,r,k,i}^{(t)} + \frac{1}{K} \rho_{j,r,k,i}^{(t,\tau)} \\
&= P_{j,r,k,i}^{(t)} - \frac{\eta}{nm} \sum_{s=0}^{\tau-1} \ell_{k,i}^{\prime (t,s)} \cdot \sigma' \left( \langle \mathbf{w}_{j,r,k}^{(t,s)}, \boldsymbol{\xi}_{k,i} \rangle \right) \cdot \|\boldsymbol{\xi}_{k,i}\|_2^2 \cdot j y_{k,i}.
\end{aligned}
\tag{19}
$$

Analogously, we can also define the positive and negative global noise coefficients,

$$
\overline{P}_{j,r,k,i}^{(t+1)} = \overline{P}_{j,r,k,i}^{(t)} - \frac{\eta}{nm} \sum_{s=0}^{\tau-1} \ell_{k,i}^{\prime (t,s)} \cdot \sigma' \left( \langle \mathbf{w}_{j,r,k}^{(t,s)}, \boldsymbol{\xi}_{k,i} \rangle \right) \cdot \|\boldsymbol{\xi}_{k,i}\|_2^2 \, \mathbb{1} \left( y_{k,i} = j \right)
\tag{20}
$$

and,

$$
\underline{P}_{j,r,k,i}^{(t+1)} = \underline{P}_{j,r,k,i}^{(t)} + \frac{\eta}{nm} \sum_{s=0}^{\tau-1} \ell_{k,i}^{\prime (t,s)} \cdot \sigma' \left( \langle \mathbf{w}_{j,r,k}^{(t,s)}, \boldsymbol{\xi}_{k,i} \rangle \right) \cdot \|\boldsymbol{\xi}_{k,i}\|_2^2 \, \mathbb{1} \left( y_{k,i} = -j \right).
\tag{21}
$$

**Lemma 4.** *(Measuring local and global signal coefficient)*

*From equation 11, it follows that*

$$
\langle \mathbf{w}_{j,r,k}^{(t,s)} - \mathbf{w}_{j,r}^{(t)}, y_{k,i} \boldsymbol{\mu} \rangle = j y_{k,i} \gamma_{j,r,k}^{(t,s)}.
\tag{22}
$$

*and from equation 17, it follows that*

$$
\langle \mathbf{w}_{j,r}^{(t)} - \mathbf{w}_{j,r}^{(0)}, \boldsymbol{\mu} \rangle = j \Gamma_{j,r}^{(t)}.
\tag{23}
$$

Since $\{\Gamma_{j,r}^{(t)}\}_t$ are non-negative and non-decreasing in $t$, the global weights $\{\mathbf{w}_{j,r}^{(t)}\}_r$ become increasing aligned with the *actual* signal $y_{k,i} \boldsymbol{\mu}$ corresponding to the filters $j = y_{k,i}$. Similarly, as $\{\gamma_{j,r,k}^{(t,s)}\}_t$ are non-negative and non-decreasing in $s$ for fixed $t$, the local weights $\{\mathbf{w}_{y_{k,i},r,k}^{(t,s)}\}_r$ become increasing aligned with the signal $y_{k,i} \boldsymbol{\mu}$ corresponding to the filters $j = y_{k,i}$.

## C  Training Error Convergence of FedAvg with Random Initialization

For the sake of completeness, we state the conditions used in our analysis (Condition 1) in full detail.

**Assumptions.**  Let $\epsilon$ be a desired training error threshold and $\delta \in (0,1)$ be some failure probability. Let $T^* = \frac{1}{\eta} \text{poly}(\epsilon^{-1}, m, n, d)$ be the maximum admissible rounds.

Suppose there exists a sufficiently large constant $C$, such that the following hold.

**Assumption 1.** *Dimension $d$ is sufficiently large, i.e.,*

$$
d \geq C \max \left\{ \frac{n \|\boldsymbol{\mu}\|_2^2 \log(T^* \tau)}{\sigma_p^2}, n^2 \log(nm/\delta)(\log(T^* \tau))^2 \right\}.
$$

**Assumption 2.** *Training sample size $n$ and neural network width $m$ satisfy*

$$m \geq C \log(n/\delta), n \geq C \log(m/\delta).$$

**Assumption 3.** *The norm of the signal satisfies,*

$$\|\boldsymbol{\mu}\|_2^2 \geq C \sigma_p^2 \log(n/\delta).$$

**Assumption 4.** *Standard deviation of Gaussian initialization is sufficiently small, i.e.,*

$$\sigma_0 \leq \frac{1}{C} \min \left\{ \frac{\sqrt{n}}{\sigma_p d \tau}, \frac{1}{\sqrt{\log(m/\delta)} \|\boldsymbol{\mu}\|_2} \right\}.$$

**Assumption 5.** *Learning rate is sufficiently small, i.e.,*

$$\eta \leq \frac{1}{C} \min \left\{ \frac{nm\sqrt{\log(m/\delta)}}{\sigma_p^2 d}, \frac{1}{\|\boldsymbol{\mu}\|_2^2}, \frac{1}{\sigma_p^2 d} \right\}.$$

The assumptions are primarily used to ensure that the model is sufficiently overparameterized, i.e., training loss can be made arbitrarily small, and that we do not begin optimization from a point where the gradient is already zero or unbounded. We provide a more intuitive reasoning behind each of the assumptions below:

- *Bounded number of communication rounds:* This is needed to ensure that the magnitude of filter weights remains bounded throughout training since they grow logarithmically with the number of updates (see Theorem 3). We note that this is quite a mild condition since the max rounds can have polynomial dependence on $1/\epsilon$ where $\epsilon$ is our desired training error.

- *Dimension $d$ is sufficiently large:* This is needed to ensure that the model is sufficiently overparameterized and the training loss can be made arbitrarily small. Recall that our input $\mathbf{x}$ consists of a signal component $\boldsymbol{\mu} \in \mathbb{R}^d$ that is common across all datapoints and noise component $\boldsymbol{\xi} \in \mathbb{R}^d$ that is independently drawn from $\mathcal{N}(0, \sigma_p^2 \cdot \boldsymbol{I})$. Having a sufficiently large $d$ ensures that the correlation between any two noise vectors, i.e. $\langle \boldsymbol{\xi}, \boldsymbol{\xi}' \rangle / \|\boldsymbol{\xi}\|^2$ is not too large. Otherwise if the correlation between two noise vectors is large and negative, then minimizing the loss on one data point could end up increasing the loss on another training point which complicates convergence and prevents loss from becoming arbitrarily small.

- *Training set size and network width is sufficiently large:* The condition ensures that a sufficient number of filters get activated at initialization with high probability (see Lemma 6 and Lemma 7) and prevents cases where the initial gradient is zero. The condition on training set size also ensures that there are a sufficient number of datapoints with negative and positive labels (see Lemma 8).

- *Standard deviation of Gaussian random initialization is sufficiently small:* This condition is needed to ensure that the magnitude of the initial correlation between the filter weights and the signal and noise components, i.e $|\langle \mathbf{w}_{j,r}^{(0)}, \boldsymbol{\mu} \rangle|, |\langle \mathbf{w}_{j,r}^{(0)}, \boldsymbol{\xi} \rangle|$ is not too large. This simplifies the analysis and prevents cases where none of the filters get activated at initialization (see Lemma 21). It also ensures that after some number of rounds all filters get aligned with the signal (see Lemma 30).

- *Norm of signal is larger than noise variance:* This condition is needed to ensure that all misaligned filters at initialization eventually become aligned with the signal after some rounds (see Lemma 30). This allows us to derive a meaningful bound on test performance that is not dominated by noise memorization.

- *Learning rate is sufficiently small:* This is a standard condition to ensure that gradient descent does not diverge. The conditions are derived from ensuring that the signal and noise coefficient remain bounded in the first stage of training and that the loss decreases monotonically in every round in the second stage of training.

For ease of reference, we restate Theorem 1 below.

**Theorem** (Training Loss Convergence)**.** *Let $T_1 = \mathcal{O}\left(\frac{mn}{\eta\sigma_p^2 d\tau}\right)$. With probability $1 - \delta$ over the random initialization, for all $T_1 \leq T \leq T^*$ we have,*

$$\frac{1}{T - T_1 + 1} \sum_{t=T_1}^{T} L(\mathbf{W}^{(t)}) \leq \frac{\left\|\mathbf{W}^{(T_1)} - \mathbf{W}^*\right\|_2^2}{\eta(T - T_1 + 1)} + \epsilon.$$

*Therefore we can find an iterate with training error smaller than $2\epsilon$ within $T = T_1 + \left\|\mathbf{W}^{(T_1)} - \mathbf{W}^*\right\|_2^2/(\eta\epsilon) = \mathcal{O}\left(\frac{mn}{\eta\sigma_p^2 d\tau}\right) + \mathcal{O}\left(\frac{mn\log(\tau/\epsilon)}{\eta\sigma_p^2 d\epsilon}\right)$ rounds.*

### C.1 Preliminary Lemmas

**Lemma 5.** *(Lemma B.4 in Cao et al. (2022)) Suppose that $\delta > 0$ and $d = \Omega\left(\log(4n/\delta)\right)$. Then with probability at least $1 - \delta$,*

$$\sigma_p^2 d/2 \leq \|\boldsymbol{\xi}_{k,i}\|_2^2 \leq 3\sigma_p^2 d/2,$$

$$|\langle\boldsymbol{\xi}_{k,i}, \boldsymbol{\xi}_{k',i'}\rangle| \leq 2\sigma_p^2\sqrt{d\log(6n^2/\delta)},$$

*for all $k, k' \in [K]$, $i, i' \in [N]$, and $(k,i) \neq (k',i')$.*

**Lemma 6.** *(Lemma B.5 in Kou et al. (2023)). Suppose that $d = \Omega\left(\log(mn/\delta)\right)$, $m = \Omega\left(\log(1/\delta)\right)$. Then with probability at least $1 - \delta$,*

$$\sigma_0^2 d/2 \leq \left\|\mathbf{w}_{j,r}^{(0)}\right\|_2^2 \leq 3\sigma_0^2 d/2,$$

$$\left|\langle\mathbf{w}_{j,r}^{(0)}, \boldsymbol{\mu}\rangle\right| \leq \sqrt{2\log(12m/\delta)}\cdot\sigma_0\|\boldsymbol{\mu}\|_2, \left|\langle\mathbf{w}_{j,r}^{(0)}, \boldsymbol{\xi}_{k,i}\rangle\right| \qquad \leq 2\sqrt{\log(12mn/\delta)}\cdot\sigma_0\sigma_p\sqrt{d},$$

*for all $r \in [m]$, $j \in \{\pm 1\}$, $k \in [K]$ and $i \in [N]$.*

**Lemma 7.** *(Lemma B.6 in Kou et al. (2023)). Let $S_{k,i}^{(0)} = \left\{r \in [m] : \langle\mathbf{w}_{y_{k,i},r}^{(0)}, \boldsymbol{\xi}_{k,i}\rangle \geq 0\right\}$. Suppose $\delta > 0$ and $m \geq 50\log(2n/\delta)$. Then with probability at least $1 - \delta$,*

$$\left|S_{k,i}^{(0)}\right| \geq 0.4m, \forall i \in [n].$$

**Lemma 8.** *(Lemma B.7 in Kou et al. (2023)) Let $\tilde{S}_{j,r}^{(0)} = \left\{k \in [K], i \in [N] : y_{k,i} = j, \langle\mathbf{w}_{j,r}^{(0)}, \boldsymbol{\xi}_{k,i}\rangle \geq 0\right\}$. Suppose $\delta > 0$ and $n \geq 32\log(4m/\delta)$. Then with probability at least $1 - \delta$,*

$$\left|\tilde{S}_{j,r}^{(0)}\right| \geq n/8, \forall i \in [n].$$

**Lemma 9.** *Let $D_j = \{k \in [K], i \in [N] : y_{k,i} = j\}$. Suppose $\delta > 0$ and $n \geq 8\log(4/\delta)$. Then with probability at least $1 - \delta$,*

$$|D_j| \geq \frac{n}{4}, \forall j \in \{\pm 1\}.$$

*Proof.* We have $|D_j| = \sum_{k,i} \mathbb{1}\left(y_{k,i} = j\right)$ and therefore $\mathbb{E}|D_j| = \sum_{k,i} \mathbb{P}(y_{k,i} = j) = n/2$. Applying Hoeffding's inequality we have with probability $1 - 2\delta$,

$$\left|\frac{|D_j|}{n} - \frac{1}{2}\right| \leq \sqrt{\frac{\log(4/\delta)}{2n}}.$$

Now if $n \geq 8\log(4/\delta)$, by applying union bound, we have with probability at least $1 - \delta$,

$$|D_j| \geq \frac{n}{4}, \forall j \in \{\pm 1\}.$$

$\square$

## C.2 Bounding the Scale of Signal and Noise Memorization Coefficients

Our first goal is to show that the coefficients of the global model, i.e., $\Gamma_{j,r}^{(t)}$, $\overline{P}_{j,r,k,i}^{(t)}$ and $\left|\underline{P}_{j,r,k,i}^{(t)}\right|$ are bounded as $\mathcal{O}\left(\log(T^*\tau)\right)$. To do so, we look at a *virtual* iteration index given by $v = 0, 1, 2, 3, \ldots, T^*\tau - 1$. For any $v$, we can define the filter weights at virtual iteration $v$ in terms of the filter weights we have seen so far. In particular,

$$\widetilde{\mathbf{w}}_{j,r,k}^{(v)} \triangleq \mathbf{w}_{j,r,k}^{\left(\lfloor \frac{v}{\tau} \rfloor, v \bmod \tau\right)}.$$

We also define the following *virtual sequence of local coefficients* which will be used in our proof. Let $\mathbb{G}_{j,r,k}^{(0)} = 0, \overline{\mathbb{P}}_{j,r,k,i}^{(0)} = 0, \underline{\mathbb{P}}_{j,r,k,i}^{(0)} = 0$. We have the following update equation for $\mathbb{G}_{j,r,k}^{(v)}, \overline{\mathbb{P}}_{j,r,k,i}^{(v)}$ and $\underline{\mathbb{P}}_{j,r,k,i}^{(v)}$ for $v \geq 1$.

$$\mathbb{G}_{j,r,k}^{(v)} = \begin{cases} \mathbb{G}_{j,r,k}^{(v-1)} - \frac{\eta}{Nm} \sum_{i \in [N]} \ell_{k,i}'^{(v-1)} \sigma'\left(\langle \widetilde{\mathbf{w}}_{j,r,k}^{(v-1)}, y_{k,i}\boldsymbol{\mu}\rangle\right) \|\boldsymbol{\mu}\|_2^2, & \text{if } v \pmod{\tau} \neq 0, \\ \mathbb{G}_{j,r,k}^{(v-\tau)} - \frac{\eta}{nm} \sum_{s=0}^{\tau-1} \sum_{k'} \sum_{i \in [N]} \ell_{k',i}'^{(v-\tau+s)} \sigma'\left(\langle \widetilde{\mathbf{w}}_{j,r,k}^{(v-\tau+s)}, y_{k,i}\boldsymbol{\mu}\rangle\right) \|\boldsymbol{\mu}\|_2^2 & \text{else,} \end{cases} \tag{24}$$

where we slightly abuse notation, using $\ell_{k,i}'^{(v)}$ to denote $\ell_{k,i}'^{\left(\lfloor \frac{v}{\tau} \rfloor, v \bmod \tau\right)}$.

$$\overline{\mathbb{P}}_{j,r,k,i}^{(v)} = \begin{cases} \overline{\mathbb{P}}_{j,r,k,i}^{(v-1)} - \frac{\eta}{Nm} \ell_{k,i}'^{(v-1)} \sigma'\left(\langle \widetilde{\mathbf{w}}_{j,r,k}^{(v-1)}, \boldsymbol{\xi}_{k,i}\rangle\right) \|\boldsymbol{\xi}_{k,i}\|_2^2 \mathbb{1}\left(j = y_{k,i}\right), & \text{if } v \pmod{\tau} \neq 0, \\ \overline{\mathbb{P}}_{j,r,k,i}^{(v-\tau)} - \frac{\eta}{nm} \sum_{s=0}^{\tau-1} \ell_{k,i}'^{(v-\tau+s)} \sigma'\left(\langle \widetilde{\mathbf{w}}_{j,r,k}^{(v-\tau+s)}, \boldsymbol{\xi}_{k,i}\rangle\right) \|\boldsymbol{\xi}_{k,i}\|_2^2 \mathbb{1}\left(j = y_{k,i}\right) & \text{else.} \end{cases} \tag{25}$$

$$\underline{\mathbb{P}}_{j,r,k,i}^{(v)} = \begin{cases} \underline{\mathbb{P}}_{j,r,k,i}^{(v-1)} + \frac{\eta}{Nm} \ell_{k,i}'^{(v-1)} \sigma'\left(\langle \widetilde{\mathbf{w}}_{j,r,k}^{(v-1)}, \boldsymbol{\xi}_{k,i}\rangle\right) \|\boldsymbol{\xi}_{k,i}\|_2^2 \mathbb{1}\left(j = -y_{k,i}\right), & \text{if } v \pmod{\tau} \neq 0, \\ \underline{\mathbb{P}}_{j,r,k,i}^{(v-\tau)} + \frac{\eta}{nm} \sum_{s=0}^{\tau-1} \ell_{k,i}'^{(v-\tau+s)} \sigma'\left(\langle \widetilde{\mathbf{w}}_{j,r,k}^{(v-\tau+s)}, \boldsymbol{\xi}_{k,i}\rangle\right) \|\boldsymbol{\xi}_{k,i}\|_2^2 \mathbb{1}\left(j = -y_{k,i}\right) & \text{else.} \end{cases} \tag{26}$$

Note that we have the relation $\mathbb{G}_{j,r,k}^{(t\tau)} = \Gamma_{j,r}^{(t)}, \overline{\mathbb{P}}_{j,r,k,i}^{(t\tau)} = \overline{P}_{j,r,k,i}^{(t)}, \underline{\mathbb{P}}_{j,r,k,i}^{(t\tau)} = \underline{P}_{j,r,k,i}^{(t)}$

for all $t = 0, 1, 2, \ldots, T^* - 1$. Intuitively, if we can bound the virtual sequence of coefficients, we can also bound the actual coefficients of the global model at every round.

### C.2.1 Decomposition of Virtual Local Filter Weights

The purpose of introducing the virtual sequence of coefficients is to write the local filter weight at each client as the following decomposition.

$$\widetilde{\mathbf{w}}_{j,r,k}^{(v)} = \mathbf{w}_{j,r}^{(0)} + j\mathbb{G}_{j,r,k}^{(v)} \|\boldsymbol{\mu}\|_2^{-2} \boldsymbol{\mu} + \sum_{k',k' \neq k} \sum_{i' \in [N]} (\overline{\mathbb{P}}_{j,r,k',i'}^{(\tau\lfloor v/\tau \rfloor)} + \underline{\mathbb{P}}_{j,r,k',i'}^{(\tau\lfloor v/\tau \rfloor)}) \|\boldsymbol{\xi}_{k',i'}\|_2^{-2} \boldsymbol{\xi}_{k',i'}$$
$$+ \sum_{i \in [N]} (\overline{\mathbb{P}}_{j,r,k,i}^{(v)} + \underline{\mathbb{P}}_{j,r,k,i}^{(v)}) \|\boldsymbol{\xi}_{k,i}\|_2^{-2} \boldsymbol{\xi}_{k,i}. \tag{27}$$

Note that $(\tau\lfloor v/\tau \rfloor)$ denotes the last iteration at which communication happened. If $v \pmod{\tau} = 0$, then $\widetilde{\mathbf{w}}_{j,r,k}^{(v)}$ is the same for all $k \in [K]$.

### C.2.2  Theorem on Scale of Coefficients

We will now state the theorem that bounds our virtual sequence of coefficients and give the proof below. We first define some quantities that will be used throughout the proof.

$$\alpha := 4\log(T^*\tau); \ \beta := 2 \max_{i,j,k,r} \left\{ \left| \langle \mathbf{w}_{j,r}^{(0)}, \boldsymbol{\mu} \rangle \right|, \left| \langle \mathbf{w}_{j,r}^{(0)}, \boldsymbol{\xi}_{k,i} \rangle \right| \right\}; \ \widehat{\gamma} = \frac{n \|\boldsymbol{\mu}\|_2^2}{\sigma_p^2 d}.$$

**Theorem 3.** *Under assumptions, for all $v = 0, 1, 2, \ldots, T^*\tau - 1$, we have that,*

$$\mathbb{G}_{j,r,k}^{(0)} = 0, \overline{\mathbb{P}}_{j,r,k,i}^{(0)} = 0, \underline{\mathbb{P}}_{j,r,k,i}^{(0)} = 0,$$

$$0 \le \overline{\mathbb{P}}_{j,r,k,i}^{(v)} \le \alpha, \tag{28}$$

$$0 \ge \underline{\mathbb{P}}_{j,r,k,i}^{(v)} \ge -\beta - 8\sqrt{\frac{\log(6n^2/\delta)}{d}} n\alpha \ge -\alpha, \tag{29}$$

$$0 \le \mathbb{G}_{j,r,k}^{(v)} \le C'\widehat{\gamma}\alpha, \tag{30}$$

*for all $r \in [m], j \in \{\pm 1\}, k \in [K], i \in [N]$, where $C'$ is some positive constant.*

We will use induction to prove this theorem. The statement is clearly true at $v = 0$. Now assuming the statement holds at $v = v'$ we will show that it holds at $v = v' + 1$. We first state and prove some intermediate lemmas that we will use in our proof.

### C.2.3  Intermediate Steps to Prove the Induction in Theorem 3

**Lemma 10.**

$$\max \left\{ \beta, 4\sqrt{\frac{\log(6n^2/\delta)}{d}} n\alpha \right\} \le \frac{1}{12}.$$

*Proof.* From Lemma 6 we have $\beta = 4\sigma_0 \max \left\{ \sqrt{\log(12mn/\delta)} \cdot \sigma_p \sqrt{d}, \sqrt{\log(12m/\delta)} \cdot \|\boldsymbol{\mu}\|_2 \right\}$. Now from Assumptions 1 and 4, by choosing $C$ large enough, the inequality is satisfied. $\square$

**Lemma 11.** *Suppose, equation 28, equation 29 and equation 30 holds for all iterations $0 \le v \le v'$. Then for all $r \in [m], j \in \{\pm 1\}, k \in [K], i \in [N]$ we have,*

$$\langle \widetilde{\mathbf{w}}_{j,r,k}^{(v')} - \mathbf{w}_{j,r}^{(0)}, \boldsymbol{\mu} \rangle = j\mathbb{G}_{j,r,k}^{(v')}, \tag{31}$$

$$\left| \langle \widetilde{\mathbf{w}}_{j,r,k}^{(v')} - \mathbf{w}_{j,r}^{(0)}, \boldsymbol{\xi}_{k,i} \rangle - \overline{\mathbb{P}}_{j,r,k,i}^{(v')} \right| \le 4\sqrt{\frac{\log(6n^2/\delta)}{d}} n\alpha, j = y_{k,i}, \tag{32}$$

$$\left| \langle \widetilde{\mathbf{w}}_{j,r,k}^{(v')} - \mathbf{w}_{j,r}^{(0)}, \boldsymbol{\xi}_{k,i} \rangle - \underline{\mathbb{P}}_{j,r,k,i}^{(v')} \right| \le 4\sqrt{\frac{\log(6n^2/\delta)}{d}} n\alpha, j \ne y_{k,i}. \tag{33}$$

*Proof of equation 31.* It follows directly from equation 27 by using our assumption that $\langle \boldsymbol{\mu}, \boldsymbol{\xi}_{k,i} \rangle = 0$ for all $k \in [K], i \in [N]$. $\square$

*Proof of equation 32.* Note that

for $y_{k,i} = j$ we have $\underline{\mathbb{P}}_{j,r,k,i}^{(v')} = 0$. Now using equation 27 for $j = y_{k,i}$ we have,

$$\left| \langle \widetilde{\mathbf{w}}_{j,r,k}^{(v')} - \mathbf{w}_{j,r}^{(0)}, \boldsymbol{\xi}_{k,i} \rangle - \overline{\mathbb{P}}_{j,r,k,i}^{(v')} \right|$$

$$= \left| \sum_{k',k'\neq k} \sum_{i'\in[N]} (\overline{\mathbb{P}}_{j,r,k',i'}^{(\tau\lfloor v'/\tau\rfloor)} + \underline{\mathbb{P}}_{j,r,k',i'}^{(\tau\lfloor v'/\tau\rfloor)}) \frac{\langle \boldsymbol{\xi}_{k,i}, \boldsymbol{\xi}_{k',i'} \rangle}{\|\boldsymbol{\xi}_{k',i'}\|_2^2} + \sum_{i'\in[N],i'\neq i} (\overline{\mathbb{P}}_{j,r,k,i'}^{(v')} + \underline{\mathbb{P}}_{j,r,k,i'}^{(v')}) \frac{\langle \boldsymbol{\xi}_{k,i}, \boldsymbol{\xi}_{k,i'} \rangle}{\|\boldsymbol{\xi}_{k,i'}\|_2^2} \right|$$

$$\overset{(a)}{\leq} \left( \sum_{k',k'\neq k} \sum_{i'\in[N]} \left( \left| \overline{\mathbb{P}}_{j,r,k',i'}^{(\tau\lfloor v'/\tau\rfloor)} \right| + \left| \underline{\mathbb{P}}_{j,r,k',i'}^{(\tau\lfloor v'/\tau\rfloor)} \right| \right) + \sum_{i'\in[N]} \left( \left| \overline{\mathbb{P}}_{j,r,k,i'}^{(v')} \right| + \left| \underline{\mathbb{P}}_{j,r,k,i'}^{(v')} \right| \right) \right) 4\sqrt{\frac{\log(6n^2/\delta)}{d}}$$

$$\overset{(b)}{\leq} 4\sqrt{\frac{\log(6n^2/\delta)}{d}} n\alpha,$$

where $(a)$ follows from triangle inequality and Lemma 5; $(b)$ follows from the induction hypothesis. $\qquad\square$

*Proof of equation 33.* Note that for

$j \neq y_{k,i}$ we have $\overline{\mathbb{P}}_{j,r,k,i}^{(v')} = 0$. Using equation 27 for $j \neq y_{k,i}$ we have,

$$\left| \langle \widetilde{\mathbf{w}}_{j,r,k}^{(v')} - \mathbf{w}_{j,r}^{(0)}, \boldsymbol{\xi}_{k,i} \rangle - \underline{\mathbb{P}}_{j,r,k,i}^{(v')} \right|$$

$$= \left| \sum_{k',k'\neq k} \sum_{i'\in[N]} (\overline{\mathbb{P}}_{j,r,k',i'}^{(\tau\lfloor v'/\tau\rfloor)} + \underline{\mathbb{P}}_{j,r,k',i'}^{(\tau\lfloor v'/\tau\rfloor)}) \frac{\langle \boldsymbol{\xi}_{k,i}, \boldsymbol{\xi}_{k',i'} \rangle}{\|\boldsymbol{\xi}_{k',i'}\|_2^2} + \sum_{i'\in[N],i'\neq i} (\overline{\mathbb{P}}_{j,r,k,i'}^{(v')} + \underline{\mathbb{P}}_{j,r,k,i'}^{(v')}) \frac{\langle \boldsymbol{\xi}_{k,i}, \boldsymbol{\xi}_{k,i'} \rangle}{\|\boldsymbol{\xi}_{k,i'}\|_2^2} \right|$$

$$\overset{(a)}{\leq} \left( \sum_{k',k'\neq k} \sum_{i'\in[N]} \left( \left| \overline{\mathbb{P}}_{j,r,k',i'}^{(\tau\lfloor v'/\tau\rfloor)} \right| + \left| \underline{\mathbb{P}}_{j,r,k',i'}^{(\tau\lfloor v'/\tau\rfloor)} \right| \right) + \sum_{i'\in[N]} \left( \left| \overline{\mathbb{P}}_{j,r,k,i'}^{(v')} \right| + \left| \underline{\mathbb{P}}_{j,r,k,i'}^{(v')} \right| \right) \right) 4\sqrt{\frac{\log(6n^2/\delta)}{d}}$$

$$\overset{(b)}{\leq} 4\sqrt{\frac{\log(6n^2/\delta)}{d}} n\alpha,$$

where $(a)$ follows from triangle inequality and Lemma 5; $(b)$ follows from the induction hypothesis.

This concludes the proof of Lemma 10. $\qquad\square$

**Lemma 12.** *Suppose equation 28, equation 29 and equation 30 hold at iteration $v'$. Then for all $k \in [K]$ and $i \in [N]$,*

1. *For $j \neq y_{k,i}$, $F_j(\widetilde{\mathbf{W}}_{j,k}^{(v')}, \mathbf{x}_{k,i}) \leq 0.5$.*

2. *For $j = y_{k,i}$, $F_j(\widetilde{\mathbf{W}}_{j,k}^{(v')}, \mathbf{x}_{k,i}) \geq \frac{1}{m}\sum_{r=1}^m \overline{\mathbb{P}}_{j,r,k,i}^{(v')} - 0.25$.*

3. *$y_{k,i}f(\widetilde{\mathbf{W}}_k^{(v')}, \mathbf{x}_{k,i}) \geq \frac{1}{m}\sum_{r=1}^m \overline{\mathbb{P}}_{y_{k,i},r,k,i}^{(v')} - 0.75$.*

*Proof of 1.* First note that for $j \neq y_{k,i}$ from Lemma 11 we have,

$$\langle \widetilde{\mathbf{w}}_{j,r,k}^{(v')}, \boldsymbol{\mu} \rangle \leq \langle \mathbf{w}_{j,r}^{(0)}, \boldsymbol{\mu} \rangle. \tag{34}$$

since $\mathbb{G}_{j,r,k}^{(v')} \geq 0$ by the induction hypothesis. Also from Lemma 11 for $j \neq y_{k,i}$ we have,

$$\langle \widetilde{\mathbf{w}}_{j,r,k}^{(v')}, \boldsymbol{\xi}_{k,i} \rangle \leq \langle \mathbf{w}_{j,r}^{(0)}, \boldsymbol{\xi}_{k,i} \rangle + \underline{\mathbb{P}}_{j,r,k,i}^{(v')} + 4\sqrt{\frac{\log(6n^2/\delta)}{d}} n\alpha$$

$$\overset{(a)}{\leq} \langle \mathbf{w}_{j,r}^{(0)}, \boldsymbol{\xi}_{k,i} \rangle + 4\sqrt{\frac{\log(6n^2/\delta)}{d}} n\alpha \tag{35}$$

where $(a)$ follows from $\underline{\mathbb{P}}_{-j,r,k,i}^{(v')} \leq 0$ (induction hypothesis). Now using the definition of $F_j(\mathbf{W}, \mathbf{x})$ for $j \neq y_{k,i}$ we have,

$$
\begin{aligned}
F_j(\widetilde{\mathbf{W}}_{j,k}^{(v')}, \mathbf{x}_{k,i}) &= \frac{1}{m} \sum_{r=1}^{m} \left[ \sigma\left( \langle \widetilde{\mathbf{w}}_{j,r,k}^{(v')}, y_{k,i}\boldsymbol{\mu} \rangle \right) + \sigma\left( \langle \widetilde{\mathbf{w}}_{j,r,k}^{(v')}, \boldsymbol{\xi}_{k,i} \rangle \right) \right] \\
&\overset{(a)}{\leq} 3 \max_{r \in [m]} \left\{ \left| \langle \mathbf{w}_{j,r}^{(0)}, \boldsymbol{\mu} \rangle \right|, \left| \langle \mathbf{w}_{j,r}^{(0)}, \boldsymbol{\xi}_{k,i} \rangle \right|, 4\sqrt{\frac{\log(6n^2/\delta)}{d}} n\alpha \right\} \\
&\overset{(b)}{\leq} 3 \max \left\{ \beta, 4\sqrt{\frac{\log(6n^2/\delta)}{d}} n\alpha \right\} \\
&\overset{(c)}{\leq} 0.5.
\end{aligned}
\tag{36}
$$

Here $(a)$ follows from equation 34 and equation 35; $(b)$ follows from the definition of $\beta$; $(c)$ follows from Lemma 10.

$\square$

*Proof of 2.* For $j = y_{k,i}$ we have,

$$
\begin{aligned}
F_j(\widetilde{\mathbf{W}}_{j,k}^{(v')}, \mathbf{x}_{k,i}) &= \frac{1}{m} \sum_{r=1}^{m} \left[ \sigma\left( \langle \widetilde{\mathbf{w}}_{j,r,k}^{(v')}, y_{k,i}\boldsymbol{\mu} \rangle \right) + \sigma\left( \langle \widetilde{\mathbf{w}}_{j,r,k}^{(v')}, \boldsymbol{\xi}_{k,i} \rangle \right) \right] \\
&\overset{(a)}{\geq} \frac{1}{m} \sum_{r=1}^{m} \left[ \langle \widetilde{\mathbf{w}}_{j,r,k}^{(v')}, y_{k,i}\boldsymbol{\mu} \rangle + \langle \widetilde{\mathbf{w}}_{j,r,k}^{(v')}, \boldsymbol{\xi}_{k,i} \rangle \right] \\
&\overset{(b)}{\geq} \frac{1}{m} \sum_{r=1}^{m} \left[ \langle \mathbf{w}_{j,r}^{(0)}, y_{k,i}\boldsymbol{\mu} \rangle + \langle \mathbf{w}_{j,r}^{(0)}, \boldsymbol{\xi}_{k,i} \rangle + \overline{\mathbb{P}}_{j,r,k,i}^{(v')} - 4\sqrt{\frac{\log(6n^2/\delta)}{d}} n\alpha \right] \\
&\overset{(c)}{\geq} \frac{1}{m} \sum_{r=1}^{m} \overline{\mathbb{P}}_{j,r,k,i}^{(v')} - 2\beta - 4\sqrt{\frac{\log(6n^2/\delta)}{d}} n\alpha \\
&\overset{(d)}{\geq} \frac{1}{m} \sum_{r=1}^{m} \overline{\mathbb{P}}_{j,r,k,i}^{(v')} - 0.25.
\end{aligned}
\tag{37}
$$

Here $(a)$ follows from $\sigma(z) \geq z$; $(b)$ follows from Lemma 11 and that $\mathbb{G}_{j,r,k}^{(v')} \geq 0$; $(c)$ follows from the definition of $\beta$; $(d)$ follows from Lemma 10.

$\square$

*Proof of 3.* Combining the results in equation 36 and equation 37 we have,

$$
\begin{aligned}
y_{k,i} f(\widetilde{\mathbf{W}}_k^{(v')}, \mathbf{x}_{k,i}) &= F_{y_{k,i}}(\widetilde{\mathbf{W}}_{y_{k,i},k}^{(v')}, \mathbf{x}_{k,i}) - F_{-y_{k,i}}(\widetilde{\mathbf{W}}_{-y_{k,i},k}^{(v')}, \mathbf{x}_{k,i}) \\
&\overset{(a)}{\geq} F_{y_{k,i}}(\widetilde{\mathbf{W}}_{y_{k,i},k}^{(v')}, \mathbf{x}_{k,i}) - 0.5 \\
&\overset{(b)}{\geq} \frac{1}{m} \sum_{r=1}^{m} \overline{\mathbb{P}}_{y_{k,i},r,k,i}^{(v')} - 0.75.
\end{aligned}
$$

where $(a)$ follows from equation 36; $(b)$ follows from equation 37.

This concludes the proof of Lemma 12. $\square$

**Lemma 13.** *Suppose equation 28, equation 29 and equation 30 hold at iteration $v'$. Then for all $j \in \{\pm 1\}$, $k \in [K]$ and $i \in [N]$, $\left| \ell_{k,i}^{\prime(v')} \right| \leq \exp\left( -F_{y_{k,i}}(\widetilde{\mathbf{W}}_{y_{k,i},k}^{(v')}, \mathbf{x}_i) + 0.5 \right)$.*

*Proof.* We have,

$$
\left| \ell'^{(v')}_{k,i} \right| = \frac{1}{1 + \exp\left( y_{k,i} \left[ F_{+1}(\widetilde{\mathbf{W}}^{(v')}_{+1,k}, \mathbf{x}_{k,i}) - F_{-1}(\widetilde{\mathbf{W}}^{(v')}_{+1,k}, \mathbf{x}_{k,i}) \right] \right)}
$$

$$
\overset{(a)}{\leq} \exp\left( -y_{k,i} \left[ F_{+1}(\widetilde{\mathbf{W}}^{(v')}_{+1,k}, \mathbf{x}_{k,i}) - F_{-1}(\widetilde{\mathbf{W}}^{(v')}_{+1,k}, \mathbf{x}_{k,i}) \right] \right)
$$

$$
= \exp\left( -F_{y_{k,i}}(\widetilde{\mathbf{W}}^{(v')}_{y_{k,i},k}, \mathbf{x}_{k,i}) + F_{-y_{k,i}}(\widetilde{\mathbf{W}}^{(v')}_{-y_{k,i},k}, \mathbf{x}_{k,i}) \right)
$$

$$
\overset{(b)}{\leq} \exp\left( -F_{y_{k,i}}(\widetilde{\mathbf{W}}^{(v')}_{y_{k,i},k}, \mathbf{x}_{k,i}) + 0.5 \right),
$$

where $(a)$ uses $1/(1 + \exp(z)) \leq \exp(-z)$; $(b)$ uses part 1 of Lemma 12. $\qquad\square$

**Lemma 14.** *Let $g(z) = \ell'(z) = -1/(1 + \exp(z))$. Further suppose $z_2 - z_1 \leq c$ where $c \geq 0$. Then,*

$$
\frac{g(z_1)}{g(z_2)} \leq \exp(c). \tag{38}
$$

*Proof.* We have,

$$
\frac{g(z_1)}{g(z_2)} = \frac{1 + \exp(z_2)}{1 + \exp(z_1)} \leq \max\{1, \exp(z_2 - z_1)\} \overset{(a)}{\leq} \exp(c),
$$

where $(a)$ follows from $c \geq 0$. $\qquad\square$

**Lemma 15.** *Suppose equation 28, equation 29 and equation 30 hold at iteration $v'$. Then for all $k \in [K]$ and $i \in [N]$,*

$$
\langle \widetilde{\mathbf{w}}^{(v')}_{y_{k,i},r,k}, \boldsymbol{\xi}_{k,i} \rangle \geq -0.25, \tag{39}
$$

$$
\langle \widetilde{\mathbf{w}}^{(v')}_{y_{k,i},r,k}, \boldsymbol{\xi}_{k,i} \rangle \leq \sigma\left( \langle \widetilde{\mathbf{w}}^{(v')}_{y_{k,i},r,k}, \boldsymbol{\xi}_{k,i} \rangle \right) \leq \langle \widetilde{\mathbf{w}}^{(v')}_{y_{k,i},r,k}, \boldsymbol{\xi}_{k,i} \rangle + 0.25. \tag{40}
$$

*Proof of equation 39.* From Lemma 11 we have,

$$
\langle \widetilde{\mathbf{w}}^{(v')}_{y_{k,i},r,k}, \boldsymbol{\xi}_{k,i} \rangle \geq \langle \mathbf{w}^{(0)}_{y_{k,i},r,k}, \boldsymbol{\xi}_{k,i} \rangle + \overline{\mathbb{P}}^{(v')}_{y_{k,i},r,k,i} - 4\sqrt{\frac{\log(6n^2/\delta)}{d}} n\alpha
$$

$$
\overset{(a)}{\geq} -\beta - 4\sqrt{\frac{\log(6n^2/\delta)}{d}} n\alpha
$$

$$
\overset{(b)}{\geq} -0.25.
$$

Here $(a)$ follows from the definition of $\beta$ and $\overline{\mathbb{P}}^{(v')}_{y_{k,i},r,k,i} \geq 0$ for all $v' \geq 0$; $(b)$ follows from Lemma 10. $\qquad\square$

*Proof of equation 40.* The first inequality of equation 40 follows naturally since $\sigma(z) \geq z$ for all $z \in \mathbb{R}$. For the second inequality we have,

$$
\sigma\left( \langle \widetilde{\mathbf{w}}^{(v')}_{y_{k,i},r,k}, \boldsymbol{\xi}_{k,i} \rangle \right) = \begin{cases} \langle \widetilde{\mathbf{w}}^{(v')}_{y_{k,i},r,k}, \boldsymbol{\xi}_{k,i} \rangle \leq \langle \widetilde{\mathbf{w}}^{(v')}_{y_{k,i},r,k}, \boldsymbol{\xi}_{k,i} \rangle + 0.25, & \text{if } \langle \widetilde{\mathbf{w}}^{(v')}_{y_{k,i},r,k}, \boldsymbol{\xi}_{k,i} \rangle \geq 0 \\ 0 \overset{(a)}{\leq} \langle \widetilde{\mathbf{w}}^{(v')}_{y_{k,i},r,k}, \boldsymbol{\xi}_{k,i} \rangle + 0.25, & \text{if } \langle \widetilde{\mathbf{w}}^{(v')}_{y_{k,i},r,k}, \boldsymbol{\xi}_{k,i} \rangle < 0, \end{cases}
$$

where $(a)$ follows from $\langle \widetilde{\mathbf{w}}^{(v')}_{y_{k,i},r,k}, \boldsymbol{\xi}_{k,i} \rangle \geq -0.25$. This completes the proof.

This concludes the proof of Lemma 15. $\qquad\square$

**Lemma 16.** *Suppose equation 28, equation 29 and equation 30 hold at iteration $v'$. Then for all $k, k' \in [K]$ and $i, i' \in [N]$,*

$$\left| y_{k,i} f(\widetilde{\mathbf{W}}_k^{(v')}, \mathbf{x}_{k,i}) - y_{k',i'} f(\widetilde{\mathbf{W}}_{k'}^{(v')}, \mathbf{x}_{k',i'}) - \frac{1}{m} \sum_{r=1}^m \left[ \overline{\mathbb{P}}_{y_{k,i},r,k,i}^{(v')} - \overline{\mathbb{P}}_{y_{k',i'},r,k',i'}^{(v')} \right] \right| \leq 1.75.$$

*Proof.* We can write,

$$\begin{aligned}
&y_{k,i} f(\widetilde{\mathbf{W}}_k^{(v')}, \mathbf{x}_{k,i}) - y_{k',i'} f(\widetilde{\mathbf{W}}_{k'}^{(v')}, \mathbf{x}_{k',i'}) \\
&= F_{y_{k,i}}(\widetilde{\mathbf{W}}_{y_{k,i},k}^{(v')}, \mathbf{x}_{k,i}) - F_{-y_{k,i}}(\widetilde{\mathbf{W}}_{-y_{k,i},k}^{(v')}, \mathbf{x}_{k,i}) \\
&\quad - F_{y_{k',i'}}(\widetilde{\mathbf{W}}_{y_{k',i'},k'}^{(v')}, \mathbf{x}_{k',i'}) + F_{-y_{k',i'}}(\widetilde{\mathbf{W}}_{-y_{k',i'},k'}^{(v')}, \mathbf{x}_{k',i'}) \\
&= F_{-y_{k',i'}}(\widetilde{\mathbf{W}}_{-y_{k',i'},k'}^{(v')}, \mathbf{x}_{k',i'}) - F_{-y_{k,i}}(\widetilde{\mathbf{W}}_{-y_{k,i},k}^{(v')}, \mathbf{x}_{k,i}) \\
&\quad + F_{y_{k,i}}(\widetilde{\mathbf{W}}_{y_{k,i},k}^{(v')}, \mathbf{x}_{k,i}) - F_{y_{k',i'}}(\widetilde{\mathbf{W}}_{y_{k',i'},k'}^{(v')}, \mathbf{x}_{k',i'}) \\
&= \underbrace{F_{-y_{k',i'}}(\widetilde{\mathbf{W}}_{-y_{k',i'},k'}^{(v')}, \mathbf{x}_{k',i'}) - F_{-y_{k,i}}(\widetilde{\mathbf{W}}_{-y_{k,i},k}^{(v')}, \mathbf{x}_{k,i})}_{I_1} \\
&\quad + \underbrace{\frac{1}{m} \sum_{r=1}^m \left[ \sigma\left( \langle \widetilde{\mathbf{w}}_{y_{k,i},r,k}^{(v')}, y_{k,i}\boldsymbol{\mu} \rangle \right) - \sigma\left( \langle \widetilde{\mathbf{w}}_{y_{k',i'},r,k'}^{(v')}, y_{k',i'}\boldsymbol{\mu} \rangle \right) \right]}_{I_2} \\
&\quad + \underbrace{\frac{1}{m} \sum_{r=1}^m \left[ \sigma\left( \langle \widetilde{\mathbf{w}}_{y_{k,i},r,k}^{(v')}, \boldsymbol{\xi}_{k,i} \rangle \right) - \sigma\left( \langle \widetilde{\mathbf{w}}_{y_{k',i'},r,k'}^{(v')}, \boldsymbol{\xi}_{k',i'} \rangle \right) \right]}_{I_3}.
\end{aligned}$$

Next we bound $I_1, I_2$ and $I_3$ as follows.

$$|I_1| \leq F_{-y_{k',i'}}(\widetilde{\mathbf{W}}_{-y_{k',i'},k'}^{(v')}, \mathbf{x}_{k',i'}) + F_{-y_{k,i}}(\widetilde{\mathbf{W}}_{-y_{k,i},k}^{(v')}, \mathbf{x}_{k,i}) \overset{(a)}{\leq} 1,$$

where $(a)$ follows from part 1 of Lemma 12. For $|I_2|$ we have the following bound,

$$\begin{aligned}
|I_2| &\leq \max\left\{ \frac{1}{m} \sum_{r=1}^m \sigma\left( \langle \widetilde{\mathbf{w}}_{y_{k,i},r,k}^{(v')}, y_{k,i}\boldsymbol{\mu} \rangle \right), \frac{1}{m} \sum_{r=1}^m \sigma\left( \langle \widetilde{\mathbf{w}}_{y_{k',i'},r,k'}^{(v')}, y_{k',i'}\boldsymbol{\mu} \rangle \right) \right\} \\
&\overset{(a)}{\leq} 2 \max_{r \in [m]} \left\{ \left| \langle \mathbf{w}_{y_{k,i},r}^{(0)}, \boldsymbol{\mu} \rangle \right|, \left| \langle \mathbf{w}_{y_{k',i'},r}^{(0)}, \boldsymbol{\mu} \rangle \right|, \mathbb{G}_{y_{k,i},r,k}^{(v')}, \mathbb{G}_{y_{k',i'},r,k'}^{(v')} \right\} \\
&\overset{(b)}{\leq} 2 \max_{r \in [m]} \left\{ \beta, C'\hat{\gamma}\alpha \right\} \\
&\overset{(c)}{\leq} 0.25.
\end{aligned}$$

Here $(a)$ follows Lemma 11, $(b)$ follows from the definition of $\beta$ and the induction hypothesis, $(c)$ follows from Lemma 10 and Assumption 1.

Next we derive an upper bound on $I_3$ as follows.

$$I_3 = \frac{1}{m} \sum_{r=1}^{m} \left[ \sigma\left( \langle \widetilde{\mathbf{w}}_{y_{k,i},r,k}^{(v')}, \boldsymbol{\xi}_{k,i} \rangle \right) - \sigma\left( \langle \widetilde{\mathbf{w}}_{y_{k',i'},r,k'}^{(v')}, \boldsymbol{\xi}_{k',i'} \rangle \right) \right]$$

$$\overset{(a)}{\leq} \frac{1}{m} \sum_{r=1}^{m} \left[ \langle \widetilde{\mathbf{w}}_{y_{k,i},r,k}^{(v')}, \boldsymbol{\xi}_{k,i} \rangle - \langle \widetilde{\mathbf{w}}_{y_{k',i'},r,k'}^{(v')}, \boldsymbol{\xi}_{k',i'} \rangle \right] + 0.25$$

$$\overset{(b)}{\leq} \frac{1}{m} \sum_{r=1}^{m} \left[ \overline{\mathbb{P}}_{y_{k,i},r,k,i}^{(v')} - \overline{\mathbb{P}}_{y_{k',i'},r,k',i'}^{(v')} \right] + 2\beta + 8\sqrt{\frac{\log(6n^2/\delta)}{d}} n\alpha + 0.25$$

$$\overset{(c)}{\leq} \frac{1}{m} \sum_{r=1}^{m} \left[ \overline{\mathbb{P}}_{y_{k,i},r,k,i}^{(v')} - \overline{\mathbb{P}}_{y_{k',i'},r,k',i'}^{(v')} \right] + 0.5.$$

Here $(a)$ follows from Lemma 15; $(b)$ follows from Lemma 11; $(c)$ follows from Lemma 10.

Similarly, we can get a lower bound for $I_3$ as follows,

$$I_3 = \frac{1}{m} \sum_{r=1}^{m} \left[ \sigma\left( \langle \widetilde{\mathbf{w}}_{y_{k,i},r,k}^{(v')}, \boldsymbol{\xi}_{k,i} \rangle \right) - \sigma\left( \langle \widetilde{\mathbf{w}}_{y_{k',i'},r,k'}^{(v')}, \boldsymbol{\xi}_{k',i'} \rangle \right) \right]$$

$$\overset{(a)}{\geq} \frac{1}{m} \sum_{r=1}^{m} \left[ \langle \widetilde{\mathbf{w}}_{y_{k,i},r,k}^{(v')}, \boldsymbol{\xi}_{k,i} \rangle - \langle \widetilde{\mathbf{w}}_{y_{k',i'},r,k'}^{(v')}, \boldsymbol{\xi}_{k',i'} \rangle \right] - 0.25$$

$$\overset{(b)}{\geq} \frac{1}{m} \sum_{r=1}^{m} \left[ \overline{\mathbb{P}}_{y_{k,i},r,k,i}^{(v')} - \overline{\mathbb{P}}_{y_{k',i'},r,k',i'}^{(v')} \right] - 2\beta - 8\sqrt{\frac{\log(6n^2/\delta)}{d}} n\alpha - 0.25$$

$$\overset{(c)}{\geq} \frac{1}{m} \sum_{r=1}^{m} \left[ \overline{\mathbb{P}}_{y_{k,i},r,k,i}^{(v')} - \overline{\mathbb{P}}_{y_{k',i'},r,k',i'}^{(v')} \right] - 0.5.$$

Here $(a)$ follows from Lemma 15; $(b)$ follows from Lemma 11; $(c)$ follows from Lemma 10.

Combining the above results, we have

$$y_{k,i} f(\widetilde{\mathbf{W}}_k^{(v')}, \mathbf{x}_{k,i}) - y_{k',i'} f(\widetilde{\mathbf{W}}_{k'}^{(v')}, \mathbf{x}_{k',i'}) \leq |I_1| + |I_2| + I_3$$

$$\leq \frac{1}{m} \sum_{r=1}^{m} \left[ \overline{\mathbb{P}}_{y_{k,i},r,k,i}^{(v')} - \overline{\mathbb{P}}_{y_{k',i'},r,k',i'}^{(v')} \right] + 1.75,$$

and,

$$y_{k,i} f(\widetilde{\mathbf{W}}_k^{(v')}, \mathbf{x}_{k,i}) - y_{k',i'} f(\widetilde{\mathbf{W}}_{k'}^{(v')}, \mathbf{x}_{k',i'}) \geq -|I_1| - |I_2| + I_3$$

$$\geq \frac{1}{m} \sum_{r=1}^{m} \left[ \overline{\mathbb{P}}_{y_{k,i},r,k,i}^{(v')} - \overline{\mathbb{P}}_{y_{k',i'},r,k',i'}^{(v')} \right] - 1.75.$$

This implies,

$$\left| y_{k,i} f(\widetilde{\mathbf{W}}_k^{(v')}, \mathbf{x}_{k,i}) - y_{k',i'} f(\widetilde{\mathbf{W}}_{k'}^{(v')}, \mathbf{x}_{k',i'}) - \frac{1}{m} \sum_{r=1}^{m} \left[ \overline{\mathbb{P}}_{y_{k,i},r,k,i}^{(v')} - \overline{\mathbb{P}}_{y_{k',i'},r,k',i'}^{(v')} \right] \right| \leq 1.75.$$

$\square$

We will now state and prove a version of Lemma C.7 that appears in Cao et al. (2022). Note that Cao et al. (2022) only considers the heterogeneity arising due to different datapoints for the same model. Interestingly, we show that the lemma can be extended to the case with different local models and different datapoints as long as the local models start from the same initialization.

**Lemma 17.** *Suppose equation 28, equation 29 and equation 30 hold for all $0 \le v \le v'$. Then the following holds for all $0 \le v \le v'$.*

1. $\frac{1}{m} \sum_{r=1}^{m} \left[ \overline{\mathbb{P}}^{(v)}_{y_{k,i},r,k,i} - \overline{\mathbb{P}}^{(v)}_{y_{k',i'},r,k',i'} \right] \le \kappa$ *for all* $k, k' \in [K], i, i' \in [N]$.

2. $y_{k,i} f(\widetilde{\mathbf{W}}^{(v)}_{k}, \mathbf{x}_{k,i}) - y_{k',i'} f(\widetilde{\mathbf{W}}^{(v)}_{k'}, \mathbf{x}_{k',i'}) \le C_1$ *for all* $k, k' \in [K]$ *and* $i, i' \in [N]$.

3. $\frac{\ell'^{(v)}_{k',i'}}{\ell'^{(v)}_{k,i}} \le C_2 = \exp(C_1)$ *for all* $k, k' \in [K]$ *and* $i, i' \in [N]$.

4. $S^{(0)}_{k,i} \subseteq S^{(v)}_{k,i}$ *where* $S^{(v)}_{k,i} := \left\{ r \in [m] : \langle \widetilde{\mathbf{w}}^{(v)}_{y_{k,i},r,k}, \boldsymbol{\xi}_{k,i} \rangle \ge 0 \right\}$, *and hence* $\left| S^{(v)}_{k,i} \right| \ge 0.4m$ *for all* $k \in [K], i \in [N]$.

5. $\tilde{S}^{(0)}_{j,r} \subseteq \tilde{S}^{(v)}_{j,r}$ *where* $\tilde{S}^{(0)}_{j,r} := \left\{ k \in [K], i \in [N] : y_{k,i} = j, \langle \widetilde{\mathbf{w}}^{(v)}_{j,r,k}, \boldsymbol{\xi}_{k,i} \rangle \ge 0 \right\}$, *and hence* $\left| \tilde{S}^{(v)}_{j,r} \right| \ge \frac{n}{8}$.

*Here we take $\kappa = 5$ and $C_1 = 6.75$.*

*Proof of 1.* We will use a proof by induction. For $v = 0$, it is simple to verify that 1 holds since $\overline{\mathbb{P}}^{(0)}_{j,r,k,i} = 0$ for all $j \in \{\pm 1\}, r \in [m], k \in [K], i \in [N]$ by definition. Now suppose 1 holds for all $0 \le v \le \tilde{v} < v'$. Then we will show that 1 also holds at $v = \tilde{v} + 1$. We have the following cases.

**Case 1:** $(\tilde{v} + 1) \pmod \tau \ne 0$

In this case, from equation 25

$$\overline{\mathbb{P}}^{(\tilde{v}+1)}_{y_{k,i},r,k,i} = \overline{\mathbb{P}}^{(\tilde{v})}_{y_{k,i},r,k,i} - \frac{\eta}{Nm} \ell'^{(\tilde{v})}_{k,i} \sigma' \left( \langle \widetilde{\mathbf{w}}^{(\tilde{v})}_{y_{k,i},r,k,i}, \boldsymbol{\xi}_{k,i} \rangle \right) \| \boldsymbol{\xi}_{k,i} \|_2^2 .$$

Thus,

$$
\begin{aligned}
\frac{1}{m} \sum_{r=1}^{m} \left[ \overline{\mathbb{P}}^{(\tilde{v}+1)}_{y_{k,i},r,k,i} - \overline{\mathbb{P}}^{(\tilde{v}+1)}_{y_{k',i'},r,k',i'} \right] &= \frac{1}{m} \sum_{r=1}^{m} \left[ \overline{\mathbb{P}}^{(\tilde{v})}_{y_{k,i},r,k,i} - \overline{\mathbb{P}}^{(\tilde{v})}_{y_{k',i'},r,k',i'} \right] \\
&\quad + \frac{\eta}{Nm^2} \left[ \left| S^{(\tilde{v})}_{k,i} \right| (-\ell'^{(\tilde{v})}_{k,i}) \| \boldsymbol{\xi}_{k,i} \|_2^2 - \left| S^{(\tilde{v})}_{k',i'} \right| (-\ell'^{(\tilde{v})}_{k',i'}) \| \boldsymbol{\xi}_{k',i'} \|_2^2 \right], \quad (41)
\end{aligned}
$$

where $S^{(\tilde{v})}_{k,i}, S^{(\tilde{v})}_{k',i'}$ are defined in 4.

We bound equation 41 in two cases, depending on the value of $\frac{1}{m} \sum_{r=1}^{m} \left[ \overline{\mathbb{P}}^{(\tilde{v})}_{y_{k,i},r,k,i} - \overline{\mathbb{P}}^{(\tilde{v})}_{y_{k',i'},r,k',i'} \right]$.

    **i)** If $\frac{1}{m} \sum_{r=1}^{m} \left[ \overline{\mathbb{P}}^{(\tilde{v})}_{y_{k,i},r,k,i} - \overline{\mathbb{P}}^{(\tilde{v})}_{y_{k',i'},r,k',i'} \right] \le 0.9\kappa$. From equation 41 we have,

$$
\begin{aligned}
\frac{1}{m} \sum_{r=1}^{m} \left[ \overline{\mathbb{P}}^{(\tilde{v}+1)}_{y_{k,i},r,k,i} - \overline{\mathbb{P}}^{(\tilde{v}+1)}_{y_{k',i'},r,k',i'} \right] &\le 0.9\kappa + \frac{\eta}{Nm^2} \left| S^{(\tilde{v})}_{k,i} \right| (-\ell'^{(\tilde{v})}_{k,i}) \| \boldsymbol{\xi}_{k,i} \|_2^2 \\
&\overset{(a)}{\le} 0.9\kappa + \frac{\eta}{Nm} \| \boldsymbol{\xi}_{k,i} \|_2^2 \\
&\overset{(b)}{\le} \kappa.
\end{aligned}
$$

$(a)$ follows from $\left| S^{(\tilde{v})}_{k,i} \right| \le m, -\ell'(\cdot) \le 1$; $(b)$ follows from Lemma 5 and Assumption 5.

**ii)** If $\frac{1}{m}\sum_{r=1}^{m}\left[\overline{\mathbb{P}}_{y_{k,i},r,k,i}^{(\tilde{v})} - \overline{\mathbb{P}}_{y_{k',i'},r,k',i'}^{(\tilde{v})}\right] > 0.9\kappa$. From Lemma 16 we know that,

$$
\begin{aligned}
y_{k,i}f(\widetilde{\mathbf{W}}_k^{(\tilde{v})},\mathbf{x}_{k,i}) - y_{k',i'}f(\widetilde{\mathbf{W}}_{k'}^{(\tilde{v})},\mathbf{x}_{k',i'}) &\geq \frac{1}{m}\sum_{r=1}^{m}\left[\overline{\mathbb{P}}_{y_{k,i},r,k,i}^{(\tilde{v})} - \overline{\mathbb{P}}_{y_{k',i'},r,k',i'}^{(\tilde{v})}\right] - 1.75 \\
&\stackrel{(a)}{\geq} 0.9\kappa - 0.35\kappa \\
&= 0.55\kappa.
\end{aligned}
\tag{42}
$$

where $(a)$ follows from $\kappa = 5$. Also note that since $\frac{1}{m}\sum_{r=1}^{m}\overline{\mathbb{P}}_{y_{k,i},r,k,i}^{(\tilde{v})} \geq \frac{1}{m}\sum_{r=1}^{m}\overline{\mathbb{P}}_{y_{k',i'},r,k',i'}^{(\tilde{v})} + 0.9\kappa \geq 0.9\kappa = 4.5$, we have from Lemma 12 that

$$
y_{k,i}f(\widetilde{\mathbf{W}}_k^{(\tilde{v})},\mathbf{x}_{k,i}) \geq 3.75.
\tag{43}
$$

Now from the definition of $\ell(\cdot)$ we have,

$$
\begin{aligned}
\frac{(-\ell'_{k,i}^{(\tilde{v})})}{(-\ell_{k',i'}^{(\tilde{v})})} &= \frac{1 + \exp(y_{k',i'}f(\widetilde{\mathbf{W}}_{k'}^{(\tilde{v})},\mathbf{x}_{k',i'}))}{1 + \exp(y_{k,i}f(\widetilde{\mathbf{W}}_k^{(\tilde{v})},\mathbf{x}_{k,i}))} \\
&\stackrel{(a)}{\leq} \frac{1 + \exp(y_{k,i}f(\widetilde{\mathbf{W}}_k^{(\tilde{v})},\mathbf{x}_{k,i}) - 0.55\kappa)}{1 + \exp(y_{k,i}f(\widetilde{\mathbf{W}}_k^{(\tilde{v})},\mathbf{x}_{k,i}))} \\
&\stackrel{(b)}{<} 1/7.5.
\end{aligned}
\tag{44}
$$

Here $(a)$ follows from equation 42; $(b)$ follows from equation 43.

Thus,

$$
\frac{\left|S_{k,i}^{(\tilde{v})}\right|\|\boldsymbol{\xi}_{k,i}\|_2^2\,(-\ell'_{k,i}^{(\tilde{v})})}{\left|S_{k',i'}^{(\tilde{v})}\right|\|\boldsymbol{\xi}_{k',i'}\|_2^2\,(-\ell_{k',i'}^{(\tilde{v})})} \stackrel{(a)}{\leq} 2.5\frac{\|\boldsymbol{\xi}_{k,i}\|_2^2\,(-\ell'_{k,i}^{(\tilde{v})})}{\|\boldsymbol{\xi}_{k',i'}\|_2^2\,(-\ell_{k',i'}^{(\tilde{v})})} \stackrel{(b)}{\leq} 2.5\cdot 3\frac{(-\ell'_{k,i}^{(\tilde{v})})}{(-\ell_{k',i'}^{(\tilde{v})})} \stackrel{(c)}{<} 1.
$$

Here $(a)$ follows from $\left|S_{k,i}^{(\tilde{v})}\right| \leq m, \left|S_{k',i'}^{(\tilde{v})}\right| \geq 0.4m$ using our induction hypothesis; $(b)$ follows from Lemma 5; $(c)$ follows from equation 44. This implies $\left|S_{k,i}^{(\tilde{v})}\right|\|\boldsymbol{\xi}_{k,i}\|_2^2\,(-\ell'_{k,i}^{(\tilde{v})}) < \left|S_{k',i'}^{(\tilde{v})}\right|\|\boldsymbol{\xi}_{k',i'}\|_2^2\,(-\ell_{k',i'}^{(\tilde{v})})$. Now from equation 41 we have,

$$
\frac{1}{m}\sum_{r=1}^{m}\left[\overline{\mathbb{P}}_{y_{k,i},r,k,i}^{(\tilde{v}+1)} - \overline{\mathbb{P}}_{y_{k',i'},r,k',i'}^{(\tilde{v}+1)}\right] \leq \frac{1}{m}\sum_{r=1}^{m}\left[\overline{\mathbb{P}}_{y_{k,i},r,k,i}^{(\tilde{v})} - \overline{\mathbb{P}}_{y_{k',i'},r,k',i'}^{(\tilde{v})}\right] \leq \kappa,
$$

where the last inequality follows from our induction hypothesis.

**Case 2:** $(\tilde{v}+1)\pmod\tau = 0$

In this case, using equation 25 we can write our update equation as follows:

$$
\begin{aligned}
&\frac{1}{m}\sum_{r=1}^{m}\left[\overline{\mathbb{P}}_{y_{k,i},r,k,i}^{(\tilde{v}+1)} - \overline{\mathbb{P}}_{y_{k',i'},r,k',i'}^{(\tilde{v}+1)}\right] \\
&= \frac{1}{m}\sum_{r=1}^{m}\left[\overline{\mathbb{P}}_{y_{k,i},r,k,i}^{(\tilde{v}+1-\tau)} - \overline{\mathbb{P}}_{y_{k',i'},r,k',i'}^{(\tilde{v}+1-\tau)}\right] \\
&\quad + \underbrace{\frac{1}{n}\frac{\eta}{m^2}\sum_{s=0}^{\tau-1}\left(\left|S_{k,i}^{(\tilde{v}+1-\tau+s)}\right|(-\ell'_{k,i}^{(\tilde{v}+1-\tau+s)})\|\boldsymbol{\xi}_{k,i}\|_2^2 - \left|S_{k',i'}^{(\tilde{v}+1-\tau+s)}\right|(-\ell'_{k',i'}^{(\tilde{v}+1-\tau+s)})\|\boldsymbol{\xi}_{k',i'}\|_2^2\right)}_{:=I_1} \\
&= \frac{1}{m}\sum_{r=1}^{m}\left[\overline{\mathbb{P}}_{y_{k,i},r,k,i}^{(\tilde{v}+1-\tau)} - \overline{\mathbb{P}}_{y_{k',i'},r,k',i'}^{(\tilde{v}+1-\tau)}\right] + \frac{I_1}{n}.
\end{aligned}
\tag{45}
$$

From our induction hypothesis

we know that

$$\frac{1}{m} \sum_{r=1}^{m} \left[ \overline{\mathbb{P}}_{y_{k,i},r,k,i}^{(\tilde{v})} - \overline{\mathbb{P}}_{y_{k',i'},r,k',i'}^{(\tilde{v})} \right] \le \kappa. \tag{46}$$

Now unrolling the LHS expression in equation 46 using equation 25, we see that this implies

$$\frac{1}{m} \sum_{r=1}^{m} \left[ \overline{\mathbb{P}}_{y_{k,i},r,k,i}^{(\tilde{v}+1-\tau)} - \overline{\mathbb{P}}_{y_{k',i'},r,k',i'}^{(\tilde{v}+1-\tau)} \right] + \frac{I_1}{N} \le \kappa \tag{47}$$

**Case 2a):** $I_1 \ge 0$.

In this case it directly follows equation 45 and equation 47 that $\frac{1}{m} \sum_{r=1}^{m} \left[ \overline{\mathbb{P}}_{y_{k,i},r,k,i}^{(\tilde{v}+1)} - \overline{\mathbb{P}}_{y_{k',i'},r,k',i'}^{(\tilde{v}+1)} \right] \le \kappa$ since $N \le n$.

**Case 2b):** If $I_1 < 0$.

In this case from equation 45 we have,

$$\frac{1}{m} \sum_{r=1}^{m} \left[ \overline{\mathbb{P}}_{y_{k,i},r,k,i}^{(\tilde{v}+1)} - \overline{\mathbb{P}}_{y_{k',i'},r,k',i'}^{(\tilde{v}+1)} \right] \le \frac{1}{m} \sum_{r=1}^{m} \left[ \overline{\mathbb{P}}_{y_{k,i},r,k,i}^{(\tilde{v}+1-\tau)} - \overline{\mathbb{P}}_{y_{k',i'},r,k',i'}^{(\tilde{v}+1-\tau)} \right] \le \kappa.$$

where the last inequality follows from our induction hypothesis. $\qquad\square$

*Proof of 2.* For any $0 \le v \le v'$ we have,

$$y_{k,i} f(\widetilde{\mathbf{W}}_k^{(v)}, \mathbf{x}_{k,i}) - y_{k',i'} f(\widetilde{\mathbf{W}}_{k'}^{(v)}, \mathbf{x}_{k',i'}) \overset{(a)}{\le} \frac{1}{m} \sum_{r=1}^{m} \left[ \overline{\mathbb{P}}_{y_{k,i},r,k,i}^{(v)} - \overline{\mathbb{P}}_{y_{k',i'},r,k',i'}^{(v)} \right] + 1.75$$

$$\overset{(b)}{\le} \kappa + 1.75 = C_1.$$

Here $(a)$ follows from Lemma 16; $(b)$ follows from 1. $\qquad\square$

*Proof of 3.* For any $0 \le v \le v'$ we have,

$$\frac{\ell_{k',i'}^{'(v)}}{\ell_{k,i}^{'(v)}} \overset{(a)}{\le} \max \left\{ 1, \exp \left( y_{k,i} f(\widetilde{\mathbf{W}}_k^{(v)}, \mathbf{x}_{k,i}) - y_{k',i'} f(\widetilde{\mathbf{W}}_{k'}^{(v)}, \mathbf{x}_{k',i'}) \right) \right\} \overset{(b)}{\le} \exp(C_1).$$

Here $(a)$ follows from Lemma 14; $(b)$ follows from 2. $\qquad\square$

*Proof of 4.* To prove 4, we will use the result in 3 and show that $\langle \widetilde{\mathbf{w}}_{y_{k,i},r,k}^{(0)}, \boldsymbol{\xi}_{k,i} \rangle > 0$ implies $\langle \widetilde{\mathbf{w}}_{y_{k,i},r,k}^{(v)}, \boldsymbol{\xi}_{k,i} \rangle > 0$ for all $1 \le v \le v'$. We use a proof by induction. Assuming $\langle \widetilde{\mathbf{w}}_{y_{k,i},r,k}^{(v)}, \boldsymbol{\xi}_{k,i} \rangle > 0$ for all $0 \le v \le \tilde{v} < v'$, we will show that $\langle \widetilde{\mathbf{w}}_{y_{k,i},r,k}^{(\tilde{v}+1)}, \boldsymbol{\xi}_{k,i} \rangle > 0$. We have the following cases.

**Case 1:** $(\tilde{v}+1) \pmod{\tau} \ne 0$.

Using the fact that $\langle \widetilde{\mathbf{w}}_{y_{k,i},r,k}^{(\tilde{v})}, \boldsymbol{\xi}_{k,i} \rangle > 0$ we have,

$$\langle \widetilde{\mathbf{w}}_{y_{k,i},r,k}^{(\tilde{v}+1)}, \boldsymbol{\xi}_{k,i} \rangle = \langle \widetilde{\mathbf{w}}_{y_{k,i},r,k}^{(\tilde{v})}, \boldsymbol{\xi}_{k,i} \rangle + \frac{\eta}{Nm}(-\ell'^{(\tilde{v})}_{k,i}) \|\boldsymbol{\xi}_{k,i}\|_2^2$$

$$+ \frac{\eta}{Nm} \sum_{i' \in [N], i' \neq i} (-\ell'^{(\tilde{v})}_{k,i'}) \sigma' \left( \langle \widetilde{\mathbf{w}}_{y_{k,i},r,k}^{(\tilde{v})}, \boldsymbol{\xi}_{k,i'} \rangle \right) \langle \boldsymbol{\xi}_{k,i}, \boldsymbol{\xi}_{k,i'} \rangle$$

$$\overset{(a)}{\geq} \langle \widetilde{\mathbf{w}}_{y_{k,i},r,k}^{(\tilde{v})}, \boldsymbol{\xi}_{k,i} \rangle + \frac{\eta \sigma_p^2 d}{2Nm}(-\ell'^{(\tilde{v})}_{k,i}) - \frac{\eta}{Nm} 2\sigma_p^2 \sqrt{d \log(4n^2/\delta)} \sum_{i' \in [N], i' \neq i} (-\ell'^{(\tilde{v})}_{k,i'})$$

$$\overset{(b)}{\geq} \langle \widetilde{\mathbf{w}}_{y_{k,i},r,k}^{(\tilde{v})}, \boldsymbol{\xi}_{k,i} \rangle + \frac{\eta \sigma_p^2 d}{2Nm}(-\ell'^{(\tilde{v})}_{k,i}) - \frac{\eta}{m} 2\sigma_p^2 \sqrt{d \log(4n^2/\delta)} C_2 (-\ell'^{(\tilde{v})}_{k,i})$$

$$\overset{(c)}{\geq} \langle \widetilde{\mathbf{w}}_{y_{k,i},r,k}^{(\tilde{v})}, \boldsymbol{\xi}_{k,i} \rangle$$

$$> 0.$$

Here $(a)$ follows from Lemma 5; $(b)$ follows from 3; $(c)$ follows from Assumption 1 by choosing a sufficiently large $d$.

**Case 2:** $(\tilde{v}+1) \pmod{\tau} = 0$.

From our induction hypothesis we know that $\langle \widetilde{\mathbf{w}}_{y_{k,i},r,k}^{(\tilde{v}+1-\tau+s)}, \boldsymbol{\xi}_{k,i} \rangle > 0$ for all $0 \leq s \leq \tau - 1$. Then,

$$\langle \widetilde{\mathbf{w}}_{y_{k,i},r,k}^{(\tilde{v})}, \boldsymbol{\xi}_{k,i} \rangle = \langle \widetilde{\mathbf{w}}_{y_{k,i},r,k}^{(\tilde{v}+1-\tau)}, \boldsymbol{\xi}_{k,i} \rangle + \underbrace{\frac{\eta}{nm} \sum_{s=0}^{\tau-1} (-\ell'^{(\tilde{v}+1-\tau+s)}_{k,i}) \|\boldsymbol{\xi}_{k,i}\|_2^2}_{I_1}$$

$$+ \underbrace{\frac{\eta}{nm} \sum_{s=0}^{\tau-1} \sum_{i' \in [N], i' \neq i} (-\ell'^{(\tilde{v}+1-\tau+s)}_{k,i'}) \sigma' \left( \langle \widetilde{\mathbf{w}}_{y_{k,i},r,k}^{(\tilde{v}+1-\tau+s)}, \boldsymbol{\xi}_{k,i'} \rangle \right) \langle \boldsymbol{\xi}_{k,i}, \boldsymbol{\xi}_{k,i'} \rangle}_{I_2}$$

$$+ \underbrace{\frac{\eta}{nm} \sum_{s=0}^{\tau-1} \sum_{k', k' \neq k} \sum_{i' \in [N]} (-\ell'^{(\tilde{v}+1-\tau+s)}_{k',i'}) \sigma' \left( \langle \widetilde{\mathbf{w}}_{y_{k,i},r,k'}^{(\tilde{v}+1-\tau+s)}, \boldsymbol{\xi}_{k',i'} \rangle \right) \langle \boldsymbol{\xi}_{k,i}, \boldsymbol{\xi}_{k',i'} \rangle}_{I_3} \tag{48}$$

Using Lemma 5 we can lower bound $I_1$ as follows:

$$I_1 \geq \frac{\eta \sigma_p^2 d}{2nm} \sum_{s=0}^{\tau-1} (-\ell'^{(\tilde{v}+1-\tau+s)}_{k,i}),$$

where the inequality follows from Lemma 5.

For $|I_2|$ we have,

Lemma 5 as follows:

$$|I_2| \overset{(a)}{\leq} \frac{\eta 2\sigma_p^2 \sqrt{d \log(4n^2/\delta)}}{nm} \sum_{s=0}^{\tau-1} \sum_{i' \in [N], i' \neq i} (-\ell'^{(\tilde{v}+1-\tau+s)}_{k,i'})$$

$$\overset{(b)}{\leq} \frac{\eta(N-1)C_2 2\sigma_p^2 \sqrt{d \log(4n^2/\delta)}}{nm} \sum_{s=0}^{\tau-1} (-\ell'^{(\tilde{v}+1-\tau+s)}_{k,i}).$$

Here $(a)$ follows from Lemma 5; $(b)$ follows from 3. Similarly we can bound $|I_3|$ as follows,

$$
\begin{aligned}
|I_3| &\overset{(a)}{\leq} \frac{\eta 2\sigma_p^2 \sqrt{d\log(4n^2/\delta)}}{nm} \sum_{s=0}^{\tau-1} \sum_{k',k'\neq k} \sum_{i'\in[N]} (-\ell_{k',i'}'^{(\tilde{v}+1-\tau+s)}) \\
&\overset{(b)}{\leq} \frac{\eta(n-N)C_2 2\sigma_p^2\sqrt{d\log(4n^2/\delta)}}{nm} \sum_{s=0}^{\tau-1} (-\ell_{k,i}'^{(\tilde{v}+1-\tau+s)}).
\end{aligned}
$$

Here $(a)$ follows from Lemma 5; $(b)$ follows from 3. Substituting the bounds for $I_1, |I_2|, |I_3|$ in equation 48 we have,

$$
\begin{aligned}
\langle \widetilde{\mathbf{w}}_{y_{k,i},r,k}^{(\tilde{v})}, \boldsymbol{\xi}_{k,i} \rangle &\geq \langle \widetilde{\mathbf{w}}_{y_{k,i},r,k}^{(\tilde{v}+1-\tau)}, \boldsymbol{\xi}_{k,i} \rangle + I_1 - |I_2| - |I_3| \\
&\geq \langle \widetilde{\mathbf{w}}_{y_{k,i},r,k}^{(\tilde{v}+1-\tau)}, \boldsymbol{\xi}_{k,i} \rangle + \frac{\eta\sigma_p^2 d}{2nm} \sum_{s=0}^{\tau-1} (-\ell_{k,i}'^{(\tilde{v}+1-\tau)+s}) \\
&\quad - \frac{\eta C_2}{m} 2\sigma_p^2 \sqrt{d\log(4n^2/\delta)} \sum_{s=0}^{\tau-1} (-\ell_{k,i}'^{(\tilde{v}+1-\tau+s)}) \\
&\overset{(a)}{\geq} \langle \widetilde{\mathbf{w}}_{y_{k,i},r,k}^{(\tilde{v}+1-\tau)}, \boldsymbol{\xi}_{k,i} \rangle \\
&\geq 0.
\end{aligned}
$$

Here $(a)$ follows from Assumption 1 by choosing a sufficiently large $d$. Thus we have shown that $\langle \widetilde{\mathbf{w}}_{y_{k,i},r,k}^{(v)}, \boldsymbol{\xi}_{k,i} \rangle \geq 0$ for all $0 \leq v \leq v'$ and $r$ such that $\langle \mathbf{w}_{y_{k,i},r,k}^{(0)}, \boldsymbol{\xi}_{k,i} \rangle \geq 0$. This implies $S_{k,i}^{(0)} \subseteq S_{k,i}^{(v)}$ for all $0 \leq v \leq v'$. Furthermore we know that $\left| S_{k,i}^{(0)} \right| \geq 0.4m$ for all $k \in [K], i \in [N]$ from Lemma 7 and thus $\left| S_{k,i}^{(v)} \right| \geq 0.4m$ for all $k \in [K], i \in [N], 0 \leq v \leq v'$. □

*Proof of 5.* Note that as part of the proof of 4 we have already shown that $\langle \widetilde{\mathbf{w}}_{j,r,k}^{(v)}, \boldsymbol{\xi}_{k,i} \rangle \geq 0$ for all $0 \leq v \leq v'$ and $k, i$ such that $y_{k,i} = j$ and $\langle \widetilde{\mathbf{w}}_{j,r,k}^{(0)}, \boldsymbol{\xi}_{k,i} \rangle \geq 0$. This implies $\tilde{S}_{j,r}^{(0)} \subseteq \tilde{S}_{j,r}^{(v)}$ for all $0 \leq v \leq v'$. Furthermore we know that $\left| \tilde{S}_{j,r}^{(0)} \right| \geq n/8$ for all $j \in \{\pm 1\}, r \in [m]$ from Lemma 8 and thus $\left| \tilde{S}_{j,r}^{(v)} \right| \geq n/8$ for all $j \in \{\pm 1\}, r \in [m]$.

This concludes the proof of Lemma 17. □

We are now ready to prove Theorem 3.

### C.2.4 Proof of Theorem 3

We will again use a proof by induction to prove this theorem.

*Proof of equation 29.* For $j = y_{k,i}$ we know from equation 26 that $\underline{\mathbb{P}}_{j,r,k,i}^{(v'+1)} = 0$ and hence we look at the case where $j \neq y_{k,i}$.

**Case 1:** $(v'+1) \pmod{\tau} \neq 0$.

**a)** If $\underline{\mathbb{P}}_{j,r,k,i}^{(v')} < -0.5\beta - 4\sqrt{\frac{\log(6n^2/\delta)}{d}} n\alpha$, then from equation 33 in Lemma 11 we know that,

$$
\begin{aligned}
\langle \widetilde{\mathbf{w}}_{j,r,k}^{(v')}, \boldsymbol{\xi}_{k,i} \rangle &\leq \langle \mathbf{w}_{j,r}^{(0)}, \boldsymbol{\xi}_{k,i} \rangle + \underline{\mathbb{P}}_{j,r,k,i}^{(v')} + 4\sqrt{\frac{\log(6n^2/\delta)}{d}} n\alpha \\
&\overset{(a)}{\leq} 0.5\beta + \underline{\mathbb{P}}_{j,r,k,i}^{(v')} + 4\sqrt{\frac{\log(6n^2/\delta)}{d}} n\alpha \\
&\overset{(b)}{<} 0.
\end{aligned}
$$

Here $(a)$ follows from definition of $\beta$ in Theorem 3; $(b)$ follows from $\underline{\mathbb{P}}_{j,r,k,i}^{(v')} < -0.5\beta - 4\sqrt{\frac{\log(6n^2/\delta)}{d}}n\alpha$. Now using the fact that $\langle \widetilde{\mathbf{w}}_{j,r,k}^{(v')}, \boldsymbol{\xi}_{k,i}\rangle < 0$ we have $\sigma'\left(\langle \widetilde{\mathbf{w}}_{j,r,k}^{(v')}, \boldsymbol{\xi}_{k,i}\rangle\right) = 0$, which implies $\underline{\mathbb{P}}_{j,r,k,i}^{(v'+1)} = \underline{\mathbb{P}}_{j,r,k,i}^{(v')} \geq -\beta - 8\sqrt{\frac{\log(6n^2/\delta)}{d}}n\alpha$ using the induction hypothesis.

**b).** If $\underline{\mathbb{P}}_{j,r,k,i}^{(v')} \geq -0.5\beta - 4\sqrt{\frac{\log(6n^2/\delta)}{d}}n\alpha$, then from equation 26 we have,

$$
\begin{aligned}
\underline{\mathbb{P}}_{j,r,k,i}^{(v'+1)} &= \underline{\mathbb{P}}_{j,r,k,i}^{(v')} + \frac{\eta}{Nm}\ell_{k,i}'^{(v')}\sigma'\left(\langle \widetilde{\mathbf{w}}_{j,r,k}^{(v')}, \boldsymbol{\xi}_{k,i}\rangle\right)\|\boldsymbol{\xi}_{k,i}\|_2^2\,\mathbb{1}\left(j = -y_{k,i}\right) \\
&\overset{(a)}{\geq} -0.5\beta - 4\sqrt{\frac{\log(6n^2/\delta)}{d}}n\alpha - \frac{3\eta\sigma_p^2 d}{2Nm} \\
&\overset{(b)}{\geq} -\beta - 8\sqrt{\frac{\log(6n^2/\delta)}{d}}n\alpha.
\end{aligned}
\tag{49}
$$

Here $(a)$ follows from $|\ell'(\cdot)| \leq 1$ and Lemma 5; $(b)$ follows from $\frac{3\eta\sigma_p^2 d}{2Nm} \leq 4\sqrt{\frac{\log(6n^2/\delta)}{d}}n\alpha$ using Assumption 5.

**Case 2:** $(v'+1) \pmod{\tau} = 0$.

In this case, from equation 26 we have,

$$
\begin{aligned}
\underline{\mathbb{P}}_{j,r,k,i}^{(v'+1)} &= \underline{\mathbb{P}}_{j,r,k,i}^{(v'+1-\tau)} + \frac{\eta}{nm}\underbrace{\sum_{s=0}^{\tau-1}\ell_{k,i}'^{(v'+1-\tau+s)}\sigma'\left(\langle \widetilde{\mathbf{w}}_{j,r,k}^{(v'+1-\tau+s)}, \boldsymbol{\xi}_{k,i}\rangle\right)\|\boldsymbol{\xi}_{k,i}\|_2^2\,\mathbb{1}\left(j = -y_{k,i}\right)}_{:=I_2} \\
&= \underline{\mathbb{P}}_{j,r,k,i}^{(v'+1-\tau)} + \frac{\eta}{nm}I_2.
\end{aligned}
\tag{50}
$$

Now suppose instead of doing the update in equation 50, we performed the following hypothetical update:

$$
\begin{aligned}
\dot{\underline{\mathbb{P}}}_{j,r,k,i}^{(v'+1)} &= \underline{\mathbb{P}}_{j,r,k,i}^{(v')} + \frac{\eta}{Nm}\ell_{k,i}'^{(v')}\sigma'\left(\langle \widetilde{\mathbf{w}}_{j,r,k}^{(v')}, \boldsymbol{\xi}_{k,i}\rangle\right)\|\boldsymbol{\xi}_{k,i}\|_2^2\,\mathbb{1}\left(j = -y_{k,i}\right) \\
&\overset{(a)}{=} \underline{\mathbb{P}}_{j,r,k,i}^{(v'+1-\tau)} + \frac{\eta}{Nm}\sum_{s=0}^{\tau-1}\ell_{k,i}'^{(v'+1-\tau+s)}\sigma'\left(\langle \widetilde{\mathbf{w}}_{j,r,k}^{(v'+1-\tau+s)}, \boldsymbol{\xi}_{k,i}\rangle\right)\|\boldsymbol{\xi}_{k,i}\|_2^2\,\mathbb{1}\left(j = -y_{k,i}\right) \\
&= \underline{\mathbb{P}}_{j,r,k,i}^{(v'+1-\tau)} + \frac{\eta}{Nm}I_2.
\end{aligned}
$$

Here $(a)$ uses equation 26 for $v = [v'+1-\tau : v']$. From the argument in Case 1 we know that $\dot{\underline{\mathbb{P}}}_{j,r,k,i}^{(v'+1)} \geq -\beta - 8\sqrt{\frac{\log(6n^2/\delta)}{d}}n\alpha$. Observe that $\underline{\mathbb{P}}_{j,r,k,i}^{(v'+1)} \geq \dot{\underline{\mathbb{P}}}_{j,r,k,i}^{(v'+1)}$ since $I_2 \leq 0$ and $N \leq n$ and thus $\underline{\mathbb{P}}_{j,r,k,i}^{(v'+1)} \geq -\beta - 8\sqrt{\frac{\log(6n^2/\delta)}{d}}n\alpha$. $\qquad\square$

*Proof of equation 28.* We know from equation 25 that for $j \neq y_{k,i}$, $\overline{\mathbb{P}}_{j,r,k,i}^{(v')} = 0$ for all $0 \leq v' \leq T^*\tau - 1$ and hence we focus on the case where $j = y_{k,i}$.

**Case 1:** $(v'+1) \pmod{\tau} \neq 0$.

Let $v'_{j,r,k,i}$ be the last iteration such that $v'_{j,r,k,i} \pmod{\tau} = 0$ and $\overline{\mathbb{P}}_{j,r,k,i}^{(v'_{j,r,k,i})} \leq 0.5\alpha$ and let $s$ be the maximum value in $\{0, 1, \ldots, \tau - 1\}$ such that $\overline{\mathbb{P}}_{j,r,k,i}^{(v'_{j,r,k,i}+s)} \leq 0.5\alpha$. Define $v_{j,r,k,i} = v'_{j,r,k,i} + s$. We see that for all

$v > v_{j,r,k,i}$ we have $\overline{\mathbb{P}}_{j,r,k,i}^{(v)} > 0.5\alpha$. Furthermore,

$$\overline{\mathbb{P}}_{j,r,k,i}^{(v'+1)} \overset{(a)}{\leq} \overline{\mathbb{P}}_{j,r,k,i}^{(v_{j,r,k,i})} - \underbrace{\frac{\eta}{Nm}\ell'^{(v_{j,r,k,i})}_{k,i}\sigma'\left(\langle\widetilde{\mathbf{w}}_{j,r,k}^{(v_{j,r,k,i})}, \boldsymbol{\xi}_{k,i}\rangle\right)\|\boldsymbol{\xi}_{k,i}\|_2^2 \mathbb{1}\left(j=y_{k,i}\right)}_{L_1}$$

$$- \underbrace{\sum_{v_{j,r,k,i}<v\leq v'} \frac{\eta}{Nm}\ell'^{(v)}_{k,i}\sigma'\left(\langle\widetilde{\mathbf{w}}_{j,r,k}^{(v)}, \boldsymbol{\xi}_{k,i}\rangle\right)\|\boldsymbol{\xi}_{k,i}\|_2^2 \mathbb{1}\left(j=y_{k,i}\right)}_{L_2}. \tag{51}$$

Here $(a)$ uses the fact that we are avoiding the scaling down by a factor of $\frac{1}{K}$ which occurs at every $v$ $(\mod \tau) = 0$ (see equation 25) for $v'_{j,r,k,i} < v \leq v'$.

We know $\overline{\mathbb{P}}_{j,r,k,i}^{(v_{j,r,k,i})} \leq 0.5\alpha$. We can bound $L_1$ and $L_2$ as follows:

$$L_1 \overset{(a)}{\leq} \frac{\eta}{Nm}\|\boldsymbol{\xi}_{k,i}\|_2^2 \overset{(b)}{\leq} \frac{3\eta\sigma_p^2 d}{2Nm} \overset{(c)}{\leq} 1 \overset{(d)}{\leq} 0.25\alpha.$$

Here $(a)$ uses $|\ell'(\cdot)| \leq 1, \sigma'(\cdot) \leq 1$; $(b)$ uses Lemma 5; $(c)$ uses Assumption 5; $(d)$ uses $T^*\tau \geq e$.

Now note that for $v_{j,r,k,i} < v \leq v'$ since $\overline{\mathbb{P}}_{j,r,k,i}^{(v)} \geq 0.5\alpha$ we have,

$$\langle\widetilde{\mathbf{w}}_{j,r,k}^{(v)}, \boldsymbol{\xi}_{k,i}\rangle \overset{(a)}{\geq} \langle\mathbf{w}_{j,r,k}^{(0)}, \boldsymbol{\xi}_{k,i}\rangle + \overline{\mathbb{P}}_{j,r,k,i}^{(v)} - 4\sqrt{\frac{\log(6n^2/\delta)}{d}}n\alpha$$

$$\overset{(b)}{\geq} -0.5\beta + 0.5\alpha - 4\sqrt{\frac{\log(6n^2/\delta)}{d}}n\alpha$$

$$\overset{(c)}{\geq} 0.25\alpha. \tag{52}$$

Here $(a)$ follows from Lemma 11, $(b)$ follows from the definition of $\beta$ (see Theorem 3) and $\overline{\mathbb{P}}_{j,r,k,i}^{(v)} \geq 0.5\alpha$, $(c)$ follows from $\beta \leq \frac{1}{12} \leq 0.1\alpha$ and $4\sqrt{\frac{\log(6n^2/\delta)}{d}}n\alpha \leq 0.2\alpha$ using Assumption 1.

Substituting the bound above in $L_2$ we have,

$$|L_2| \overset{(a)}{\leq} \sum_{v_{j,r,k,i}<v\leq v'} \frac{\eta}{Nm}\exp\left(-\langle\widetilde{\mathbf{w}}_{j,r,k}^{(v)}, \boldsymbol{\xi}_{k,i}\rangle + 0.5\right)\sigma'\left(\langle\widetilde{\mathbf{w}}_{j,r,k}^{(v)}, \boldsymbol{\xi}_{k,i}\rangle\right)\|\boldsymbol{\xi}_{k,i}\|_2^2 \mathbb{1}\left(j=y_{k,i}\right)$$

$$\overset{(b)}{\leq} \sum_{v_{j,r,k,i}<v\leq v'} \frac{2\eta}{Nm}\exp\left(-\langle\widetilde{\mathbf{w}}_{j,r,k}^{(v)}, \boldsymbol{\xi}_{k,i}\rangle\right)\|\boldsymbol{\xi}_{k,i}\|_2^2 \tag{53}$$

$$\overset{(c)}{\leq} \sum_{v_{j,r,k,i}<v\leq v'} \frac{2\eta}{Nm}\exp(-0.25\alpha)\frac{3\sigma_p^2 d}{2}$$

$$= \frac{2\eta(v'-v_{j,r,k,i}-1)}{Nm}\exp(-\log T^*\tau)\frac{3\sigma_p^2 d}{2}$$

$$\leq \frac{2\eta(T^*\tau)}{Nm}\exp(-\log T^*\tau)\frac{3\sigma_p^2 d}{2}$$

$$= \frac{3\eta\sigma_p^2 d}{Nm}$$

$$\overset{(d)}{\leq} 0.25\alpha.$$

For $(a)$ we use Lemma 13; for $(b)$ we use $\exp(0.5) \leq 2$ and $\langle\widetilde{\mathbf{w}}_{j,r,k}^{(v)}, \boldsymbol{\xi}_{k,i}\rangle \geq 0$ from equation 52, $(c)$ follows from Lemma 5 and equation 52; $(d)$ follows from Assumption 5.

Thus substituting the bounds for $L_1$ and $L_2$ we have,

$$\overline{\mathbb{P}}_{j,r,k,i}^{(v'+1)} \leq \alpha,$$

which completes our proof.

**Case 2:** $(v'+1) \pmod{\tau} = 0$.

Suppose instead of doing the update in equation 25, we performed the following hypothetical update

$$\dot{\overline{\mathbb{P}}}_{j,r,k,i'}^{(v'+1)} = \overline{\mathbb{P}}_{j,r,k,i}^{(v')} - \frac{\eta}{Nm} \ell'^{(v')}_{k,i} \sigma'\left(\langle \widetilde{\mathbf{w}}_{j,r,k}^{(v')}, \boldsymbol{\xi}_{k,i} \rangle\right) \|\boldsymbol{\xi}_{k,i}\|_2^2 \mathbb{1}\left(j = y_{k,i}\right). \tag{54}$$

From the argument in Case 1 we know that $\dot{\overline{\mathbb{P}}}_{j,r,k,i'}^{(v'+1)} \leq \alpha$. Observe that $\overline{\mathbb{P}}_{j,r,k,i}^{(v'+1)} \leq \dot{\overline{\mathbb{P}}}_{j,r,k,i'}^{(v'+1)}$ and thus $\overline{\mathbb{P}}_{j,r,k,i}^{(v'+1)} \leq \alpha$. $\qquad\square$

*Proof of equation 30.* This part bounds $\mathbb{G}_{j,r,k}^{(v'+1)}$. To do so we show that the growth of $\mathbb{G}_{j,r,k}^{(v'+1)}$ is upper bounded by the growth of $\overline{\mathbb{P}}_{y_{k,1},r^*,k,1}^{(v'+1)}$ for any $r^* \in S_{k,1}^{(0)}$, that is,

$$\frac{\mathbb{G}_{j,r,k}^{(v'+1)}}{\overline{\mathbb{P}}_{y_{k,1},r^*,k,1}^{(v'+1)}} \leq C'\widehat{\gamma}.$$

We will again use a proof by induction. We first argue the base case of our induction. Since $r^* \in S_{k,1}^{(0)} \subseteq S_{k,1}^{(v)}$, so,

$$\begin{aligned}
\overline{\mathbb{P}}_{y_{k,1},r^*,k,1}^{(1)} &= \underbrace{\overline{\mathbb{P}}_{y_{k,1},r^*,k,1}^{(0)}}_{=0} - \frac{\eta}{Nm} \ell'^{(0)}_{k,1} \underbrace{\sigma'\left(\left\langle \mathbf{w}_{y_{k,1},r^*,k}^{(0)}, \boldsymbol{\xi}_{k,1} \right\rangle\right)}_{=1(\because r^* \in S_{k,1}^{(0)})} \|\boldsymbol{\xi}_{k,1}\|_2^2 \\
&= \frac{\eta \|\boldsymbol{\xi}_{k,1}\|_2^2}{Nm}\left(-\ell'^{(0)}_{k,1}\right) \overset{(a)}{\geq} \frac{\eta \sigma_p^2 d}{2Nm},
\end{aligned}$$

where $(a)$ follows from Lemma 5. On the other hand,

$$\mathbb{G}_{j,r,k}^{(1)} = \underbrace{\mathbb{G}_{j,r,k}^{(0)}}_{=0} - \frac{\eta}{Nm} \sum_{i \in [N]} \ell'^{(0)}_{k,i} \sigma'\left(\langle \mathbf{w}_{j,r,k}^{(0)}, y_{k,i}\boldsymbol{\mu} \rangle\right) \|\boldsymbol{\mu}\|_2^2 \leq \frac{\|\boldsymbol{\mu}\|_2^2 \eta}{m}.$$

Therefore,

$$\frac{\mathbb{G}_{j,r,k}^{(1)}}{\overline{\mathbb{P}}_{y_{k,1},r^*,k,1}^{(1)}} \leq \frac{2N \|\boldsymbol{\mu}\|_2^2}{\sigma_p^2 d} \leq C'\widehat{\gamma},$$

if $C' \geq 2$. Now assuming equation 55 holds at $v'$ we have the following cases for $(v'+1)$.

$$\frac{\mathbb{G}_{j,r,k}^{(v)}}{\overline{\mathbb{P}}_{y_{k,1},r^*,k,1}^{(v)}} \leq C'\widehat{\gamma}.$$

**Case 1:** $(v'+1) \pmod{\tau} \neq 0$. From equation 24 we have,

$$\begin{aligned}
\mathbb{G}_{j,r,k}^{(v'+1)} &= \mathbb{G}_{j,r,k}^{(v')} + \frac{\eta}{Nm} \sum_{i \in [N]} (-\ell'^{(v')}_{k,i}) \sigma'\left(\langle \widetilde{\mathbf{w}}_{j,r,k}^{(v')}, y_{k,i}\boldsymbol{\mu} \rangle\right) \|\boldsymbol{\mu}\|_2^2 \\
&\overset{(a)}{\leq} \mathbb{G}_{j,r,k}^{(v')} + \frac{\eta C_2}{m}(-\ell'^{(v')}_{k,1}) \|\boldsymbol{\mu}\|_2^2
\end{aligned}$$

$$\tag{55}$$

where $(a)$ follows from part (3) in Lemma 17. At the same time since $\langle \mathbf{w}_{y_{k,1},r^*,k}^{(v)}, \boldsymbol{\xi}_{k,1} \rangle \geq 0$ for any $r^* \in S_{k,1}^{(0)}$ and for all $0 \leq v \leq T^*\tau - 1$, we have from equation 25:

$$\overline{\mathbb{P}}_{y_{k,1},r^*,k,1}^{(v'+1)} = \overline{\mathbb{P}}_{y_{k,1},r^*,k,1}^{(v')} + \frac{\eta}{Nm}(-\ell_{k,1}'^{(v')}) \|\boldsymbol{\xi}_{k,1}\|_2^2$$

$$\overset{(a)}{\geq} \overline{\mathbb{P}}_{y_{k,1},r^*,k,1}^{(v')} + \frac{\eta}{Nm}(-\ell_{k,1}'^{(v')}) \frac{\sigma_p^2 d}{2},$$

where $(a)$ follows from Lemma 5.

Thus,

$$\frac{\mathbb{G}_{j,r,k}^{(v'+1)}}{\overline{\mathbb{P}}_{y_{k,1},r^*,k,1}^{(v'+1)}} \leq \max\left\{ \frac{\mathbb{G}_{j,r,k}^{(v')}}{\overline{\mathbb{P}}_{y_{k,1},r^*,k,1}^{(v')}}, \frac{2C_2 N \|\boldsymbol{\mu}\|_2^2}{\sigma_p^2 d} \right\} \overset{(a)}{\leq} \max\{C'\widehat{\gamma}, 2C_2\widehat{\gamma}\} \overset{(b)}{\leq} C'\widehat{\gamma}.$$

Here $(a)$ follows from the definition of $\widehat{\gamma}$; $(b)$ follows from setting $C' = 2C_2$.

**Case 2:** $(v'+1) \pmod{\tau} = 0$.

We have from equation 24,

$$\mathbb{G}_{j,r,k}^{(v'+1)} = \mathbb{G}_{j,r,k}^{(v'+1-\tau)} + \frac{\eta}{nm} \sum_{s=0}^{\tau-1} \sum_{k'} \sum_{i \in [N]} (-\ell_{k',i}'^{(v'+1-\tau+s)}) \sigma'\left(\langle \widetilde{\mathbf{w}}_{j,r,k}^{(v-\tau+s)}, y_{k,i}\boldsymbol{\mu} \rangle\right) \|\boldsymbol{\mu}\|_2^2$$

$$\overset{(a)}{\leq} \mathbb{G}_{j,r,k}^{(v'+1-\tau)} + \frac{\eta C_2}{m} \sum_{s=0}^{\tau-1} (-\ell_{k,1}'^{(v'+1-\tau+s)}) \|\boldsymbol{\mu}\|_2^2,$$

where $(a)$ follows from part (3) in Lemma 17. At the same time since $\langle \mathbf{w}_{y_{k,1},r^*,k}^{(v)}, \boldsymbol{\xi}_{k,1} \rangle \geq 0$ for any $r^* \in S_{k,1}^{(0)}$ and for all $0 \leq v \leq T^*\tau - 1$, we have from equation 25,

$$\overline{\mathbb{P}}_{y_{k,1},r^*,k,1}^{(v'+1)} = \overline{\mathbb{P}}_{y_{k,1},r^*,k,1}^{(v'+1-\tau)} + \frac{\eta}{nm} \sum_{s=0}^{\tau-1} (-\ell_{k,1}'^{(v'+1-\tau+s)}) \|\boldsymbol{\xi}_{k,1}\|_2^2$$

$$\overset{(a)}{\geq} \overline{\mathbb{P}}_{y_{k,1},r^*,k,1}^{(v'+1-\tau)} + \frac{\eta}{nm} \sum_{s=0}^{\tau-1} (-\ell_{k,1}'^{(v'+1-\tau+s)}) \frac{\sigma_p^2 d}{2},$$

where $(a)$ follows from Lemma 5. Thus,

$$\frac{\mathbb{G}_{j,r,k}^{(v'+1)}}{\overline{\mathbb{P}}_{y_{k,1},r^*,k,1}^{(v'+1)}} \leq \max\left\{ \frac{\mathbb{G}_{j,r,k}^{(v'+1-\tau)}}{\overline{\mathbb{P}}_{y_{k,1},r^*,k,1}^{(v'+1-\tau)}}, \frac{2C_2 n \|\boldsymbol{\mu}\|_2^2}{\sigma_p^2 d} \right\} \overset{(a)}{\leq} \max\{C'\widehat{\gamma}, 2C_2\widehat{\gamma}\} \overset{(b)}{\leq} C'\widehat{\gamma}.$$

Here $(a)$ follows from the definition of $\widehat{\gamma}$; $(b)$ follows from setting $C' = 2C_2$. Thus we have shown $\mathbb{G}_{j,r,k}^{(v'+1)} \leq C'\widehat{\gamma}\overline{\mathbb{P}}_{y_{k,1},r^*,k,1}^{(v'+1)} \leq C'\widehat{\gamma}\alpha$ where the last inequality follows from $\overline{\mathbb{P}}_{y_{k,1},r^*,k,1}^{(v'+1)} \leq \alpha$. $\qquad\square$

Now that we have proved Theorem 3, that is, equation 28, equation 29 and equation 30 hold for all $0 \leq v \leq T^*\tau - 1$, we state a simple proposition that extends the result in Lemma 17 for all $0 \leq v \leq T^*\tau - 1$.

**Proposition 2.** *Under assumptions, for all $0 \leq v \leq T^*\tau - 1$ we have*

1. $\frac{1}{m} \sum_{r=1}^{m} \left[ \overline{\mathbb{P}}_{y_{k,i},r,k,i}^{(v)} - \overline{\mathbb{P}}_{y_{k',i'},r,k',i'}^{(v)} \right] \leq \kappa$ *for all* $k, k' \in [K], i, i' \in [N]$.

2. $y_{k,i} f(\widetilde{\mathbf{W}}_k^{(v)}, \mathbf{x}_{k,i}) - y_{k',i'} f(\widetilde{\mathbf{W}}_{k'}^{(v)}, \mathbf{x}_{k',i'}) \leq C_1$ *for all* $k, k' \in [K]$ *and* $i, i' \in [N]$.

3. $\frac{\ell_{k',i'}'^{(v)}}{\ell_{k,i}'^{(v)}} \leq C_2 = \exp(C_1)$ *for all* $k, k' \in [K]$ *and* $i, i' \in [N]$.

4. $S_{k,i}^{(0)} \subseteq S_{k,i}^{(v)}$ where $S_{k,i}^{(v)} := \left\{ r \in [m] : \langle \widetilde{\mathbf{w}}_{y_{k,i},r,k}^{(v)}, \boldsymbol{\xi}_{k,i} \rangle \geq 0 \right\}$, and hence $\left| S_{k,i}^{(v)} \right| \geq 0.4m$ for all $k \in [K], i \in [N]$.

5. $\tilde{S}_{j,r}^{(0)} \subseteq \tilde{S}_{j,r}^{(v)}$ where $\tilde{S}_{j,r}^{(v)} := \left\{ k \in [K], i \in [N] : y_{k,i} = j, \langle \widetilde{\mathbf{w}}_{j,r,k}^{(v)}, \boldsymbol{\xi}_{k,i} \rangle \geq 0 \right\}$, and hence $\left| \tilde{S}_{j,r}^{(v)} \right| \geq \frac{n}{8}$.

Here we take $\kappa = 5$ and $C_1 = 6.75$.

### C.3 First Stage of Training.

Define,

$$T_1 = \frac{C_3 nm}{\eta \sigma_p^2 d \tau} \tag{56}$$

where $C_3 = \Theta(1)$ is some large constant. In this stage, our goal is to show that $\overline{P}_{y_{k,i},r^*,k,i}^{(T_1)} \geq 2$ for all $r^*$ such that $r^* \in S_{k,i}^{(0)} := \left\{ r \in [m] : \langle \mathbf{w}_{y_{k,i},r^*}^{(0)}, \boldsymbol{\xi}_{k,i} \rangle \geq 0 \right\}$. To do so, we first introduce the following lemmas.

**Lemma 18.** *For all $0 \leq t \leq T_1 - 1$ and $0 \leq s \leq \tau - 1$ we have,*

$$\max_{j,r,k} \left\{ \Gamma_{j,r}^{(t)} + \gamma_{j,r,k}^{(t,s)} \right\} \leq \frac{C_3 n \|\boldsymbol{\mu}\|_2^2}{\sigma_p^2 d} = \mathcal{O}(1).$$

*Proof.* We have,

$$
\begin{aligned}
\Gamma_{j,r}^{(t)} + \gamma_{j,r,k}^{(t,s)} &= -\frac{\eta}{nm} \sum_{t'=0}^{t-1} \sum_k \sum_{i \in [N]} \sum_{s=0}^{\tau-1} \ell'^{(t',s)}_{k,i} \sigma' \left( \langle \mathbf{w}_{j,r,k}^{(t',s)}, y_{k,i} \boldsymbol{\mu} \rangle \right) \|\boldsymbol{\mu}\|_2^2 \\
&\quad - \frac{\eta}{Nm} \sum_{s'=0}^{s} \sum_{i \in [N]} \ell'^{(t,s')}_{k,i} \sigma' \left( \langle \mathbf{w}_{j,r,k}^{(t,s')}, y_{k,i} \boldsymbol{\mu} \rangle \right) \|\boldsymbol{\mu}\|_2^2 \\
&\stackrel{(a)}{\leq} -\frac{\eta}{nm} \sum_{t'=0}^{t-1} \sum_k \sum_{i \in [N]} \sum_{s=0}^{\tau-1} \ell'^{(t',s)}_{k,i} \|\boldsymbol{\mu}\|_2^2 - \frac{\eta}{Nm} \sum_{s'=0}^{s} \sum_{i \in [N]} \ell'^{(t,s')}_{k,i} \|\boldsymbol{\mu}\|_2^2 \\
&\stackrel{(b)}{\leq} \frac{\eta(t+1)\tau \|\boldsymbol{\mu}\|_2^2}{m} \\
&\leq \frac{\eta T_1 \tau \|\boldsymbol{\mu}\|_2^2}{m} \\
&= \frac{C_3 n \|\boldsymbol{\mu}\|_2^2}{\sigma_p^2 d} \\
&\stackrel{(c)}{=} \mathcal{O}(1).
\end{aligned}
$$

Here $(a)$ follows from $\sigma'(\cdot) \in \{0,1\}$, $(b)$ follows from $|\ell'(\cdot)| \leq 1$, $(c)$ follows from Assumption 1. $\qquad \square$

**Lemma 19.** *For all $0 \leq t \leq T_1 - 1$ and $0 \leq s \leq \tau - 1$ we have,*

$$\max_{j,r,k,i} \left\{ \overline{P}_{j,r,k,i}^{(t)} + \overline{\rho}_{j,r,k,i}^{(t,s)} \right\} = \mathcal{O}(1).$$

*Proof.* We have from equation 14 and equation 20,

$$
\begin{aligned}
\overline{P}_{j,r,k,i}^{(t)} + \overline{\rho}_{j,r,k,i}^{(t,s)} &= -\frac{\eta}{nm} \sum_{t'=0}^{t-1} \sum_{s=0}^{\tau-1} \ell'_{k,i}^{(t',s)} \sigma'\left(\langle \widetilde{\mathbf{w}}_{j,r,k}^{(v')}, \boldsymbol{\xi}_{k,i}\rangle\right) \|\boldsymbol{\xi}_{k,i}\|_2^2 \mathbb{1}\left(y_{k,i}=j\right) \\
&\quad - \frac{\eta}{Nm} \sum_{s'=0}^{s} \ell'_{k,i}^{(t,s')} \sigma'\left(\langle \mathbf{w}_{j,r,k}^{(t,s')}, \boldsymbol{\xi}_{k,i}\rangle\right) \|\boldsymbol{\xi}_{k,i}\|_2^2 \mathbb{1}\left(y_{k,i}=j\right) \\
&\overset{(a)}{\leq} -\frac{\eta}{nm} \sum_{t'=0}^{t-1} \sum_{s=0}^{\tau-1} \ell'_{k,i}^{(t',s)} \|\boldsymbol{\xi}_{k,i}\|_2^2 - \frac{\eta}{Nm} \sum_{s'=0}^{s} \ell'_{k,i}^{(t,s')} \|\boldsymbol{\xi}_{k,i}\|_2^2 \\
&\leq \frac{\eta(t+1)\tau \|\boldsymbol{\xi}_{k,i}\|_2^2}{Nm} \\
&\overset{(b)}{\leq} \frac{3\eta T_1 \tau \sigma_p^2 d}{2Nm} \\
&\leq \frac{3C_3 n}{2N} \\
&= \mathcal{O}\left(1\right).
\end{aligned}
$$

Here $(a)$ follows from $\sigma'(\cdot) \leq 1$, $(b)$ follows from $t \leq T_1 - 1$ and Lemma 5. □

**Lemma 20.** *For any $k \in [K]$ and $i \in [N]$, we have $F_j(\mathbf{W}_{j,k}^{(t,s)}, \mathbf{x}_{k,i}) = \mathcal{O}(1)$ for all $j \in \{\pm 1\}$, $0 \leq t \leq T_1 - 1$ and $0 \leq s \leq \tau - 1$.*

*Proof.* We have,

$$
\begin{aligned}
&F_j(\mathbf{W}_{j,k}^{(t,s)}, \mathbf{x}_{k,i}) \\
&= \frac{1}{m} \sum_{r=1}^{m} \left[ \sigma\left(\langle \mathbf{w}_{j,r,k}^{(t,s)}, y_{k,i}\boldsymbol{\mu}\rangle\right) + \sigma\left(\langle \mathbf{w}_{j,r,k}^{(t,s)}, \boldsymbol{\xi}_{k,i}\rangle\right) \right] \\
&\overset{(a)}{\leq} \frac{1}{m} \sum_{r=1}^{m} \left[ \left|\langle \mathbf{w}_{j,r,k}^{(t,s)}, y_{k,i}\boldsymbol{\mu}\rangle\right| + \left|\langle \mathbf{w}_{j,r,k}^{(t,s)}, \boldsymbol{\xi}_{k,i}\rangle\right| \right] \\
&\overset{(b)}{\leq} \frac{1}{m} \sum_{r=1}^{m} \left[ \left|\langle \mathbf{w}_{j,r}^{(0)}, \boldsymbol{\mu}\rangle\right| + \Gamma_{j,r}^{(t)} + \gamma_{j,r,k}^{(t,s)} + \left|\langle \mathbf{w}_{j,r}^{(0)}, \boldsymbol{\xi}_{k,i}\rangle\right| + \overline{P}_{j,r,k,i}^{(t)} + \overline{\rho}_{j,r,k,i}^{(t,s)} + 4\sqrt{\frac{\log(6n^2/\delta)}{d}} n\alpha \right] \\
&\leq 5 \max_{r \in [m]} \left\{ \left|\langle \mathbf{w}_{j,r}^{(0)}, \boldsymbol{\mu}\rangle\right|, \Gamma_{j,r}^{(t)} + \gamma_{j,r,k}^{(t,s)}, \left|\langle \mathbf{w}_{j,r}^{(0)}, \boldsymbol{\xi}_{k,i}\rangle\right|, \overline{P}_{j,r,k,i}^{(t)} + \overline{\rho}_{j,r,k,i}^{(t,s)}, 4\sqrt{\frac{\log(6n^2/\delta)}{d}} n\alpha \right\} \\
&\overset{(c)}{\leq} 5 \max_{r \in [m]} \left\{ \beta, \Gamma_{j,r}^{(t)} + \gamma_{j,r,k}^{(t,s)}, \overline{P}_{j,r,k,i}^{(t)} + \overline{\rho}_{j,r,k,i}^{(t,s)}, 4\sqrt{\frac{\log(6n^2/\delta)}{d}} n\alpha \right\} \\
&\overset{(d)}{=} \mathcal{O}(1).
\end{aligned}
$$

Here $(a)$ follows from $\sigma(z) \leq |z|$, $(b)$ follows from Lemma 11, $(c)$ follows from the definition of $\beta$, $(d)$ follows from Lemma 10, Lemma 18 and Lemma 19. □

**Lemma 21.** *For all $t \geq T_1$ and $0 \leq s \leq \tau - 1$ we have,*

$$
\overline{P}_{y_{k,i},r^*,k,i}^{(t)} + \overline{\rho}_{y_{k,i},r^*,k,i}^{(t,s)} \geq \overline{P}_{y_{k,i},r^*,k,i}^{(T_1)} \geq 2. \tag{57}
$$

*where $r^* \in S_{k,i}^{(0)} := \left\{ r \in [m] : \langle \mathbf{w}_{y_{k,i},r,k}^{(0)}, \boldsymbol{\xi}_{k,i}\rangle > 0 \right\}$.*

*Proof.* First note that from Lemma 20, we have for any $k \in [K]$, $i \in [N]$, $F_{+1}(\mathbf{W}_{+1,k}^{(t,s)}, \mathbf{x}_{k,i})$, $F_{-1}(\mathbf{W}_{-1,k}^{(t,s)}, \mathbf{x}_{k,i}) = \mathcal{O}(1)$ for all $t \in \{0, 1, \ldots, T_1 - 1\}$, $s \in \{0, 1, \ldots, \tau - 1\}$. Thus there exists a positive constant $C$ such that for

all $0 \le t \le T_1 - 1$ and $0 \le s \le \tau - 1$ we have,

$$-\ell'^{(t',s)}_{k,i} \ge C. \tag{58}$$

Next we know from Proposition 2 part 4 that,

$$\langle \mathbf{w}^{(t,s)}_{y_{k,i},r^*,k}, \boldsymbol{\xi}_{k,i} \rangle > 0 \quad \text{for all } 0 \le t \le T_1 - 1, 0 \le s \le \tau - 1,$$

where $r^* \in S^{(0)}_{k,i} := \left\{ r \in [m] : \langle \mathbf{w}^{(0)}_{y_{k,i},r,k}, \boldsymbol{\xi}_{k,i} \rangle > 0 \right\}$. This implies that for $t \ge T_1$,

$$
\begin{aligned}
\overline{P}^{(t)}_{y_{k,i},r^*,k,i} + \overline{\rho}^{(t,s)}_{y_{k,i},r^*,k,i} &\ge \overline{P}^{(T_1)}_{y_{k,i},r^*,k,i} \\
&\overset{(a)}{=} -\sum_{t'=0}^{T_1} \frac{\eta}{nm} \sum_{s=0}^{\tau-1} \ell'^{(t',s)}_{k,i} \cdot \|\boldsymbol{\xi}_{k,i}\|_2^2 \\
&\overset{(b)}{\ge} \frac{\eta C T_1 \tau \sigma_p^2 d}{2nm} \\
&\overset{(b)}{\ge} 2.
\end{aligned}
\tag{59}
$$

Here $(a)$ follows from equation 20; $(b)$ follows from equation 58 and Lemma 5; $(b)$ follows from the definition of $T_1$ in equation 56 and setting $C_3 = 4/C$.

$\square$

## C.4 Second Stage of Training

In the first stage we have shown that for any $k \in [K]$ and $i \in [N]$, $\overline{P}^{(t)}_{y_{k,i},r^*,k,i} + \overline{\rho}^{(t,s)}_{y_{k,i},r^*,k,i} \ge 2$ for all $t \ge T_1$ and $s \in [0 : \tau - 1]$. Our goal in the second stage is to show that for every round in $T_1 \le t \le T^* - 1$, the loss of the global model is decreasing. To do so, we will show that our objective satisfies the following property

$$\langle \nabla L_k(\mathbf{W}^{(t,s)}_k), \mathbf{W}^{(t,s)}_k - \mathbf{W}^* \rangle \ge L_k(\mathbf{W}^{(t,s)}_k) - \frac{\epsilon}{2\tau},$$

where $\mathbf{W}^*$ is defined as follows.

$$\mathbf{w}^*_{j,r} := \mathbf{w}^{(0)}_{j,r} + 5 \log(2\tau/\epsilon) \left[ \sum_k \sum_{i \in [N]} \mathbb{1}(j = y_{k,i}) \frac{\boldsymbol{\xi}_{k,i}}{\|\boldsymbol{\xi}_{k,i}\|_2^2} \right]. \tag{60}$$

Using this we can easily show that the loss of the global model is decreasing in every round leading to convergence. We now state and prove some intermediate lemmas.

**Lemma 22.** *Under Condition 1, we have*

$$\left\| \mathbf{W}^{(T_1)} - \mathbf{W}^* \right\|_2 = \mathcal{O}\left( \sqrt{\frac{mn}{\sigma_p^2 d}} \log(\tau/\epsilon) \right).$$

*Proof.*

$$\left\|\mathbf{W}^{(T_1)} - \mathbf{W}^*\right\|_2 \leq \left\|\mathbf{W}^{(T_1)} - \mathbf{W}^{(0)}\right\|_2 + \left\|\mathbf{W}^* - \mathbf{W}^{(0)}\right\|_2$$

$$\overset{(a)}{=} \mathcal{O}\left(m^{1/2} \|\boldsymbol{\mu}\|_2^{-1} \max_{j,r} \Gamma_{j,r}^{(T_1)}\right) + \mathcal{O}\left(m^{1/2} n^{1/2} \sigma_p^{-1} d^{-1/2} \max_{j,r,k,i} \left\{\overline{P}_{j,r,k,i}^{(T_1)}, \underline{P}_{j,r,k,i}^{(T_1)}\right\}\right)$$

$$+ \mathcal{O}\left(m^{1/2} n \sigma_p^{-1} d^{-3/4}\right) + \left\|\mathbf{W}^* - \mathbf{W}^{(0)}\right\|_2$$

$$\overset{(b)}{=} \mathcal{O}\left(m^{1/2} n \|\boldsymbol{\mu}\|_2 \sigma_p^{-2} d^{-1}\right) + \mathcal{O}\left(m^{1/2} n^{1/2} \sigma_p^{-1} d^{-1/2}\right) + \mathcal{O}\left(m^{1/2} n^{1/2} \log(\tau/\epsilon) \sigma_p^{-1} d^{-1/2}\right)$$

$$\overset{(c)}{=} \mathcal{O}\left(m^{1/2} n^{1/2} \sigma_p^{-1} d^{-1/2}\right) + \mathcal{O}\left(m^{1/2} n^{1/2} \log(\tau/\epsilon) \sigma_p^{-1} d^{-1/2}\right)$$

$$= \mathcal{O}\left(m^{1/2} n^{1/2} \log(\tau/\epsilon) \sigma_p^{-1} d^{-1/2}\right).$$

Here $(a)$ follows from the following argument:

$$\left\|\mathbf{W}^{(T_1)} - \mathbf{W}^{(0)}\right\|_2^2$$

$$= \sum_{j,r} \left\|\Gamma_{j,r}^{(T_1)} \cdot \|\boldsymbol{\mu}\|_2^{-2} \cdot \boldsymbol{\mu}\right\|_2^2 + \sum_{j,r} \left\|\sum_{k=1}^{K} \sum_{i \in [N]} P_{j,r,k,i}^{(T_1)} \cdot \|\boldsymbol{\xi}_{k,i}\|_2^{-2} \cdot \boldsymbol{\xi}_{k,i}\right\|_2^2$$

$$+ \underbrace{2m \left\langle \Gamma_{j,r}^{(t)} \cdot \|\boldsymbol{\mu}\|_2^{-2} \boldsymbol{\mu}, \sum_{k=1}^{2} \sum_{i \in [N]} P_{j,r,k,i}^{(t)} \cdot \|\boldsymbol{\xi}_{k,i}\|_2^{-2} \cdot \boldsymbol{\xi}_{k,i}\right\rangle}_{=0}$$

$$= \mathcal{O}\left(\frac{m}{\|\boldsymbol{\mu}\|_2^2} \max_{j,r}(\Gamma_{j,r}^{(t)})^2\right) + \mathcal{O}\left(\frac{mn}{\|\boldsymbol{\xi}_{k,i}\|_2^2} \max_{j,r,k,i}(P_{j,r,k,i}^{(t)})^2\right) + \mathcal{O}\left(mn^2 \max_{k,k,k',i'} \frac{\langle \boldsymbol{\xi}_{k,i}, \boldsymbol{\xi}_{k',i'}\rangle}{\|\boldsymbol{\xi}_{k,i}\|_2^4}\right)$$

$$= \mathcal{O}\left(\frac{m}{\|\boldsymbol{\mu}\|_2^2} \max_{j,r}(\Gamma_{j,r}^{(t)})^2\right) + \mathcal{O}\left(\frac{mn}{\|\boldsymbol{\xi}_{k,i}\|_2^2} \max_{j,r,k,i}(P_{j,r,k,i}^{(t)})^2\right) + \mathcal{O}\left(\frac{mn^2}{\sigma_p^2 d^{3/2}}\right)$$

where the last equality follows from Lemma 5. Getting back to our proof, we see that $(b)$ follows from Lemma 18, Lemma 19 and definition of $\mathbf{W}^*$ in equation 60; $(c)$ follows from Assumption 1. $\qquad \square$

**Lemma 23.** *For any $k \in [K]$, $i \in [N]$ we have for all $t \in \{T_1, T_1 + 1, \ldots, T^* - 1\}$, $s \in \{0, 1, \ldots, \tau - 1\}$,*

$$y_{k,i}\langle \nabla f(\mathbf{W}_k^{(t,s)}, \mathbf{x}_{k,i}), \mathbf{W}^*\rangle \geq \log(2\tau/\epsilon).$$

*Proof.*

$$y_{k,i}\langle \nabla f(\mathbf{W}_k^{(t,s)}, \mathbf{x}_{k,i}), \mathbf{W}^* \rangle$$

$$= \frac{1}{m}\sum_{j,r}\sigma'\left(\langle \mathbf{w}_{j,r,k}^{(t,s)}, y_{k,i}\boldsymbol{\mu}\rangle\right)\langle \boldsymbol{\mu}, j\mathbf{w}_{j,r}^*\rangle + \frac{1}{m}\sum_{j,r}\sigma'\left(\langle \mathbf{w}_{j,r,k}^{(t,s)}, \boldsymbol{\xi}_{k,i}\rangle\right)\langle y_{k,i}\boldsymbol{\xi}_{k,i}, j\mathbf{w}_{j,r}^*\rangle$$

$$= \frac{1}{m}\sum_{j,r}\sum_{k',i'}\sigma'\left(\langle \mathbf{w}_{j,r,k}^{(t,s)}, \boldsymbol{\xi}_{k,i}\rangle\right)5\log(2/\epsilon)\mathbb{1}\left(j = y_{k',i'}\right)\frac{\langle y_{k,i}\boldsymbol{\xi}_{k,i}, j\boldsymbol{\xi}_{k',i'}\rangle}{\|\boldsymbol{\xi}_{k',i'}\|_2^2}$$

$$+ \frac{1}{m}\sum_{j,r}\sum_{k',i'}\sigma'\left(\langle \mathbf{w}_{j,r,k}^{(t,s)}, y_{k,i}\boldsymbol{\mu}\rangle\right)5\log(2/\epsilon)\mathbb{1}\left(j = y_{k',i'}\right)\frac{\langle \boldsymbol{\mu}, j\boldsymbol{\xi}_{k',i'}\rangle}{\|\boldsymbol{\xi}_{k',i'}\|_2^2}$$

$$+ \frac{1}{m}\sum_{j,r}\sigma'\left(\langle \mathbf{w}_{j,r,k}^{(t,s)}, y_{k,i}\boldsymbol{\mu}\rangle\right)\langle \boldsymbol{\mu}, j\mathbf{w}_{j,r}^{(0)}\rangle + \frac{1}{m}\sum_{j,r}\sigma'\left(\langle \mathbf{w}_{j,r,k}^{(t,s)}, \boldsymbol{\xi}_{k,i}\rangle\right)\langle y_{k,i}\boldsymbol{\xi}_{k,i}, j\mathbf{w}_{j,r}^{(0)}\rangle$$

$$\geq \underbrace{\frac{1}{m}\sum_{j=y_{k,i},r}\sigma'\left(\langle \mathbf{w}_{j,r,k}^{(t,s)}, \boldsymbol{\xi}_{k,i}\rangle\right)5\log(2\tau/\epsilon)}_{I_1}$$

$$\underbrace{-\frac{1}{m}\sum_{j,r}\sum_{(k',i')\neq(k,i)}\sigma'\left(\langle \mathbf{w}_{j,r,k}^{(t,s)}, \boldsymbol{\xi}_{k,i}\rangle\right)5\log(2\tau/\epsilon)\frac{|\langle \boldsymbol{\xi}_{k,i}, \boldsymbol{\xi}_{k',i'}\rangle|}{\|\boldsymbol{\xi}_{k',i'}\|_2^2}}_{I_2}$$

$$\underbrace{-\frac{1}{m}\sum_{j,r}\sum_{k',i'}\sigma'\left(\langle \mathbf{w}_{j,r,k}^{(t,s)}, y_{k,i}\boldsymbol{\mu}\rangle\right)5\log(2\tau/\epsilon)\frac{|\langle \boldsymbol{\mu}, \boldsymbol{\xi}_{k',i'}\rangle|}{\|\boldsymbol{\xi}_{k',i'}\|_2^2}}_{I_3}$$

$$\underbrace{-\frac{1}{m}\sum_{j,r}\sigma'\left(\langle \mathbf{w}_{j,r,k}^{(t,s)}, y_{k,i}\boldsymbol{\mu}\rangle\right)\left|\langle \boldsymbol{\mu}, j\mathbf{w}_{j,r}^{(0)}\rangle\right|}_{I_4} \underbrace{-\frac{1}{m}\sum_{j,r}\sigma'\left(\langle \mathbf{w}_{j,r,k}^{(t,s)}, \boldsymbol{\xi}_{k,i}\rangle\right)\left|\langle y_{k,i}\boldsymbol{\xi}_{k,i}, j\mathbf{w}_{j,r}^{(0)}\rangle\right|}_{I_5}.$$

Now noting that $\sigma'(z) \leq 1$ and $\langle \boldsymbol{\mu}, \boldsymbol{\xi}_{k,i}\rangle = 0$ $\forall k \in [K], i \in [N]$ we have the following bounds for $I_2, I_3, I_4, I_5$ using Lemma 5, Lemma 6 and Lemma 10.

$$I_2 = \log(2\tau/\epsilon)\mathcal{O}\left(n\sqrt{\log(n^2/\delta)}/\sqrt{d}\right), I_3 = 0,$$

$$I_4 = \mathcal{O}\left(\sqrt{\log(m/\delta)}\cdot \sigma_0\|\boldsymbol{\mu}\|_2\right), I_5 = \mathcal{O}\left(\sqrt{\log(mn/\delta)}\cdot \sigma_0\sigma_p\sqrt{d}\right).$$

For $I_1$ we know that, $\langle \mathbf{w}_{y_{k,i},r^*,k}^{(t,s)}, \boldsymbol{\xi}_{k,i}\rangle \geq 0$ $\forall t \in [0 : T^* - 1], \forall s \in [0 : \tau - 1]$ (Lemma 21 ) and $r^*$ such that $r^* \in S_{k,i}^{(0)} := \left\{r \in [m] : \langle \mathbf{w}_{y_{k,i},r,k}^{(0)}, \boldsymbol{\xi}_{k,i}\rangle \geq 0\right\}$. Thus,

$$I_1 \geq \frac{1}{m}|S_{k,i}^{(0)}|5\log(2\tau/\epsilon) \geq 2\log(2\tau/\epsilon).$$

where the last inequality follows from Lemma 7. Applying triangle inequality we have,

$$y_{k,i}\langle \nabla f(\mathbf{W}_k^{(t,s)}, \mathbf{x}_{k,i}), \mathbf{W}^* \rangle \geq I_1 - |I_2| - |I_3| - |I_4| - |I_5| \geq \log(2\tau/\epsilon),$$

where the last inequality follows from Assumption 1 and Assumption 4. $\qquad\square$

**Lemma 24.** *(Lemma D.4 in Kou et al. (2023)) Under assumptions, for $0 \leq t \leq T^*$ and $0 \leq s \leq \tau - 1$, the following result holds,*

$$\left\|\nabla L_k(\mathbf{W}_k^{(t,s)})\right\|_2^2 \leq \mathcal{O}\left(\max\left\{\|\boldsymbol{\mu}\|_2^2, \sigma_p^2 d\right\}\right)L_k(\mathbf{W}_k^{(t,s)}).$$

**Lemma 25.** *For all $k \in [K]$, $T_1 \leq t \leq T^* - 1$, $0 \leq s \leq \tau - 1$ we have,*

$$\langle \nabla L_k(\mathbf{W}_k^{(t,s)}), \mathbf{W}_k^{(t,s)} - \mathbf{W}^* \rangle \geq L_k(\mathbf{W}_k^{(t,s)}) - \frac{\epsilon}{2\tau}.$$

*Proof.*

$$\langle \nabla L_k(\mathbf{W}_k^{(t,s)}), \mathbf{W}_k^{(t,s)} - \mathbf{W}^* \rangle$$

$$= \frac{1}{N} \sum_{i \in [N]} \ell'^{(t,s)}_{k,i} \langle y_{k,i} \nabla f(\mathbf{W}_k^{(t,s)}, \mathbf{x}_{k,i}), \mathbf{W}_k^{(t,s)} - \mathbf{W}^* \rangle$$

$$\overset{(a)}{=} \frac{1}{N} \sum_{i \in [N]} \ell'^{(t,s)}_{k,i} \left[ y_{k,i} f(\mathbf{W}_k^{(t,s)}, \mathbf{x}) - y_{k,i} \langle \nabla f(\mathbf{W}_k^{(t,s)}, \mathbf{x}_{k,i}), \mathbf{W}^* \rangle \right]$$

$$\overset{(b)}{\geq} \frac{1}{N} \sum_{i \in [N]} \ell'^{(t,s)}_{k,i} \left[ y_{k,i} f(\mathbf{W}_k^{(t,s)}, \mathbf{x}_{k,i}) - \log(2\tau/\epsilon) \right]$$

$$\overset{(c)}{\geq} \frac{1}{N} \sum_{i \in [N]} \left[ \ell(y_{k,i} f(\mathbf{W}_k^{(t,s)}, \mathbf{x}_{k,i})) - \epsilon/2\tau \right]$$

$$= L_k(\mathbf{W}_k^{(t,s)}) - \frac{\epsilon}{2\tau}.$$

Here $(a)$ follows from the property that $\langle \nabla f(\mathbf{W}, \mathbf{x}), \mathbf{W} \rangle = f(\mathbf{W}, \mathbf{x})$ for our two-layer CNN model; $(b)$ follows from equation 23 (note that $\ell'^{(t,s)}_{k,i} \leq 0$), $(c)$ follows from $\ell'(z)(z - z') \geq \ell(z) - \ell(z')$ since $\ell(\cdot)$ is convex and $\log(1 + z) \leq z$. $\qquad \square$

**Lemma 26.** *(Local Model Convergence)  Under assumptions, for all $t \geq T_1$ we have,*

$$\left\| \mathbf{W}_k^{(t,\tau)} - \mathbf{W}^* \right\|_2^2 \leq \left\| \mathbf{W}^{(t)} - \mathbf{W}^* \right\|_2^2 - \eta \sum_{s=0}^{\tau-1} L_k(\mathbf{W}_k^{(t,s)}) + \eta\epsilon.$$

*Proof.*

$$\left\| \mathbf{W}_k^{(t,s+1)} - \mathbf{W}^* \right\|_2^2$$

$$= \left\| \mathbf{W}_k^{(t,s)} - \mathbf{W}^* \right\|_2^2 - 2\eta \langle \nabla L_k(\mathbf{W}_k^{(t,s)}), \mathbf{W}_k^{(t,s)} - \mathbf{W}^* \rangle + \eta^2 \left\| \nabla L_k(\mathbf{W}_k^{(t,s)}) \right\|_2^2$$

$$\overset{(a)}{\leq} \left\| \mathbf{W}_k^{(t,s)} - \mathbf{W}^* \right\|_2^2 - 2\eta L_k(\mathbf{W}_k^{(t,s)}) + \frac{\eta\epsilon}{\tau} + \eta^2 \left\| \nabla L_k(\mathbf{W}_k^{(t,s)}) \right\|_2^2$$

$$\overset{(b)}{\leq} \left\| \mathbf{W}_k^{(t,s)} - \mathbf{W}^* \right\|_2^2 - \eta L_k(\mathbf{W}_k^{(t,s)}) + \frac{\eta\epsilon}{\tau},$$

where $(a)$ follows from Lemma 25; $(b)$ follows from Lemma 24 and Assumption 5. Now starting from $s = \tau - 1$ and unrolling the recursion we have,

$$\left\| \mathbf{W}_k^{(t,\tau)} - \mathbf{W}^* \right\|_2^2 \leq \left\| \mathbf{W}_k^{(t,0)} - \mathbf{W}^* \right\|_2^2 - \eta \sum_{s=0}^{\tau-1} L_k(\mathbf{W}_k^{(t,s)}) + \eta\epsilon.$$

$\qquad \square$

### C.5 Proof of Theorem 1

For any $t \geq T_1$ we have,

$$
\begin{aligned}
\left\|\mathbf{W}^{(t+1)} - \mathbf{W}^*\right\|_2^2 &= \left\|\sum_{k=1}^{K} \frac{1}{K} \mathbf{W}_k^{(t,\tau)} - \mathbf{W}^*\right\|_2^2 \\
&\overset{(a)}{\leq} \sum_{k=1}^{K} \frac{1}{K} \left\|\mathbf{W}_k^{(t,\tau)} - \mathbf{W}^*\right\|_2^2 \\
&\overset{(b)}{\leq} \left\|\mathbf{W}^{(t)} - \mathbf{W}^*\right\|_2^2 - \eta \frac{1}{K} \sum_{k=1}^{K} \sum_{s=0}^{\tau-1} L_k(\mathbf{W}_k^{(t,s)}) + \eta\epsilon \\
&\overset{(c)}{\leq} \left\|\mathbf{W}^{(t)} - \mathbf{W}^*\right\|_2^2 - \eta \frac{1}{K} \sum_{k=1}^{K} L_k(\mathbf{W}^{(t)}) + \eta\epsilon \\
&= \left\|\mathbf{W}^{(t)} - \mathbf{W}^*\right\|_2^2 - \eta L(\mathbf{W}^{(t)}) + \eta\epsilon,
\end{aligned}
\tag{61}
$$

where $(a)$ follows from Jensen's inequality, $(b)$ follows from Lemma 26; $(c)$ follows from $\sum_{s=0}^{\tau-1} L_k(\mathbf{W}_k^{(t,s)}) \leq L_k(\mathbf{W}_k^{(t,0)}) = L_k(\mathbf{W}^{(t)})$. From equation 61 we get,

$$
\eta L(\mathbf{W}^{(t)}) \leq \left\|\mathbf{W}^{(t)} - \mathbf{W}^*\right\|_2^2 - \left\|\mathbf{W}^{(t+1)} - \mathbf{W}^*\right\|_2^2 + \eta\epsilon.
$$

Summing over $t = T_1, T_1 + 1, \ldots, T$ and dividing by $\eta(T - T_1 + 1)$ we have,

$$
\frac{1}{T - T_1 + 1} \sum_{t=T_1}^{T} L(\mathbf{W}^{(t)}) \leq \frac{\left\|\mathbf{W}^{(T_1)} - \mathbf{W}^*\right\|_2^2}{\eta(T - T_1 + 1)} + \epsilon,
\tag{62}
$$

for all $T_1 \leq T \leq T^* - 1$. Now equation 62 implies that we can find an iterate with training error less than $2\epsilon$ within,

$$
T = T_1 + \frac{\left\|\mathbf{W}^{(t)} - \mathbf{W}^*\right\|_2^2}{\eta\epsilon} = \mathcal{O}\left(\frac{mn}{\eta\sigma_p^2 d\tau}\right) + \mathcal{O}\left(\frac{mn \log(\tau/\epsilon)}{\eta\sigma_p^2 d\epsilon}\right)
$$

rounds where the last equality follows from the definition of $T_1$ in equation 56 and Lemma 22. This completes our proof of Theorem 1.

$\square$

# D    Proof of Theorem 2

We first state some intermediate lemmas that will be used in the proof.

**Lemma 27.** *Suppose* $\langle \mathbf{w}_{j,r}^{(t')}, j\boldsymbol{\mu} \rangle \geq 0$ *for some* $t' \geq 0$. *Then for all* $t \geq t', s \in [0 : \tau - 1], k \in [K]$, *we have* $\langle \mathbf{w}_{j,r,k}^{(t,s)}, j\boldsymbol{\mu} \rangle \geq 0$.

*Proof.* We will use a proof by induction. We will show that our claim holds for $t = t', s \in [0 : \tau - 1]$ and also $t = (t' + 1), s = 0$. Using this fact we can argue that the claim holds for all $t \geq t'$ and $s \in [0 : \tau - 1]$.

**Case 1:** First let us look at the local iterations $s \in [0 : \tau - 1]$ for $t = t'$. From Lemma 4 we have,

$$\langle \mathbf{w}_{j,r,k}^{(t',s)}, j\boldsymbol{\mu} \rangle = \langle \mathbf{w}_{j,r}^{(t')}, j\boldsymbol{\mu} \rangle + \gamma_{j,r,k}^{(t',s)}$$
$$\overset{(a)}{\geq} \langle \mathbf{w}_{y,r}^{(t')}, j\boldsymbol{\mu} \rangle$$
$$\overset{(b)}{\geq} 0,$$

where $(a)$ uses $\gamma_{j,r,k}^{(\cdot,\cdot)} \geq 0$ by definition; $(b)$ uses $\langle \mathbf{w}_{j,r}^{(t')}, j\boldsymbol{\mu} \rangle \geq 0$.

**Case 2:** Now let us look at the round update $t = t' + 1, s = 0$. We have,

$$\langle \mathbf{w}_{j,r,k}^{(t'+1,0)}, j\boldsymbol{\mu} \rangle = \langle \mathbf{w}_{j,r}^{(t'+1)}, j\boldsymbol{\mu} \rangle$$
$$= \langle \mathbf{w}_{j,r}^{(t')}, j\boldsymbol{\mu} \rangle + \frac{1}{K} \sum_{i=1}^{K} \gamma_{j,r,k}^{(t',\tau)}$$
$$\overset{(a)}{\geq} \langle \mathbf{w}_{j,r}^{(t')}, j\boldsymbol{\mu} \rangle$$
$$\overset{(b)}{\geq} 0,$$

where $(a)$ uses $\gamma_{j,r,k}^{(\cdot,\cdot)} \geq 0$ by definition; $(b)$ uses $\langle \mathbf{w}_{j,r}^{(t')}, j\boldsymbol{\mu} \rangle \geq 0$. $\qquad\square$

**Lemma 28.** *Under Condition 1, for any* $0 \leq t \leq T^* - 1$ *we have,*

$$\Gamma_{j,r}^{(t)} \geq \Gamma_{j,r}^{(t-1)} + \frac{\eta \|\boldsymbol{\mu}\|_2^2}{4m} \sum_{s=0}^{\tau-1} \min_{k,i} \left| \ell_{k,i}'^{(t-1,s)} \right| \qquad \textit{if } \langle \mathbf{w}_{j,r}^{(t-1)}, j\boldsymbol{\mu} \rangle \geq 0, \tag{63}$$

*and,*

$$\Gamma_{j,r}^{(t)} \geq \Gamma_{j,r}^{(t-1)} + \frac{\eta \|\boldsymbol{\mu}\|_2^2}{4m} \left( \min_{k,i} \left| \ell_{k,i}'^{(t-1,0)} \right| + h \sum_{s=1}^{\tau-1} \min_{k,i} \left| \ell_{k,i}'^{(t-1,s)} \right| \right) \qquad \textit{if } \langle \mathbf{w}_{j,r}^{(0)}, j\boldsymbol{\mu} \rangle < 0. \tag{64}$$

**Proof.**

From equation 18 we have the following update equation for $\Gamma_{j,r}^{(t)}$,

$$\Gamma_{j,r}^{(t)} = \Gamma_{j,r}^{(t-1)} - \frac{\eta}{nm} \sum_{s=0}^{\tau-1} \sum_{k,i} \ell_{k,i}'^{(t-1,s)} \cdot \sigma' \left( \langle \mathbf{w}_{j,r,k}^{(t-1,s)}, y_{k,i}\boldsymbol{\mu} \rangle \right) \cdot \|\boldsymbol{\mu}\|_2^2. \tag{65}$$

*Proof of equation 63.* In this case we know from Lemma 27 that if $\langle \mathbf{w}_{j,r}^{(t)}, j\boldsymbol{\mu} \rangle \geq 0$, then

$$\langle \mathbf{w}_{j,r,k}^{(t,s)}, j\boldsymbol{\mu} \rangle \geq 0 \text{ for all } k \in [K], s \in [0 : \tau - 1]. \tag{66}$$

Using this observation we have from equation 65,

$$\Gamma_{j,r}^{(t)} \overset{(a)}{\geq} \Gamma_{j,r}^{(t-1)} + \frac{\eta|D_j|\,\|\boldsymbol{\mu}\|_2^2}{nm} \sum_{s=0}^{\tau-1} \min_{(k,i)\in D_j} \left|\ell_{k,i}'^{(t-1,s)}\right|$$

$$\overset{(b)}{\geq} \Gamma_{j,r}^{(t-1)} + \frac{\eta\,\|\boldsymbol{\mu}\|_2^2}{4m} \sum_{s=0}^{\tau-1} \min_{k,i} \left|\ell_{k,i}'^{(t-1,s)}\right| \tag{67}$$

where $(a)$ follows from the definition of $D_j := \{k \in [K], i \in [N] : y_{k,i} = j\}$; $(b)$ follows from Lemma 9 and $\min_{(k,i)\in D_j} \left|\ell_{k,i}'^{(t',s)}\right| \geq \min_{k,i} \left|\ell_{k,i}'^{(t',s)}\right|$. $\qquad\square$

*Proof of equation 64.* First let us look at the iteration $s = 0$. In this case we know that $\langle \mathbf{w}_{j,r,k}^{(t-1,0)}, j\boldsymbol{\mu}\rangle = \langle \mathbf{w}_{j,r}^{(t-1)}, j\boldsymbol{\mu}\rangle < 0$ and thus $\langle \mathbf{w}_{j,r}^{(t-1)}, y_{k,i}\boldsymbol{\mu}\rangle > 0$ for $y_{k,i} = -j$. Using this observation we have,

$$-\frac{\eta}{nm} \sum_{k,i} \ell_{k,i}'^{(t-1,0)} \cdot \sigma'\left(\langle \mathbf{w}_{j,r,k}^{(t-1,0)}, y_{k,i}\boldsymbol{\mu}\rangle\right) \cdot \|\boldsymbol{\mu}\|_2^2 \geq \frac{\eta\,|D_{-j}|\,\|\boldsymbol{\mu}\|_2^2}{nm} \min_{(k,i)\in D_{-j}} \left|\ell_{k,i}'^{(t-1,0)}\right|$$

$$\overset{(a)}{\geq} \frac{\eta\,\|\boldsymbol{\mu}\|_2^2}{4m} \min_{k,i} \left|\ell_{k,i}'^{(t-1,0)}\right|$$

where $(a)$ follows from Lemma 9 and $\min_{(k,i)\in D_j} \left|\ell_{k,i}'^{(t',s)}\right| \geq \min_{k,i} \left|\ell_{k,i}'^{(t',s)}\right|$.

Now let us look at the case $1 \leq s \leq \tau - 1$. In this case if $\langle \mathbf{w}_{j,r,k}^{(t-1,s)}, j\boldsymbol{\mu}\rangle < 0$ then,

$$-\frac{\eta}{nm} \sum_i \ell_{k,i}'^{(t-1,s)} \cdot \sigma'\left(\langle \mathbf{w}_{j,r,k}^{(t-1,s)}, y_{k,i}\boldsymbol{\mu}\rangle\right) \cdot \|\boldsymbol{\mu}\|_2^2 \geq \frac{\eta\,|D_{-j,k}|\,\|\boldsymbol{\mu}\|_2^2}{nm} \min_{(k,i)\in D_{-j,k}} \left|\ell_{k,i}'^{(t-1,s)}\right|, \tag{68}$$

and if $\langle \mathbf{w}_{j,r,k}^{(t-1,s)}, j\boldsymbol{\mu}\rangle \geq 0$ then,

$$-\frac{\eta}{nm} \sum_i \ell_{k,i}'^{(t-1,s)} \cdot \sigma'\left(\langle \mathbf{w}_{j,r,k}^{(t-1,s)}, y_{k,i}\boldsymbol{\mu}\rangle\right) \cdot \|\boldsymbol{\mu}\|_2^2 \geq \frac{\eta\,|D_{j,k}|\,\|\boldsymbol{\mu}\|_2^2}{nm} \min_{(k,i)\in D_{j,k}} \left|\ell_{k,i}'^{(t-1,s)}\right|.$$

Thus,

$$-\frac{\eta}{nm} \sum_i \ell_{k,i}'^{(t-1,s)} \cdot \sigma'\left(\langle \mathbf{w}_{j,r,k}^{(t-1,s)}, y_{k,i}\boldsymbol{\mu}\rangle\right) \cdot \|\boldsymbol{\mu}\|_2^2 \geq \frac{\eta\min\{|D_{+,k}|,|D_{-,k}|\}\,\|\boldsymbol{\mu}\|_2^2}{nm} \min_{(k,i)\in D_k} \left|\ell_{k,i}'^{(t-1,s)}\right|. \tag{69}$$

Using the results in equation 68 and equation 69 we have,

$$\Gamma_{j,r}^{(t)} \geq \Gamma_{j,r}^{(t-1)} + \frac{\eta\,\|\boldsymbol{\mu}\|_2^2}{4m} \min_{k,i} \left|\ell_{k,i}'^{(t-1,0)}\right| + \frac{\eta\,\|\boldsymbol{\mu}\|_2^2}{m} \sum_k \frac{\min\{|D_{+,k}|,|D_{-,k}|\}}{n} \sum_{s=1}^{\tau-1} \min_{(k,i)} \left|\ell_{k,i}'^{(t-1,s)}\right|$$

$$\overset{(a)}{\geq} \Gamma_{j,r}^{(t-1)} + \frac{\eta\,\|\boldsymbol{\mu}\|_2^2}{4m} \left(\min_{k,i} \left|\ell_{k,i}'^{(t-1,0)}\right| + h \sum_{s=1}^{\tau-1} \min_{k,i} \left|\ell_{k,i}'^{(t-1,s)}\right|\right),$$

where $(a)$ follows from our definition of $h$ in equation 1. $\qquad\square$

**Lemma 29.** *Let $A_j := \{r \in [m] : \langle \mathbf{w}_{j,r}^{(0)}, j\boldsymbol{\mu}\rangle \geq 0\}$. For any $0 \leq t \leq T^* - 1$ we have,*

1. *For any $j \in \{\pm 1\}, r \in [m] : \Gamma_{j,r}^{(t)} \leq \frac{\eta\|\boldsymbol{\mu}\|_2^2}{m} \sum_{t'=0}^{t-1} \sum_{s=0}^{\tau-1} \max_{k,i} \left|\ell_{k,i}'^{(t',s)}\right|.$*

2. *For any $r \in A_j : \Gamma_{j,r}^{(t)} \geq \frac{\eta\|\boldsymbol{\mu}\|_2^2}{4m} \sum_{t'=0}^{t-1} \sum_{s=0}^{\tau-1} \min_{(k,i)} \left|\ell_{k,i}'^{(t',s)}\right|.$*

3. *For any $r \notin A_j : \Gamma_{j,r}^{(t)} \geq \frac{\eta \|\boldsymbol{\mu}\|_2^2}{4m} \sum_{t'=0}^{t-1} \left( \min_{k,i} \left| \ell'_{k,i}^{(t',0)} \right| + h \sum_{s=1}^{\tau-1} \min_{k,i} \left| \ell'_{k,i}^{(t',s)} \right| \right).$*

$\square$

**Proof.**

Unrolling the iterative update in equation 18 we have,

$$\Gamma_{j,r}^{(t)} = \frac{\eta}{nm} \sum_{t'=0}^{t-1} \sum_{s=0}^{\tau-1} \sum_{k,i} (-\ell'_{k,i}^{(t',s)}) \cdot \sigma' \left( \langle \mathbf{w}_{j,r,k}^{(t',s)}, y_{k,i} \boldsymbol{\mu} \rangle \right) \cdot \|\boldsymbol{\mu}\|_2^2 . \tag{70}$$

*Proof of equation 1.* Using equation 70, we can get an upper bound on $\Gamma_{j,r}^{(t)}$ as follows.

$$\Gamma_{j,r}^{(t)} \leq \frac{\eta \|\boldsymbol{\mu}\|_2^2}{m} \sum_{t'=0}^{t-1} \sum_{s=0}^{\tau-1} \max_{k,i} \left| \ell'_{k,i}^{(t',s)} \right| ,$$

where the inequality follows from $\sigma'(\cdot) \leq 1$.

*Proof of equation 2.* From Lemma 27 we know that if $\langle \mathbf{w}_{j,r}^{(0)}, j\boldsymbol{\mu} \rangle \geq 0$ then $\langle \mathbf{w}_{j,r}^{(t')}, j\boldsymbol{\mu} \rangle \geq 0$ for all $t' \geq 0$. Thus using equation 63 repeatedly for all $0 \leq t' \leq t-1$ we get,

$$\Gamma_{j,r}^{(t)} \geq \frac{\eta \|\boldsymbol{\mu}\|_2^2}{4m} \sum_{t'=0}^{t-1} \sum_{s=0}^{\tau-1} \min_{k,i} \left| \ell'_{k,i}^{(t',s)} \right| .$$

*Proof of equation 3.* Note that the bound in equation 64 holds even if $\langle \mathbf{w}_{j,r}^{(t-1)}, j\boldsymbol{\mu} \rangle \geq 0$. Thus applying equation 64 repeatedly for all $0 \leq t' \leq t-1$ we get,

$$\Gamma_{j,r}^{(t)} \geq \frac{\eta \|\boldsymbol{\mu}\|_2^2}{4m} \sum_{t'=0}^{t-1} \left( \min_{k,i} \left| \ell'_{k,i}^{(t',0)} \right| + h \sum_{s=1}^{\tau-1} \min_{k,i} \left| \ell'_{k,i}^{(t',s)} \right| \right) .$$

**Lemma 30.** *Under assumptions, for any $0 \leq t \leq T^* - 1$ we have,*

1. $\sum_{k,i} \overline{P}_{j,r,k,i}^{(t)} \leq \frac{3\eta \sigma_p^2 d}{2m} \sum_{t'=0}^{t-1} \sum_{s=0}^{\tau-1} \max_{k,i} \left| \ell'_{k,i}^{(t',s)} \right| .$

2. $\sum_{k,i} \overline{P}_{j,r,k,i}^{(t)} \geq \frac{\eta \sigma_p^2 d}{16m} \sum_{t'=0}^{t-1} \sum_{s=0}^{\tau-1} \min_{(k,i) \in \tilde{S}_{j,r}^{(t',s)}} \left| \ell'_{k,i}^{(t',s)} \right|$

*where $\tilde{S}_{j,r}^{(t',s)} := \left\{ k \in [K], i \in [N] : \langle \mathbf{w}_{j,r,k}^{(t',s)}, \boldsymbol{\xi}_{k,i} \rangle \geq 0 \right\}$.*

**Proof.**

From equation 20 we have the following update equation for $\overline{P}_{j,r,k,i}^{(t)}$.

$$\sum_{k,i} \overline{P}_{j,r,k,i}^{(t)} = \sum_{k,i} \overline{P}_{j,r,k,i}^{(t-1)} - \frac{\eta}{nm} \sum_{s=0}^{\tau-1} \sum_{k,i:y_{k,i}=j} \ell'_{k,i}^{(t-1,s)} \cdot \sigma' \left( \langle \mathbf{w}_{j,r,k}^{(t-1,s)}, \boldsymbol{\xi}_{k,i} \rangle \right) \cdot \|\boldsymbol{\xi}_{k,i}\|_2^2$$

$$= \sum_{k,i} \overline{P}_{j,r,k,i}^{(t-1)} - \frac{\eta}{nm} \sum_{s=0}^{\tau-1} \sum_{(k,i) \in \tilde{S}_{j,r}^{(t-1,s)}} \ell'_{k,i}^{(t-1,s)} \cdot \|\boldsymbol{\xi}_{k,i}\|_2^2 . \tag{71}$$

where the last equality follows from the definition of $\tilde{S}_{j,r}^{(t,s)}$.

*Proof of equation 1.* Now using equation 71 we have,

$$\sum_{k,i} \overline{P}_{j,r,k,i}^{(t)} \overset{(a)}{\leq} \sum_{k,i} \overline{P}_{j,r,k,i}^{(t-1)} + \frac{3\eta\sigma_p^2 d}{2m} \sum_{s=0}^{\tau-1} \max_{k,i} \left| \ell'^{(t-1,s)}_{k,i} \right|$$

where $(a)$ follows from Lemma 5. Unrolling the recursion above we have the following upper bound,

$$\sum_{k,i} \overline{P}_{j,r,k,i}^{(t)} \leq \frac{3\eta\sigma_p^2 d}{2m} \sum_{t'=0}^{t-1} \sum_{s=0}^{\tau-1} \max_{k,i} \left| \ell'^{(t',s)}_{k,i} \right|.$$

*Proof of equation 2.* From equation 71 we have,

$$\sum_{k,i} \overline{P}_{j,r,k,i}^{(t)} \overset{(a)}{\geq} \sum_{k,i} \overline{P}_{j,r,k,i}^{(t-1)} + \frac{\eta\sigma_p^2 d}{16m} \sum_{s=0}^{\tau-1} \min_{(k,i) \in \tilde{S}_{j,r}^{(t-1,s)}} \left| \ell'^{(t-1,s)}_{k,i} \right|$$

where $(a)$ follows from Lemma 5 and Proposition 2 part 5 which implies $\left| \tilde{S}_{j,r}^{(t-1,s)} \right| \geq n/8$. Unrolling the recursion above we have,

$$\sum_{k,i} \overline{P}_{j,r,k,i}^{(t)} \geq \frac{\eta\sigma_p^2 d}{16m} \sum_{t'=0}^{t-1} \sum_{s=0}^{\tau-1} \min_{(k,i) \in \tilde{S}_{j,r}^{(t',s)}} \left| \ell'^{(t',s)}_{k,i} \right|.$$

$\square$

**Lemma 31.** *For all $t \geq T_1$, we have $\langle \mathbf{w}_{y,r}^{(t)}, y\boldsymbol{\mu} \rangle > 0$.*

*Proof.* We have,

$$
\begin{aligned}
\langle \mathbf{w}_{y,r}^{(t)}, y\boldsymbol{\mu} \rangle &= \langle \mathbf{w}_{y,r}^{(0)}, y\boldsymbol{\mu} \rangle + \Gamma_{j,r}^{(t)} \\
&\overset{(a)}{\geq} -\Theta\left( \sqrt{\log(m/\delta)} \cdot \sigma_0 \left\| \boldsymbol{\mu} \right\|_2 \right) + \Gamma_{j,r}^{(t)} \\
&\overset{(b)}{\geq} -\Theta\left( \sqrt{\log(m/\delta)} \cdot \sigma_0 \left\| \boldsymbol{\mu} \right\|_2 \right) + \frac{\eta \left\| \boldsymbol{\mu} \right\|_2^2}{4m} \sum_{t'=0}^{T_1-1} \min_{k,i} \left| \ell'^{(t',0)}_{k,i} \right| \\
&\overset{(c)}{=} -\Theta\left( \sqrt{\log(m/\delta)} \cdot \sigma_0 \left\| \boldsymbol{\mu} \right\|_2 \right) + \Omega\left( \frac{n \left\| \boldsymbol{\mu} \right\|_2^2}{\sigma_p^2 d\tau} \right) \\
&\overset{(d)}{\geq} \Theta\left( \sqrt{\log(m/\delta)} \cdot \frac{\sqrt{n} \left\| \boldsymbol{\mu} \right\|_2}{\sigma_p d\tau} \right) + \Omega\left( \frac{n \left\| \boldsymbol{\mu} \right\|_2^2}{\sigma_p^2 d\tau} \right) \\
&\overset{(e)}{\geq} 0.
\end{aligned}
\tag{72}
$$

Here $(a)$ follows from Lemma 6; $(b)$ follows from Lemma 29; $(c)$ follows from the definition of $T_1$ in Equation (56); $(d)$ follows from Assumption 4; $(e)$ follows from Assumption 3 and Assumption 2. $\square$

**Lemma 32.** *Under Condition 1, for any $T_1 \leq t \leq T^* - 1$ we have,*

1. $\dfrac{\left\| \mathbf{w}_{j,r}^{(0)} \right\|_2}{\Theta\left( \sigma_p^{-1} d^{-1/2} n^{-1/2} \right) \sum_{k,i} \overline{P}_{j,r,k,i}^{(t)}} = \mathcal{O}(1)$

2. $\dfrac{\Gamma_{j,r}^{(t)} \left\| \boldsymbol{\mu} \right\|_2^{-1}}{\Theta\left( \sigma_p^{-1} d^{-1/2} n^{-1/2} \right) \sum_{k,i} \overline{P}_{j,r,k,i}^{(t)}} = \mathcal{O}(1)$

*Proof of equation 1.* Note from our proof of Lemma 21, we know that for all $T_1 \le t \le T^* - 1$ we have $\overline{P}_{j,r,k^*,i^*}^{(t)} \ge 2$ for all $(k^*, i^*) \in \tilde{S}_{j,r}^{(0)} = \left\{ k \in [K], i \in [N] : y_{k,i} = j, \langle \mathbf{w}_{j,r,k}^{(0)}, \boldsymbol{\xi}_{k,i} \rangle \ge 0 \right\}$. Thus,

$$\sum_{k,i} \overline{P}_{j,r,k,i}^{(t)} \ge 2 \left| \tilde{S}_{j,r}^{(0)} \right| \overset{(a)}{=} \Omega(n), \tag{73}$$

where $(a)$ follows from Lemma 8. This implies,

$$\frac{\left\| \mathbf{w}_{j,r}^{(0)} \right\|_2}{\Theta\left( \sigma_p^{-1} d^{-1/2} n^{-1/2} \right) \sum_{k,i} \overline{P}_{j,r,k,i}^{(t)}} \overset{(a)}{=} \frac{\Theta\left( \sigma_0 \sqrt{d} \right)}{\Theta\left( \sigma_p^{-1} d^{-1/2} n^{-1/2} \right) \sum_{k,i} \overline{P}_{j,r,k,i}^{(t)}}$$

$$\overset{(b)}{=} \mathcal{O}\left( \sigma_0 \sigma_p d n^{-1/2} \right)$$

$$\overset{(c)}{=} \mathcal{O}(1).$$

Here $(a)$ follows from Lemma 6; $(b)$ follows from equation 73; $(c)$ follows from Assumption 4. $\qquad \square$

*Proof of equation 2.* From Lemma 28 and Lemma 30 we have,

$$\frac{\Gamma_{j,r}^{(t)}}{\sum_{k,i} \overline{P}_{j,r,k,i}^{(t)}} \le \frac{16 \|\boldsymbol{\mu}\|_2^2}{\sigma_p^2 d} \frac{\sum_{t'=0}^{t-1} \sum_{s=0}^{\tau-1} \max_{k,i} \left| \ell_{k,i}^{\prime(t',s)} \right|}{\sum_{t'=0}^{t-1} \sum_{s=0}^{\tau-1} \min_{(k,i) \in \tilde{S}_{j,r}^{(t',s)}} \left| \ell_{k,i}^{\prime(t',s)} \right|} \overset{(a)}{\le} \frac{16 C_2 \|\boldsymbol{\mu}\|_2^2}{\sigma_p^2 d},$$

where $(a)$ follows from Proposition 2 part 3 which implies $\max_{k,i} \left| \ell_{k,i}^{\prime(t'-1,s)} \right| \le C_2 \min_{(k,i) \in \tilde{S}_{j,r}^{(t'-1,s)}} \left| \ell_{k,i}^{\prime(t'-1,s)} \right|$ for all $0 \le t' \le T^* - 1, 0 \le s \le \tau - 1$. Thus,

$$\frac{\Gamma_{j,r}^{(t)} \|\boldsymbol{\mu}\|_2^{-1}}{\Theta\left( \sigma_p^{-1} d^{-1/2} n^{-1/2} \right) \sum_{k,i} \overline{P}_{j,r,k,i}^{(t)}} = \mathcal{O}\left( \frac{n^{1/2} \|\boldsymbol{\mu}\|_2}{\sigma_p d^{1/2}} \right) \overset{(a)}{=} \mathcal{O}(1).$$

where $(a)$ follows from Assumption 1. $\qquad \square$

**Lemma 33.** *For any $T_1 \le t \le T^* - 1$ we have,*

$$\frac{\sum_r \sigma\left( \langle \mathbf{w}_{y,r}^{(t)}, y\boldsymbol{\mu} \rangle \right)}{\sum_{r,k,i} \overline{P}_{-y,r,k,i}^{(t)}} \ge \frac{C_4 \|\boldsymbol{\mu}\|_2^2}{\sigma_p^2 md} \left( |A_y| + (m - |A_y|) \left( h + \frac{1}{\tau}(1-h) \right) \right),$$

*where $C_4 > 0$ is some constant.*

**Proof.**

We can write,

$$\sum_r \sigma\left( \langle \mathbf{w}_{y,r}^{(t)}, y\boldsymbol{\mu} \rangle \right) = \underbrace{\sum_{r : \langle \mathbf{w}_{y,r}^{(0)}, y\boldsymbol{\mu} \rangle \ge 0} \sigma\left( \langle \mathbf{w}_{y,r}^{(t)}, y\boldsymbol{\mu} \rangle \right)}_{I_1} + \underbrace{\sum_{r : \langle \mathbf{w}_{y,r}^{(0)}, y\boldsymbol{\mu} \rangle < 0} \sigma\left( \langle \mathbf{w}_{y,r}^{(t)}, y\boldsymbol{\mu} \rangle \right)}_{I_2}. \tag{74}$$

First note that if $\langle \mathbf{w}_{y,r}^{(0)}, y\boldsymbol{\mu} \rangle \ge 0$ then from Lemma 27 we know that ,

$$\langle \mathbf{w}_{y,r,k}^{(t,s)}, y\boldsymbol{\mu} \rangle \ge 0 \text{ for all } k \in [K], 0 \le t \le T^* - 1, 0 \le s \le \tau - 1. \tag{75}$$

We can bound $I_1$ as follows:

$$
\begin{aligned}
I_1 &= \sum_{r:\langle \mathbf{w}_{y,r}^{(0)}, y\boldsymbol{\mu}\rangle \geq 0} \sigma\left(\langle \mathbf{w}_{y,r}^{(t)}, y\boldsymbol{\mu}\rangle\right) \\
&\overset{(a)}{=} \sum_{r:\langle \mathbf{w}_{y,r}^{(0)}, y\boldsymbol{\mu}\rangle \geq 0} \langle \mathbf{w}_{y,r}^{(t)}, y\boldsymbol{\mu}\rangle \\
&\overset{(b)}{\geq} \sum_{r:\langle \mathbf{w}_{y,r}^{(0)}, y\boldsymbol{\mu}\rangle \geq 0} \Gamma_{y,r}^{(t)} \\
&\overset{(c)}{=} \Omega\left(|A_y|\eta\,\|\boldsymbol{\mu}\|_2^2 \sum_{t'=0}^{t-1}\sum_{s=0}^{\tau-1} \min_{k,i}\left|\ell'^{(t',s)}_{k,i}\right|\right).
\end{aligned}
\tag{76}
$$

Here $(a)$ follows from equation 75; $(b)$ follows from Lemma 4; $(c)$ follows from Lemma 29 part 2. For $I_2$, we have the following bound:

$$
\begin{aligned}
I_2 &= \sum_{r:\langle \mathbf{w}_{y,r}^{(0)}, y\boldsymbol{\mu}\rangle < 0} \sigma\left(\langle \mathbf{w}_{y,r}^{(t)}, y\boldsymbol{\mu}\rangle\right) \\
&\overset{(a)}{\geq} \sum_{r:\langle \mathbf{w}_{y,r}^{(0)}, y\boldsymbol{\mu}\rangle < 0} \langle \mathbf{w}_{y,r}^{(0)}, y\boldsymbol{\mu}\rangle + \Gamma_{j,r}^{(t)} \\
&\overset{(b)}{\geq} -(m-|A_y|)\Theta\left(\sqrt{\log(m/\delta)}\cdot\sigma_0\,\|\boldsymbol{\mu}\|_2\right) + \sum_{r:\langle \mathbf{w}_{y,r}^{(0)}, y\boldsymbol{\mu}\rangle < 0} \Gamma_{j,r}^{(t)} \\
&\overset{(c)}{=} \Omega\left(\sum_{r:\langle \mathbf{w}_{y,r}^{(0)}, y\boldsymbol{\mu}\rangle < 0} \Gamma_{j,r}^{(t)}\right) \\
&\overset{(d)}{\geq} \Omega\Bigg((m-|A_y|)\eta\,\|\boldsymbol{\mu}\|_2^2\left(\sum_{t'=0}^{T_1-1}\min_{k,i}\left|\ell'^{(t',0)}_{k,i}\right| + h\sum_{t'=0}^{T_1-1}\sum_{s=1}^{\tau-1}\min_{k,i}\left|\ell'^{(t',s)}_{k,i}\right|\right) \\
&\quad + (m-|A_y|)\eta\,\|\boldsymbol{\mu}\|_2^2 \sum_{t'=T_1}^{t-1}\sum_{s=0}^{\tau-1}\min_{k,i}\left|\ell'^{(t',s)}_{k,i}\right|\Bigg).
\end{aligned}
\tag{77}
$$

Here $(a)$ follows from $\sigma(z) \geq z$; $(b)$ follows from Lemma 6 and Assumption 4; $(c)$ follows from Lemma 31; $(d)$ follows from Lemma 29. Substituting equation 76 and equation 77 in equation 74 we have,

$$
\begin{aligned}
\sum_r \sigma\left(\langle \mathbf{w}_{y,r}^{(t)}, y\boldsymbol{\mu}\rangle\right) &\geq \Omega\Bigg(|A_y|\eta\,\|\boldsymbol{\mu}\|_2^2 \sum_{t'=0}^{t-1}\sum_{s=0}^{\tau-1}\min_{k,i}\left|\ell'^{(t',s)}_{k,i}\right| \\
&\quad + (m-|A_y|)\eta\,\|\boldsymbol{\mu}\|_2^2\left(\sum_{t'=0}^{T_1-1}\min_{k,i}\left|\ell'^{(t',0)}_{k,i}\right| + h\sum_{t'=0}^{T_1-1}\sum_{s=1}^{\tau-1}\min_{k,i}\left|\ell'^{(t',s)}_{k,i}\right|\right) \\
&\quad + (m-|A_y|)\eta\,\|\boldsymbol{\mu}\|_2^2 \sum_{t'=T_1}^{t-1}\sum_{s=0}^{\tau-1}\min_{k,i}\left|\ell'^{(t',s)}_{k,i}\right|\Bigg)
\end{aligned}
\tag{78}
$$

Now using equation 78 and Lemma 30 we have,

$$\frac{\sum_r \sigma\left(\langle \mathbf{w}_{y,r}^{(t)}, y\boldsymbol{\mu}\rangle\right)}{\sum_{r,k,i} \overline{P}_{-y,r,k,i}^{(t)}}$$

$$\overset{(a)}{\geq} \Omega\left(\frac{\|\boldsymbol{\mu}\|_2^2}{\sigma_p^2 md}\left(|A_y|\frac{\sum_{t'=0}^{t-1}\sum_{s=0}^{\tau-1}\min_{k,i}\left|\ell'^{(t',s)}_{k,i}\right|}{\sum_{t'=0}^{t-1}\sum_{s=0}^{\tau-1}\max_{k,i}\left|\ell'^{(t',s)}_{k,i}\right|}\right.\right.$$

$$\left.\left.+ (m-|A_y|)\frac{\sum_{t'=0}^{T_1-1}\left(\min_{k,i}\left|\ell'^{(t',0)}_{k,i}\right| + h\sum_{s=1}^{\tau-1}\min_{k,i}\left|\ell'^{(t',s)}_{k,i}\right|\right) + \sum_{t'=0}^{t-1}\sum_{s=0}^{\tau-1}\min_{k,i}\left|\ell'^{(t',s)}_{k,i}\right|}{\sum_{t'=0}^{T_1-1}\sum_{s=0}^{\tau-1}\max_{k,i}\left|\ell'^{(t',s)}_{k,i}\right| + \sum_{t'=T_1}^{t-1}\sum_{s=0}^{\tau-1}\max_{k,i}\left|\ell'^{(t',s)}_{k,i}\right|}\right)\right)$$

$$\overset{(b)}{\geq} \Omega\left(\frac{\|\boldsymbol{\mu}\|_2^2}{\sigma_p^2 md}\left(|A_y| + (m-|A_y|)\left(h + \frac{1}{\tau}(1-h)\right)\right)\right)$$

where $(a)$ follows from Lemma 30; $(b)$ follows from Proposition 2 part 3 and Equation (58).

$\square$

**Lemma 34.** *Under assumptions, for all $T_1 \leq t \leq T^* - 1$ we have*

$$\frac{\sum_r \sigma\left(\langle \mathbf{w}_{y,r}^{(t)}, y\boldsymbol{\mu}\rangle\right)}{\sigma_p \sum_{r=1}^m \left\|\mathbf{w}_{-y,r}^{(t)}\right\|_2} \geq \Theta\left(\frac{n^{1/2}\|\boldsymbol{\mu}\|_2^2}{\sigma_p^2 md^{1/2}}\left(|A_y| + (m-|A_y|)\left(h + \frac{1}{\tau}(1-h)\right)\right)\right).$$

*Proof.* To prove this, we first show that $\left\|\mathbf{w}_{j,r}^{(t)}\right\|_2 = \mathcal{O}\left(\sigma_p^{-1}d^{-1/2}n^{-1/2}\right) \cdot \sum_{k,i} \overline{P}_{j,r,k,i}^{(t)}$ for all $j \in \{\pm 1\}$.

We first bound the norm of the noise components as follows.

$$\left\|\sum_{k,i} P_{j,r,k,i}^{(t)} \cdot \|\boldsymbol{\xi}_{k,i}\|_2^{-2} \cdot \boldsymbol{\xi}_{k,i}\right\|_2^2$$

$$= \sum_{k,i}\left(P_{j,r,k,i}^{(t)}\right)^2 \cdot \|\boldsymbol{\xi}_{k,i}\|_2^{-2} + 2\sum_{k,k'>k,i,i'>i} P_{j,r,k,i}^{(t)}P_{j,r,k',i'}^{(t)} \cdot \|\boldsymbol{\xi}_{k,i}\|_2^{-2} \cdot \|\boldsymbol{\xi}_{k',i'}\|_2^{-2} \cdot \langle \boldsymbol{\xi}_{k,i}, \boldsymbol{\xi}_{k',i'}\rangle$$

$$\overset{(a)}{\leq} 4\sigma_p^{-2}d^{-1}\sum_{k,i}\left(P_{j,r,k,i}^{(t)}\right)^2 + 2\sum_{k,k'>k,i,i'>i}\left|P_{j,r,k,i}^{(t)}P_{j,r,k',i'}^{(t)}\right|(16\sigma_p^{-4}d^{-2})(2\sigma_p^2\sqrt{d\log(6n^2/\delta)})$$

$$= 4\sigma_p^{-2}d^{-1}\sum_{k,i}\left(P_{j,r,k,i}^{(t)}\right)^2 + 32\sigma_p^{-2}d^{-3/2}\left(\left(\sum_{k,i}\left|P_{j,r,k,i}^{(t)}\right|\right)^2 - \sum_{k,i}\left(P_{j,r,k,i}^{(t)}\right)^2\right)$$

$$= \Theta\left(\sigma_p^{-2}d^{-1}\right)\sum_{k,i}\left(P_{j,r,k,i}^{(t)}\right)^2 + \widetilde{\Theta}\left(\sigma_p^{-2}d^{-3/2}\right)\left(\sum_{k,i}\left|P_{j,r,k,i}^{(t)}\right|\right)^2$$

$$\overset{(b)}{\leq} \left[\Theta\left(\sigma_p^{-2}d^{-1}\right) + \widetilde{\Theta}\left(\sigma_p^{-2}d^{-3/2}\right)\right]\left(\sum_{k,i}\left|\overline{P}_{j,r,k,i}^{(t)}\right| + \sum_{k,i}\left|\underline{P}_{j,r,k,i}^{(t)}\right|\right)^2$$

$$= \Theta\left(\sigma_p^{-2}d^{-1}n^{-1}\right)\left(\sum_{k,i}\overline{P}_{j,r,k,i}^{(t)}\right)^2. \tag{79}$$

Here for $(a)$ uses Lemma 5; $(b)$ uses $\max_{j,r,k,i} \left| \underline{P}_{j,r,k,i}^{(t)} \right| \leq \beta + 8\sqrt{\frac{\log(6n^2/\delta)}{d}} n\alpha = \mathcal{O}(1)$ from Theorem 3 and so $\sum_{k,i} \left| \underline{P}_{j,r,k,i}^{(t)} \right| = \mathcal{O}\left( \sum_{k,i} \overline{P}_{j,r,k,i}^{(t)} \right)$. Now from equation 17 we know that,

$$\mathbf{w}_{j,r}^{(t)} = \mathbf{w}_{j,r}^{(0)} + j\Gamma_{j,r}^{(t)} \cdot \|\boldsymbol{\mu}\|_2^{-2} \boldsymbol{\mu} + \sum_{k=1}^{2} \sum_{i \in [N]} P_{j,r,k,i}^{(t)} \cdot \|\boldsymbol{\xi}_{k,i}\|_2^{-2} \cdot \boldsymbol{\xi}_{k,i}.$$

Using triangle inequality and equation 79 we have,

$$\left\| \mathbf{w}_{j,r}^{(t)} \right\|_2 \leq \left\| \mathbf{w}_{j,r}^{(0)} \right\|_2 + \Gamma_{j,r}^{(t)} \|\boldsymbol{\mu}\|_2^{-1} + \Theta\left( \sigma_p^{-1} d^{-1/2} n^{-1/2} \right) \sum_{k,i} \overline{P}_{j,r,k,i}^{(t)}$$

$$\overset{(a)}{=} \Theta\left( \sigma_p^{-1} d^{-1/2} n^{-1/2} \right) \sum_{k,i} \overline{P}_{j,r,k,i}^{(t)}$$

where $(a)$ follows from Lemma 32.

Thus,

$$\frac{\sum_r \sigma\left( \langle \mathbf{w}_{y,r}^{(t)}, y\boldsymbol{\mu} \rangle \right)}{\sigma_p \sum_{r=1}^{m} \left\| \mathbf{w}_{-y,r}^{(t)} \right\|_2} \geq \frac{\sum_r \sigma\left( \langle \mathbf{w}_{y,r}^{(t)}, y\boldsymbol{\mu} \rangle \right)}{\Theta\left( d^{-1/2} n^{-1/2} \right) \sum_{k,i} \overline{P}_{j,r,k,i}^{(t)}}$$

$$\overset{(a)}{=} \Theta\left( \frac{n^{1/2} \|\boldsymbol{\mu}\|_2^2}{\sigma_p^2 m d^{1/2}} \left( |A_y| + (m - |A_y|)\left( h + \frac{1}{\tau}(1 - h) \right) \right) \right)$$

where $(a)$ follows from Lemma 33. $\qquad\square$

**Lemma 35.** *(sub-result in Theorem E.1 in Cao et al. (2022).)* Denote $g(\boldsymbol{\xi}) = \sum_r \sigma\left( \langle \mathbf{w}_{-y,r}^{(t)}, \boldsymbol{\xi} \rangle \right)$. Then for any $x \geq 0$ it holds that

$$\Pr(g(\boldsymbol{\xi}) - \mathbb{E}g(\boldsymbol{\xi}) > x) \leq \exp\left( -\frac{cx^2}{\sigma_p^2 \left( \sum_{r=1}^{m} \left\| \mathbf{w}_{-y,r}^{(t)} \right\|_2 \right)^2} \right)$$

where $c$ is a constant and $\mathbb{E}g(\boldsymbol{\xi}) = \frac{\sigma_p}{\sqrt{2\pi}} \sum_{r=1}^{m} \left\| \mathbf{w}_{-y,r}^{(t)} \right\|_2$.

### D.1 Test Error Upper Bound

We now prove the upper bound on our test error in the benign overfitting regime as stated in Theorem 2. First note that for some given $(\mathbf{x}, y)$ we have,

$$\mathbb{P}(y \neq \text{sign}(f(\mathbf{W}^{(t)}, \mathbf{x}))) = \mathbb{P}(yf(\mathbf{W}^{(t)}, \mathbf{x}) \leq 0).$$

We can write,

$$yf(\mathbf{W}^{(t)}, \mathbf{x}) = F_y(\mathbf{W}_y^{(t)}, \mathbf{x}) - F_{-y}(\mathbf{W}_{-y}^{(t)}, \mathbf{x})$$

$$= \frac{1}{m} \sum_{r=1}^{m} \left[ \sigma\left( \langle \mathbf{w}_{y,r}^{(t)}, y\boldsymbol{\mu} \rangle \right) + \sigma\left( \langle \mathbf{w}_{y,r}^{(t)}, \boldsymbol{\xi} \rangle \right) \right] - \frac{1}{m} \sum_{r=1}^{m} \left[ \sigma\left( \langle \mathbf{w}_{-y,r}^{(t)}, y\boldsymbol{\mu} \rangle \right) + \sigma\left( \langle \mathbf{w}_{-y,r}^{(t)}, \boldsymbol{\xi} \rangle \right) \right]. \qquad (80)$$

Now note that since $t \geq T_1$ we know that $\sigma\left( \langle \mathbf{w}_{-y,r}^{(t)}, y\boldsymbol{\mu} \rangle \right) = 0$ for all $r \in [m]$ from Lemma 31. Thus,

$$\mathbb{P}(yf(\mathbf{W}^{(t)}, \mathbf{x}) \le 0) \le \mathbb{P}\left(\sum_{r=1}^m \sigma\left(\langle \mathbf{w}_{-y,r}^{(t)}, \boldsymbol{\xi}\rangle\right) \ge \sum_{r=1}^m \sigma\left(\langle \mathbf{w}_{y,r}^{(t)}, y\boldsymbol{\mu}\rangle\right)\right)$$

$$\overset{(a)}{=} \mathbb{P}\left(g(\boldsymbol{\xi}) - \mathbb{E}g(\boldsymbol{\xi}) \ge \sum_{r=1}^m \sigma\left(\langle \mathbf{w}_{y,r}^{(t)}, y\boldsymbol{\mu}\rangle\right) - \frac{\sigma_p}{\sqrt{2\pi}}\sum_{r=1}^m \left\|\mathbf{w}_{-y,r}^{(t)}\right\|_2\right)$$

$$\overset{(b)}{\le} \exp\left(-\frac{c\left(\sum_{r=1}^m \sigma\left(\langle \mathbf{w}_{y,r}^{(t)}, y\boldsymbol{\mu}\rangle\right) - \frac{\sigma_p}{\sqrt{2\pi}}\sum_{r=1}^m \left\|\mathbf{w}_{-y,r}^{(t)}\right\|_2\right)^2}{\sigma_p^2\left(\sum_{r=1}^m \left\|\mathbf{w}_{-y,r}^{(t)}\right\|_2\right)^2}\right)$$

$$= \exp\left(-c\left(\frac{\sum_{r=1}^m \sigma\left(\langle \mathbf{w}_{y,r}^{(t)}, y\boldsymbol{\mu}\rangle\right)}{\sigma_p \sum_{r=1}^m \left\|\mathbf{w}_{-y,r}^{(t)}\right\|_2} - \frac{1}{\sqrt{2\pi}}\right)^2\right)$$

$$\overset{(c)}{\le} \exp\left(\frac{c}{2\pi} - \frac{c}{2}\left(\frac{\sum_{r=1}^m \sigma\left(\langle \mathbf{w}_{y,r}^{(t)}, y\boldsymbol{\mu}\rangle\right)}{\sigma_p \sum_{r=1}^m \left\|\mathbf{w}_{-y,r}^{(t)}\right\|_2}\right)^2\right)$$

$$\overset{(d)}{\le} \exp\left(\frac{c}{2\pi} - \frac{n\|\boldsymbol{\mu}\|_2^4\left(|A_y| + (m - |A_y|)\left(h + \frac{1}{\tau}(1-h)\right)\right)^2}{C_5\sigma_p^4 m^2 d}\right)$$

$$\overset{(e)}{\le} \exp\left(-\frac{n\|\boldsymbol{\mu}\|_2^4\left(|A_y| + (m - |A_y|)\left(h + \frac{1}{\tau}(1-h)\right)\right)^2}{2C_5\sigma_p^4 m^2 d}\right).$$

Here $(a)$ follows from the definition of $g(\boldsymbol{\xi})$ in Lemma 35; $(b)$ follows from the result in Lemma 35; $(c)$ uses $(a-b)^2 \ge a^2/2 - b^2, \forall a, b \ge 0$; $(d)$ uses Lemma 34; $(e)$ follows from the benign overfitting condition $n\|\boldsymbol{\mu}\|_2^4 = \Omega\left(\sigma_p^4 d\right)$ and choosing sufficiently large $C_6$. Now note that,

$$L_{\mathcal{D}}^{0-1}(\mathbf{W}^{(T)}) = \sum_{j\in\{\pm 1\}} \mathbb{P}(y = j)\mathbb{P}(y \ne \text{sign}(f(\mathbf{W}^{(t)}, \mathbf{x}))$$

$$= \frac{1}{2}\sum_{j\in\{\pm 1\}} \exp\left(-\frac{n\|\boldsymbol{\mu}\|_2^4\left(|A_j| + (m - |A_j|)\left(h + \frac{1}{\tau}(1-h)\right)\right)^2}{2C_5\sigma_p^4 m^2 d}\right).$$

This completes our proof for the upper bound on the test error in the benign overfitting regime.

## D.2 Test Error Lower Bound

We first state some intermediate lemmas that we use in our proof.

**Lemma 36.** *(Lemma 5.8 in Kou et al. (2023))* *Let* $g(\boldsymbol{\xi}) = \sum_{j,r} j\sigma\left(\langle \mathbf{w}_{j,r}^{(T)}, \boldsymbol{\xi}\rangle\right)$. *If* $n\|\boldsymbol{\mu}\|_2^4 = \mathcal{O}\left(\sigma_p^4 d\right)$ *(harmful overfitting condition) then there exists a fixed vector v with* $\|\mathbf{v}\|_2^2 \le 0.06\sigma_p$ *such that*

$$\sum_{j'\in\{\pm 1\}} [g(j'\boldsymbol{\xi} + \mathbf{v}) - g(j'\boldsymbol{\xi})] \ge 4C_6 \max_{j\in\{\pm 1\}} \left\{\sum_r \Gamma_{j,r}^{(T)}\right\}$$

*for all* $\boldsymbol{\xi} \in \mathbb{R}^d$.

**Lemma 37.** *(Proposition 2.1 in Devroye et al. (2018))* *The* TV *distance between* $\mathcal{N}(\mathbf{0}, \sigma_p^2\mathbf{I}_d)$ *and* $\mathcal{N}(\mathbf{v}, \sigma_p^2\mathbf{I}_d)$ *is less than* $\|\mathbf{v}\|_2^2/2\sigma_p$.

**Proof.**

We have,

$$L_{\mathcal{D}}^{0-1}(\mathbf{W}^{(T)})$$

$$= \mathbb{P}_{(\mathbf{x},y)\sim\mathcal{D}}\left(y \neq \text{sign}(f(\mathbf{W},\mathbf{x}))\right)$$

$$= \mathbb{P}_{(\mathbf{x},y)\sim\mathcal{D}}\left(yf(\mathbf{W},\mathbf{x}) \leq 0\right)$$

$$\overset{(a)}{=} \mathbb{P}_{(\mathbf{x},y)\sim\mathcal{D}}\left(\sum_r \sigma\left(\langle\mathbf{w}_{-y,r}^{(T)},\boldsymbol{\xi}\rangle\right) - \sum_r \sigma\left(\langle\mathbf{w}_{y,r}^{(T)},\boldsymbol{\xi}\rangle\right) \geq \sum_r \sigma\left(\langle\mathbf{w}_{y,r}^{(T)},y\boldsymbol{\mu}\rangle\right) - \sum_r \sigma\left(\langle\mathbf{w}_{-y,r}^{(T)},y\boldsymbol{\mu}\rangle\right)\right)$$

$$\overset{(b)}{\geq} \mathbb{P}_{(\mathbf{x},y)\sim\mathcal{D}}\left(\sum_r \sigma\left(\langle\mathbf{w}_{-y,r}^{(T)},\boldsymbol{\xi}\rangle\right) - \sum_r \sigma\left(\langle\mathbf{w}_{y,r}^{(T)},\boldsymbol{\xi}\rangle\right) \geq C_6 \max\left\{\sum_r \Gamma_{1,r}^{(T)}, \sum_r \Gamma_{-1,r}^{(T)}\right\}\right)$$

$$\geq 0.5\mathbb{P}_{(\mathbf{x},y)\sim\mathcal{D}}\left(\left|\sum_r \sigma\left(\langle\mathbf{w}_{1,r}^{(T)},\boldsymbol{\xi}\rangle\right) - \sum_r \sigma\left(\langle\mathbf{w}_{-1,r}^{(T)},\boldsymbol{\xi}\rangle\right)\right| \geq C_6 \max\left\{\sum_r \Gamma_{1,r}^{(T)}, \sum_r \Gamma_{-1,r}^{(T)}\right\}\right)$$

$$\overset{(c)}{=} 0.5\mathbb{P}_{(\mathbf{x},y)\sim\mathcal{D}}\left(|g(\boldsymbol{\xi})| \geq C_6 \max\left\{\sum_r \Gamma_{1,r}^{(T)}, \sum_r \Gamma_{-1,r}^{(T)}\right\}\right)$$

$$\overset{(d)}{=} 0.5\mathbb{P}(\Omega). \tag{81}$$

Here $(a)$ follows from equation 80; $\mathbb{P}(y \neq \text{sign}(f(\mathbf{W}^{(t)},\mathbf{x}))) = \mathbb{P}(yf(\mathbf{W}^{(t)},\mathbf{x}) \leq 0)$; $(b)$ follows from $\sigma\left(\langle\mathbf{w}_{-y,r}^{(t)},y\boldsymbol{\mu}\rangle\right) = 0$ (Lemma 31) and $\sigma\left(\langle\mathbf{w}_{y,r}^{(t)},y\boldsymbol{\mu}\rangle\right) = \Theta\left(\Gamma_{y,r}^{(t)}\right)$; $(c)$ follows from defining $g(\boldsymbol{\xi}) = \sum_r \sigma\left(\langle\mathbf{w}_{1,r}^{(T)},\boldsymbol{\xi}\rangle\right) - \sum_r \sigma\left(\langle\mathbf{w}_{-1,r}^{(T)},\boldsymbol{\xi}\rangle\right)$; $(d)$ follows from defining $\Omega := \left\{\boldsymbol{\xi} : |g(\boldsymbol{\xi})| \geq C_6 \max\left\{\sum_r \Gamma_{1,r}^{(T)}, \sum_r \Gamma_{-1,r}^{(T)}\right\}\right\}$.

Now we know from Lemma Lemma 36, that $\sum_j \left[(g(j\boldsymbol{\xi}+\mathbf{v}) - g(j\boldsymbol{\xi})\right] \geq 4C_6 \max_j\{\sum_r \Gamma_{j,r}^{(T)}\}$. This implies that one one of the $\boldsymbol{\xi}, \boldsymbol{\xi}+v, -\boldsymbol{\xi}, -\boldsymbol{\xi}+v$ must belong to $\Omega$. Therefore,

$$\min\left\{\mathbb{P}(\Omega), \mathbb{P}(-\Omega), \mathbb{P}(\Omega-\mathbf{v}), \mathbb{P}(-\Omega-\mathbf{v})\right\} \geq 0.25 \tag{82}$$

Also note that by symmetry $\mathbb{P}(\Omega) = \mathbb{P}(-\Omega)$. Furthermore,

$$|\mathbb{P}(\Omega) - \mathbb{P}(\Omega-\mathbf{v})| = \left|\mathbb{P}_{\boldsymbol{\xi}\sim\mathcal{N}(\mathbf{0},\sigma_p^2\mathbb{I}_d)}(\boldsymbol{\xi} \in \Omega) - \mathbb{P}_{\boldsymbol{\xi}\sim\mathcal{N}(\mathbf{v},\sigma_p^2\mathbb{I}_d)}(\boldsymbol{\xi} \in \Omega)]\right|$$

$$\overset{(a)}{\leq} \text{TV}\left(\mathcal{N}(\mathbf{0},\sigma_p^2\mathbf{I}_d), \mathcal{N}(\mathbf{v},\sigma_p^2\mathbf{I}_d)\right)$$

$$\overset{(b)}{\leq} \frac{\|\mathbf{v}\|_2^2}{2\sigma_p}$$

$$\leq 0.03. \tag{83}$$

Here $(a)$ follows from the definition of TV distance; $(b)$ follows from Lemma Lemma 37. Thus we see that equation 83 along with equation 82 implies that $\mathbb{P}(\Omega) = 0.22$. Substituting this in equation 81 we get $L_{\mathcal{D}}^{0-1}(\mathbf{W}^{(T)}) = 0.1$ as claimed.

# E    Main Paper Lemma Proofs

## E.1    Proof of Lemma 1

This lemma follows from directly from Lemma 29 and the constant lower bound on cross-entropy loss derivatives, i.e., Equation (58).

## E.2    Proof of Lemma 2

This lemma follows from directly from Lemma 30 and the constant lower bound on cross-entropy loss derivatives, i.e., Equation (58).

### E.3 Proof of Lemma 3

Using our result in Lemma 28 with $\tau = 1$ and $h = 0$, we have after $T_1 = \mathcal{O}\left(\frac{mn}{\eta \sigma_p^2 d}\right)$ iterations for all $j \in \{\pm 1\}$ and $r \in [m]$,

$$\Gamma_{j,r}^{(\text{pre},T_1)} \geq \frac{\eta \left\|\boldsymbol{\mu}^{(\text{pre})}\right\|_2^2}{4m} \sum_{t=0}^{T_1 - 1} \min_i \left| \ell_i'^{(\text{pre},t)} \right| \overset{(a)}{\geq} \frac{\eta \left\|\boldsymbol{\mu}^{(\text{pre})}\right\|_2^2 C T_1}{4m} = \Omega\left( \frac{n \left\|\boldsymbol{\mu}^{(\text{pre})}\right\|_2^2}{\sigma_p^2 d} \right).$$

Here $(a)$ follows from equation 58. Now for any $t \geq T_1$ we have from Lemma 4,

$$\langle \mathbf{w}_{j,r}^{(\text{pre},t)}, j\boldsymbol{\mu}^{(\text{pre})} \rangle = \langle \mathbf{w}_{j,r}^{(\text{pre},0)}, j\boldsymbol{\mu}^{(\text{pre})} \rangle + \Gamma_{j,r}^{(\text{pre},t)}$$

$$\overset{(a)}{\geq} \langle \mathbf{w}_{j,r}^{(\text{pre},0)}, j\boldsymbol{\mu}^{(\text{pre})} \rangle + \Gamma_{j,r}^{(\text{pre},T_1)}$$

$$\overset{(b)}{\geq} -\Theta\left( \sqrt{\log(m/\delta)}(\sigma_p d)^{-1}\sqrt{n} \left\|\boldsymbol{\mu}^{(\text{pre})}\right\|_2 \right) + \Omega\left( \sigma_p^{-2} d^{-1} n \left\|\boldsymbol{\mu}^{(\text{pre})}\right\|_2^2 \right)$$

$$\overset{(c)}{\geq} 0,$$

where $(a)$ follows from the fact that $\Gamma_{j,r}^{(t)}$ is non-decreasing with respect to $t$, $(b)$ follows from Assumption 4 and Lemma 6; $(c)$ follows from Assumption 3.

$\square$

## F   Additional Experimental Details and Results

**Implementation.**   We use PyTorch Paszke et al. (2019) to run all our algorithms and also simulate our synthetic data setting. For experiments on neural network training we use one H100 GPU with 2 cores and 20GB memory. For synthetic data experiments we use one T4 GPU. The approximate total run-time for all our experiments on neural networks is about 36 hours. The approximate total run-time for all experiments on the synthetic data setting is about 1 hour.

**Details for Figure 1.**   We simulate a FL setup with $K = 10$ clients on the CIFAR10 data partitioned using Dirichlet($\alpha$) with $\alpha = 0.1$ for the non-IID setting and $\alpha = 10$ for the IID setting. For pre-training, we consider a Squeezenet model pre-trained on ImageNet Russakovsky et al. (2015) which is available in PyTorch. Following Nguyen et al. (2022) we replace the BatchNorm layers in the model with GroupNorm Wu & He (2018). For FL optimization we use the vanilla `FedAvg` optimizer with server step size $\eta_g = 1$ and train the model for 500 rounds and 1 local epoch at each client. For centralized optimization we use `SGD` optimizer and run the optimization for 200 epochs. Learning rates were tuned using grid search with the grid $\{0.1, 0.01, 0.001\}$. Final accuracies were reported after averaging across 3 random seeds.

**Details for and Figure 2 and Figure 3.**   For these experiments we simulate a synthetic data setup following our data model in Section 2. We set the dimension $d = 200$, $n = 20$ datapoints (we keep $n$ small to ensure we are in the over-parameterized regime), $m = 10$ filters, $K = 2$ clients, $N = 10$ local datapoints. The signal strength is $\|\boldsymbol{\mu}\|_2^2 = 3$, noise variance is $\sigma_p^2 = 0.1$ and variance of Gaussian initialization is $\sigma_0 = 0.01$. The global dataset has 10 datapoints with positive labels and 10 datapoints with negative labels. We also create a test dataset of 1000 datapoints following the same setup to evaluate our test error.

**Details for Figure 5 and Figure 6.**   We simulate a FL setup with $K = 20$ clients using Dirichlet($\alpha$) Hsu et al. (2019). For pre-training, we consider a ResNet18 model pre-trained on ImageNet Russakovsky et al. (2015) which is available in PyTorch. Following Nguyen et al. (2022) we replace the BatchNorm layers in the model with GroupNorm Wu & He (2018). For FL optimization we use the `FedAvg` optimizer with server step size $\eta_g = 1$ and 1 local epoch at each client. In the case of random initiation, for local optimization we use SGD optimizer with a learning rate of 0.01 and 0.9 momentum. In the case of pre-trained initiation, for local optimization we use SGD optimizer with a learning rate of 0.001 and 0.9 momentum. The learning rate is decayed by a factor of 0.998 in every round in the case for both initializations. Each experiment is repeated with 3 different random seeds.

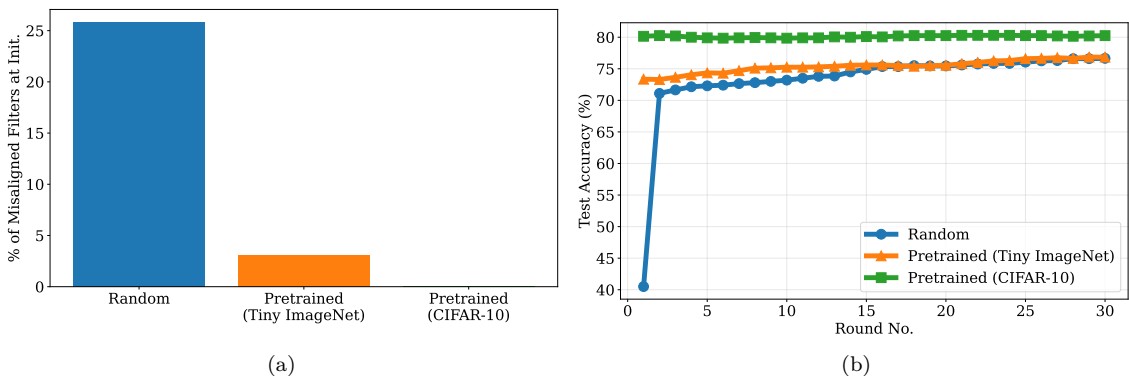

Figure 7: The percentage of misaligned filters (Figure 7a) and test accuracy (Figure 7b) for different initializations on CIFAR-10 with $\alpha = 0.05$ heterogeneity when training a 2-layer CNN.

**Additional Experiment.** To simulate an experimental setting aligned with our theoretical analysis, we conduct the following study. We partitioned two classes of CIFAR-10 (airplane and car) across $K = 20$ clients with $\alpha = 0.05$ heterogeneity. We then considered FL training of a 2-layer CNN model with $m = 64$ filters and evaluated three initialization strategies:

1. **Pre-training on centralized CIFAR-10**: The model is pre-trained centrally for 5 epochs on the same CIFAR-10 dataset later used for federated training.

2. **Pre-training on TinyImagenet:** We pre-trained the CNN on two semantically related classes in TinyImagenet - albatross and racecar, which resemble the airplane and car categories in CIFAR-10.

3. **Random initialization**

As expected, pre-training on the same dataset yields the strongest improvement, with 0% misaligned filters at initialization and the highest test accuracy at the end of FL training. Importantly, pre-training on TinyImagenet, while not perfectly aligned, still substantially reduces the proportion of misaligned filters (3.1% vs. 25.7%) and leads to better test accuracy than random initialization. These findings strongly support our theoretical insights: the reduction in misaligned filters at initialization directly correlates with improved FL performance, thereby reinforcing the connection between our theory and empirical results.

