# OpenReview forum: "Initialization Matters: Unraveling the Impact of Pre-Training on Federated Learning"
_TMLR — Accepted by TMLR_

### Review · Reviewer_8fwx · 2025-06-20

**Summary Of Contributions:**

The paper investigates the idea of using pre-trained models as starting points for models that are used in a federated learning setting.  In particular, the authors provide a justification as to why the use of such pre-trained models can provide faster and more stable convergence to a good enough learnt model.  In this direction the authors study convolutional neural networks (CNNs) that have two layers and are trained using the FedAvg algorithm for a federated learning setting.  One justification for the choice for this class is that they have been investigated recently in the context of batch learning.  Another justification (this is my interpretation) is that they allow for the proof of some intuitive theoretical results that the authors provide in their work.  In particular, the authors provide rigorous results on the convergence rate and on the test error bound for such models.  In addition, the authors also provide an empirical evaluation for the behavior of the tools that they have defined (or developed) in this context using the CIFAR-10 and TinyImageNet datasets.

**Audience:**

Yes

**Broader Impact Concerns:**

**Q1.**  Will you make your code publicly available upon acceptance?

**Q2.**  Condition (C2) states that we have much fewer total examples in all the clients compared to the dimensionality of the data.  Is this a reasonable assumption?  I believe it is the other way around (e.g., n is number of patients in a federated healthcare system).  In that light, the results that you obtain are now weaker because one, in principle, would care about the situation where our data points far exceed their dimensionality.  Even in traditional batch learning, interesting model spaces have VC-dimension that is linear or superlunar in the dimensionality of the data and therefore the number of examples that we need for generating "good" hypotheses is linear/super-linear in the dimensionality of the data.  Having said all these, the question is to what extent your results may hold for much smaller d?  If you do not have any rigorous results in this direction, do you have any experimental results?

**Q3.**  Do you have any minor comments on the class-imbalance issue that I raised above?

**Claims And Evidence:**

Yes

**Requested Changes:**

**W1.**  While the simplicity of the class of models that the authors investigate (two-layer CNNs) allows them to provide some rigorous guarantees, nevertheless, it is unclear how many people will work with such a class of models in the real-world, especially outside of standard benchmarks of computer vision.

**W2.**  While Strength S4 is welcome, it should be stated that the paper is longer than usual papers.  Personally, I would prefer in some cases sketches of the proofs of some theorems in the main text (i.e., first 12 pages), but given the length of the proofs in the appendix, I understand that this may not be a feasible after all.  So, I would try to urge the authors to consider providing 1-2 paragraphs giving a sketch of the proofs and the important pieces that when put together can justify intuitively even more the results that the authors obtain.

**W3.**  Missing discussion on class-imbalance?

The parameter h for heterogeneity is like an "average" metric for the class-imbalance that you have among the various clients.  In other words, while you use some global distribution D in page 3 (paragraph "Measure of Data Heterogeneity") the fact that you are partitioning the generated dataset arbitrarily, leads to different class ratios among the clients. Which now begs the question if you are missing discussion on class imbalance.


**W4.**  Minor issues in the text.

- In page 3, before the paragraph "Measure of Data Heterogeneity" you mention Condition (C2).  At that point in time (C2) is undefined.  First of all, I think you need to write "Condition (C2)" instead of just "(C2)".  Then you also need some surrounding text to refer to "Condition 1" in page 5, which has these six "sub-conditions".  Of course you will lose a line in the text there and this may push back some of your content, but sentences should be complete.  Related to that, I acknowledge that it is a little bit cumbersome to have "Condition 1" and then the conditions (C1)-(C6) listed there, because now in the actual text if one refers to "Condition (C1)" this can be confused with the environment "Condition 1".  I think something needs to change there but there is no easy way out.  Especially since you are also using the terms "Assumption 1", etc. in the appendix!  Anyway, I do not have very strong feelings, but it seems a little bit cumbersome there.

- Defining the loss function $\ell$ to be taking only one argument is odd.

- I have a feeling that Figure 4 in page 8 is never mentioned in the actual text.  Also, while we are at it, I believe you should introduce it it before you start discussing Figure 3 and its results because it really allows one to understand the values of the different filters and why the results of Figure 3 make sense as they are.  In the current flow of presentation they did not make much sense to me initially, until I actually "ignored" my issue in the paragraph of page 8 and then later focused on Figure 4 that really elucidated the results of Figure 3.

**Strengths And Weaknesses:**

**S1.**  Providing an analysis and efficient rigorous computational results is always welcome in machine learning.

**S2.**  This is especially true for federated learning that is much less studied than batch learning.

**S3.**  The authors define meaningful quantities and obtain reasonable and intuitive results.

**S4.**  The authors provide detailed analysis for their results and the appendix goes even beyond that and provides additional information on related work and also has a table with notation.

---

> ### Author Response · Authors · 2025-08-28
> **Response to Reviewer 8fwx [1/2]**
>
> Thank you for your thoughtful review. We greatly appreciate your recognition of the strengths of our paper and address the requested changes and broader impact questions in turn.
>
> ## Re: Requested Changes
>
> > W1. While the simplicity of the class of models that the authors investigate (two-layer CNNs) ...it is unclear how many people will work with such a class of models in the real-world, especially outside of standard benchmarks of computer vision.
>
> While the class of two-layer CNNs may appear simple, the theoretical analysis is non-trivial: we establish convergence to global minima in a non-convex setting while also providing generalization guarantees. This setting serves as a rich and tractable playground to theoretically model several important ML phenomena. For instance, this model class has already been used to:
>
> * Explain the inferior generalization performance of Adam compared to SGD [a]
> * Provide theoretical insights into benign overfitting [b, c]
> * Demonstrate the generalization benefits of FedAvg over local training [d]
> * Show the efficacy of patch-level data augmentation techniques such as Cutout and CutMix [e]
> * Prove that federated learning with local steps can capture heterogeneous features more efficiently than parallel SGD [f]
>
> Thus, despite their simplicity, two-layer CNNs continue to play an important role as a theoretically grounded testbed for understanding broader ML phenomena. To address concerns about generalizability, we have added a new paragraph in our Conclusion section titled “Extension to Deeper CNNs and Transformer Architectures” that outlines the challenges and opportunities in extending filter alignment analysis beyond shallow CNNs, and motivates this as an exciting avenue for future work.
>
> > W2. While Strength S4 is welcome...urge the authors to consider providing 1-2 paragraphs giving a sketch of the proofs and the important pieces that when put together can justify intuitively even more the results that the authors obtain.
>
> Thank you for pointing this out. We have added a proof sketch after both Theorem 1 and Theorem 2 in our revised paper.
>
> > W3. Missing discussion on class-imbalance? The parameter h for heterogeneity is like an "average" metric for the class-imbalance ...Which now begs the question if you are missing discussion on class imbalance.
>
> Thank you for pointing this out. We consider class imbalance as an orthogonal dimension to our current analysis. Intuitively, $h$ measures how much the local client distributions deviate, on average, from the global distribution $\mathcal{D}$. Since the global distribution $\mathcal{D}$ is balanced we can consider the local deviations as class imbalance. Nonetheless, our analysis can be extended to the case where the global distribution is itself unbalanced, as we explain below. Let us consider a global data distribution with probability of label $1$ as $p$ and probability of label $-1$ as $1-p$. Further let us assume without loss of generality that $p \leq 1-p$. In this case, we can modify our definition of $h$ as follows:
>
> $$
> h_p := \frac{\sum_{k=1}^K \min \{D_{+,k}, D_{-,k}\}}{2pn} \in \left[0, \tfrac{1}{2}\right].
> $$
>
> We see that compared to the original definition of heterogeneity in Equation (1), we have to divide by a factor of $2p$ to account for the data imbalance. When the data is balanced, i.e., $p = \tfrac{1}{2}$, $h_p$ reduces to $h$. Also the range of values taken by $h_p$, i.e., $\left[0, \tfrac{1}{2}\right]$ remains the same for all $p$, with a lower value of $h_p$ indicating a more heterogeneous setup. Thus our definition of heterogeneity as well as subsequent analysis can be easily modified to account for data imbalance.
>
>
> > W4. Minor Issues in the text
>
> Thank you for pointing this out. We have fixed these, namely referring to (C2) as Condition (C2) and renaming Condition 1 to Main Condition to avoid confusion, redefining the loss function to take two arguments and introducing Figure 4 before discussing the Figure 3 and its results in Section 3.4.

---

> > ### Author Response · Authors · 2025-08-28
> > **Response to Reviewer 8fwx [2/2]**
> >
> > ## Re: Broader Impact Concerns
> >
> > > Q1. Will you make your code publicly available upon acceptance?
> >
> > Yes, we will make our code publicly available upon acceptance.
> >
> > > Q2. Condition (C2) states that we have much fewer total examples in all the clients compared to the dimensionality of the data. Is this a reasonable assumption...If you do not have any rigorous results in this direction, do you have any experimental results?
> >
> > Condition (C2) is needed to ensure that the model is sufficiently overparameterized and the training loss can be made arbitrarily small (similar assumptions have also been made in [b,c]). Recall that our input $\mathbf{x}$ consists of a signal component $\mathbf{\mu}\in \mathbb{R}^d$ that is common across all datapoints and noise component $\mathbf{\xi} \in \mathbb{R}^d$ that is independently drawn from $\mathcal{N}(0,\sigma_p^2\cdot \mathbf{I})$. Having a sufficiently large $d$ ensures that the correlation between any two noise vectors, i.e. $\langle \mathbf{\xi}, \mathbf{\xi}' \rangle/ \|\|\mathbf{\xi}\|\|^2$ is not too large. Otherwise if the correlation between two noise vectors is large and negative, then minimizing the loss on one data point could end up increasing the loss on another training point which complicates convergence and prevents loss from becoming arbitrarily small.
> >
> >
> > We acknowledge, that in domains such as federated healthcare, the opposite situation is more natural: the total sample size $n$ (e.g., patients) usually exceed the dimensionality of the feature space. Empirically, in our experiments with high-dimensional models (e.g., CIFAR-10 with CNNs, $n \sim 10^4, d \sim 10^3$), we observe that the proposed methods remain effective even when this condition does not hold. While we do not provide rigorous results for the $n \gg d$ regime, the empirical evidence suggests that our approach is not restricted to strictly over-parameterized settings.
> >
> > > Q3. Do you have any minor comments on the class-imbalance issue that I raised above?
> >
> > Please see discussion on W3 above.
> >
> > ---
> >
> > **References**
> >
> > [a] Zou, D., Cao, Y., Li, Y., and Gu, Q. Understanding the generalization of adam in learning neural networks with proper regularization. *ICLR, 2023*.
> >
> > [b] Cao, Y., Chen, Z., Belkin, M., and Gu, Q. Benign overfitting in two-layer convolutional neural networks. *NeurIPS, 2022*
> >
> > [c] Kou, Y., Chen, Z., Chen, Y., and Gu, Q. Benign overfitting in two-layer relu convolutional neural networks. *ICML, 2023*
> >
> > [d] Huang, W., Shi, Y., Cai, Z., and Suzuki, T. Understanding convergence and generalization in federated learning through feature learning theory. *ICLR, 2023*.
> >
> > [e] Oh, Junsoo, and Chulhee Yun. "Provable Benefit of Cutout and CutMix for Feature Learning." High-dimensional Learning Dynamics 2024: The Emergence of Structure and Reasoning. 2024.
> >
> > [f] Bao, Y., Crawshaw, M., and Liu, M. Provable benefits of local steps in heterogeneous federated learning for neural networks: A feature learning perspective. *ICML, 2024*.

---

> > > ### Comment · Reviewer_8fwx · 2025-09-10
> > > **Thank you**
> > >
> > > Dear Authors,
> > >
> > > Thank you for the very detailed response to my comments.  I also appreciate the detailed responses that you have provided to the comments of the other reviewers.

---

### Review · Reviewer_5iQi · 2025-07-03

**Summary Of Contributions:**

This work studies the effect of model initialization for federated learning, specifically focusing on the FedAvg algorithm. Studies have shown that there are advantages to using a pre-trained instead of a randomly-initialized model when wanting to fine-tune on a downstream task, especially when it comes to generalization. The authors look into this phenomenon from the point of view of CNNs and a particular learning algorithm, FedAvg.

In this, mostly theoretical work, the authors consider a data model and two-layer CNN that is quite similar to the one from Cao, et al, 2022[^1]. In this setup, data points contain features $x=[x(1)^T, x(2)^T]\in \mathbb{R}^{2d}$ and each data point is assigned one of two labels, $y\in \{-1,1\}$. At random, one component is set to a signal vector $y\mu$ or to a Gaussian noise vector.

With this setup, the authors introduce the notion of aligned filters – a filter is aligned with a signal vector if their inner product is positive.

The main contributions after this setup are:
1. Based on the data model and assumptions on the SNR, dimensionality of input, training set size, etc., the authors show the training loss convergence of FedAvg as well as lower and upper bounds on the test error according to a respective assumption on the value of SNR. The upper bound is especially interesting as it ties together the contribution of the aligned and misaligned filters to the worst test loss we can get.
2. The authors also show that there is a time during pre-training after which all filters are aligned and so we get the benefits of Theorem 2. without any misaligned parts (and thus no impact from heterogeneity).
3. An empirical measure of misalignment is defined and used with ResNet18, both a randomly-initialised version and a pre-trained one, and image datasets to show some of the predictions of the theory for more realistic examples. The authors demonstrate that the pre-trained version shows smaller misalignment and better test accuracy.

Overall I enjoyed the notion of misalignment enough that I'm wondering if parallel notions to that are applicable for other setups.

[^1]: Cao, Y., Chen, Z., Belkin, M. and Gu, Q., 2022. Benign overfitting in two-layer convolutional neural networks. _Advances in neural information processing systems_, _35_, pp.25237-25250.

**Audience:**

Yes

**Claims And Evidence:**

Yes

**Requested Changes:**

Typos:
- Page 3, missing citation?!, “can be easily relaxed as done in ”.
-  Page 4, towards the end, please capitalize proper nouns, “equation 2 - equation 3”.
- Page 8, Empirical Verification part, “Figure Figure 3a” (delete first “Figure). Please proofread that paragraph for typos.

## Not necessary for recommendation

- Equation (8), the misalignment measure: was that perhaps meant to be an average over the batch?

**Strengths And Weaknesses:**

## Strengths

- The goals of the paper are clear and the theoretical analysis seems sound.
- The theoretical results (especially Theorem 2) tie all the important quantities with the test error and motivate the study of misalignment.
- Empirical results on CIFAR-10 and TinyImageNet at the very least show correlation with what is expected from theory as well as the correlation between complexity of data and expected misalignment when starting from a random initialization.

## Weaknesses

- The analysis, while strong, is limited to a binary classification task with a two-layer CNN and the standard FedAvg algorithm (the authors acknowledge this in the conclusion as future work). I think the paper would be strengthened by a brief discussion on the challenges of extending this "filter alignment" concept to other architectures (e.g., Transformers, where the notion of a filter is less direct) or other federated algorithms.

---

> ### Author Response · Authors · 2025-08-28
> **Response to Reviewer 5iQi**
>
> Thank you for your thoughtful review. We greatly appreciate your recognition of the strengths of our work specifically, the soundness of our theoretical analysis, the motivation for studying misalignment, and the empirical evidence we provide showing strong correlation with theory. Below, we address the noted weaknesses and requested changes in detail.
>
> ## Re: Weaknesses
>
> > The analysis, while strong, is limited to a binary classification task with a two-layer CNN ....extending this "filter alignment" concept to other architectures (e.g., Transformers, where the notion of a filter is less direct) or other federated algorithms.
>
>
> Thank you for this suggestion. We have added a new paragraph in the Conclusion section titled *“Extension to Deeper CNNs and Transformer Architectures”* to discuss both the challenges and opportunities in extending filter alignment beyond shallow CNNs.
>
>
> For deeper CNNs, we anticipate that we will need a hierarchial feature model [a,b,c] to capture alignment in deeper layers. Intuitively, while deeper architectures provide the capacity to learn more complex hierarchical signals, aligning a filter with the signal will also progressively become harder across layers with the alignment of the $L$-th layer potentially depending on the alignment of the ($L$ - 1)-th layer. In non-IID FL, this suggests that feature learning deteriorates as we move from the base layer toward the final layer. This intuition is supported by empirical evidence (see experiment in Table 1 in [d]) and explains why pre-trained initialization offers greater benefits for deeper networks compared to shallower ones.
>
> For extending to Transformers, recall that alignment of a convolutional filter is defined via the sign of its inner product with an input patch containing the signal (see Definition 1). For Transformers, the convolution operation is replaced by the self-attention mechanism. However, the core principle remains: intermediate representations are computed as inner products between the previous-layer embeddings and the current-layer attention weights. Specifically, parameterizing a self-attention layer by $(Q, K, V) \in \mathbb{R}^d$ and input $z \in \mathbb{R}^{N \times d}$, we have:
>
> $$q = zQ, \quad k = zK, \quad v = zV \quad \text{(intermediate representations)}$$
>
> $$z' = \mathrm{softmax}\left(\frac{qk^\top}{\sqrt{d}}\right)v \quad \text{(self-attention aggregation)}.$$
>
> Notice that each column of $Q$, $K$, and $V$ is involved in an inner product with the input $z$, analogous to filters in CNNs. Thus, our notion of filter alignment and its empirical measurement (Eq. (8)) can naturally be extended to these matrices. This connection provides a promising foundation for generalizing alignment analysis to Transformer-based architectures. Lastly, we note that recent work has also begun to explain benign overfitting in Transformer models [e, f], which could serve as a starting point for extending our analysis.
>
> ## Re: Requested Changes
>
> Thank you for pointing out the typos. We have corrected all of them. Regarding Eq. (8), note that our analysis only depends on the sign of $\mathcal{A}(w^{(t)})$ to determine alignment. Since the sign remains unaffected by averaging, the distinction does not impact our conclusions.
>
> ---
>
> **References**
>
> [a] Binghui Li and Yuanzhi Li. Adversarial training can provably improve robustness: Theoretical analysis of feature learning process under structured data. *arXiv preprint arXiv:2410.08503, 2024.*
>
> [b] Zeyuan Allen-Zhu and Yuanzhi Li. Backward feature correction: How deep learning performs deep (hierarchical) learning. *COLT, 2023*
>
> [c] Peng Wang et al.. Understanding deep representation learning via layerwise feature compression and discrimination. *arXiv preprint arXiv:2311.02960,2023b*
>
> [d] Yaodong Yu, et al. TCT: Convexifying federated learning using bootstrapped neural tangent kernels. *NeurIPS, 2022*
>
> [e] Jiarui Jiang et al. Unveil benign overfitting for transformer in vision: Training dynamics, convergence, and generalization. *NeurIPS, 2024*
>
> [f] Spencer Frei and Gal Vardi. Trained transformer classifiers generalize and exhibit benign overfitting in-context. *arXiv preprint arXiv:2410.01774, 2024.*

---

> > ### Comment · Reviewer_5iQi · 2025-09-04
> > **Thank you for the in-depth response.**
> >
> > I'll also be reviewing the replies to the comments the rest of the reviewers left.

---

### Review · Reviewer_p1Zg · 2025-08-11

**Summary Of Contributions:**

This paper provides a theoretical analysis of the impact of pre-training on federated learning with non-IID data. This is achieved via studying the class of two-layer convolutional neural networks (CNNs) for binary classification. The studies provide bounds on the training error convergence and generalized test error of the network trained with FedAvg. At the core of this analysis is the notion of aligned and misaligned filters at initialization and a demonstration that data heterogeneity only affects learning on misaligned filters. The study goes on and show that starting with a pre-trained model results in fewer misaligned filters at initialization, reducing the generalized test error.

**Audience:**

Yes

**Claims And Evidence:**

Yes

**Requested Changes:**

I suggest the following changes:

1. To connect better between experiment & theory, the authors can consider re-partition the 10-class CIFAR-10 dataset into a 2-class dataset, pre-train the CNN & use it as initialization for FedAvg. We expect to see that (1) the pre-trained CNN has good generalization and (2) using it as initialization, the FL performance improves significantly.

2. Repeat the above experiment with the pre-trained CNN on a different dataset, e.g., TinyImageNet, and see if we also see a drastic improvement on the target dataset with FL.

3. Elaborate on the alignment notion. It is currently mentioned in Lemma 3 and very briefly even though this is a central concept. Particularly, I would like to see discussion on how this notion can or can be generalized to accommodate multi-class classification.

4. Elaborate on how the pre-trained model's generalizability comes into the picture.

5. It'd also be good to discuss how the current model can be generalized to deep CNN settings (e.g., ResNet-18) with multiple CNN layers. There might be a difference between filter alignment at raw input level and alignment at higher feature level. In other words, it is unclear from the current theory whether filter alignment at input level alone is sufficient to characterize the entire benefit of pre-training on federated learning.

**Strengths And Weaknesses:**

Strengths:

This paper is well-written in general (pending a few minor points)
The research angle is interesting and can potentially lead to new insights on using pre-trained model in federated learning.

Weaknesses:

First, I am concerned that the theoretical setting here is too detached from existing practice in federated fine-tuning. For example, the CNN construct is restricted to a binary classification setting with 1 convolution layer followed by a linear layer. This is disconnected from most federated learning work.

I am aware that this setting has been adopted in previous works for different forms of analysis in centralized and/or federated learning w/o initializing with pre-trained models. I am however not sure if this simplified setting will fit with the broad research question raised by the authors (2nd page):

"Why can pre-trained initialization drastically improve model performance in FL?"

Drastic performance improvement requires the pre-trained model to be expressive and well-trained with large volumes of relevant data. How is this going to be the case with such simplified CNNs?

I do not see how this is demonstrated in the empirical verification. It is also unclear to me how the pre-trained model generalizability (i.e., its quality) does not come into the picture?

Second, the experiment was designed with a much more generic pre-trained ResNet which does reiterate known observations that initializing with pre-trained model helps but it does not support the theoretical model here. To connect better between experiment & theory, the authors can consider re-partition the 10-class CIFAR-10 dataset into a 2-class dataset, pre-train the CNN & use it as initialization for FedAvg. We expect to see that (1) the pre-trained CNN has good generalization and (2) using it as initialization, the FL performance improves significantly.

Third, the overall analysis seems to prove that with a sufficient no. of training iterations, the FL model can align the learning signal with that of the pre-trained model which is perhaps implicitly assumed to align with the data generation model. This raises another question: suppose the pre-trained model was not created using the same (centralized) dataset, how will it affect alignment theoretically? This is important to address because most of the time, FL algorithms were not initialized with models pre-trained on the exact target data.

--

Minor points:

I feel that the concept of aligned and misaligned filters need additional elaboration on the high level. It is not clear what is being aligned when these concepts were mentioned the first time in the introduction. Furthermore, going by the inline definition in Lemma 1, I have the impression that such directional alignment might not be sufficient when we generalize to multi-class classification. Can the author elaborate more on this?

At multiple points, the authors mentioned that some assumptions (e.g., orthogonality) are meant to ease the technical exposition and relaxing them is not difficult. Can you please elaborate on high level how can these be easily relaxed?

--

Overall, I feel that the paper might have over-claimed their theoretical contribution. The key research question probably needs to be rephrased to reflect more accurately the theoretical setting. Otherwise, there is a risk of over-generalization.

---

> ### Author Response · Authors · 2025-08-28
> **Response to Reviewer p1Zg [1/3]**
>
> Thank you for appreciating the writing in our paper, research angle and potential for new insights. We discuss the weaknesses, minor points and requested changes below.
>
> ## Re: Weaknesses
>
> We first address Weakness 3, as we believe that sets up better context for our response to Weakness 1.
>
> > W3. This raises another question....This is important to address because most of the time, FL algorithms were not initialized with models pre-trained on the exact target data.
>
> This seems to be a misunderstanding. Our analysis does **not** require the pre-training dataset to be identical to the fine-tuning dataset. Instead, it relies on a measure of distance between the signal in the pre-training data and the fine-tuning data, expressed as $\|\| \mu^{\text{(pre)}} - \mu \|\|$. Lemma 3 shows that, given sufficient training iterations, CNN filters will eventually aligned with the signal present in the training data. Suppose we pre-train on a dataset with signal $\mu^{\text{(pre)}}$. By the end of pre-training, from Lemma 3 we know that $\langle w, \mu^{\text{(pre)}} \rangle > 0$. Now consider fine-tuning on a dataset with signal $\mu$. In this case, $\langle w, \mu \rangle = \langle w, \mu^{\text{(pre)}} \rangle + \langle w, \mu - \mu^{\text{(pre)}} \rangle$. Since $\langle w, \mu^{\text{(pre)}} \rangle > 0$, if the distance between the signals is small enough, namely
> $$
> \| \mu^{\text{(pre)}} - \mu \| \leq \frac{\langle w, \mu^{\text{(pre)}} \rangle}{\|w\|}
> $$
> then we can guarantee that $\langle w, \mu \rangle > 0$. This means the filter remains aligned with the signal in the fine-tuning data.  Thus, the *generalizability of pre-training* depends on how close the two signals are, as captured by $\|\| \mu^{\text{(pre)}} - \mu \|\|$.
>
>
> > W1. Drastic performance improvement requires the pre-trained model...It is also unclear to me how the pre-trained model generalizability (i.e., its quality) does not come into the picture?
>
> We fully agree that achieving substantial performance improvements requires the pre-trained model to be both expressive and trained on large volumes of relevant data. Our theory explicitly supports this point. In particular, by requiring $d$ to be large (Condition C2), we ensure that the model is overparameterized and thus capable of aligning with multiple distinct signals simultaneously (i.e., expressiveness). At the same time, relevance of the pre-training data is crucial: the distance between the pre-training signal and the fine-tuning signal should be small, $\| \mu^{\text{(pre)}} - \mu \|$ as explained in our response to Weakness 3. This is precisely how the generalizability of the pre-trained model enters our analysis.
>
> We also note that the goal of our empirical experiments is not to re-establish the already well-known result that pre-training improves test accuracy for downstream FL tasks. Instead, our focus is to show that these improvements can be directly explained through our theoretical analysis of *misalignment*. To this end, we introduce a novel misalignment measure in Equation (8) and empirically demonstrate that pre-trained models exhibit fewer misaligned filters at initialization (Figure 5 (a) and (c), thereby providing a principled explanation for the observed gains.
>
>
> >W2.  Second, the experiment was designed with a much more generic pre-trained ResNet..wexpect to see that (1) the pre-trained CNN has good generalization and (2) using it as initialization, the FL performance improves significantly.
>
> Thank you for this helpful suggestion. We conducted an additional experiment along the lines you proposed. We partitioned two classes of CIFAR-10 (airplane and car) across $K = 20$ clients with $\alpha = 0.05$ heterogeneity. We then considered FL training of a 2-layer CNN model with $m=64$ filters and evaluated three initialization strategies:
>
> 1. **Pre-training on centralized CIFAR-10**: The model is pre-trained centrally on the same dataset later used for federated fine-tuning.
> 2. **Pre-training on TinyImagenet**: Following your suggestion in *Requested Changes 2*, we pre-trained the CNN on two semantically related classes in TinyImagenet - albatross and racecar, which resemble the airplane and car categories in CIFAR-10.
> 3. **Random initialization**

---

> > ### Author Response · Authors · 2025-08-28
> > **Response to Reviewer p1Zg [2/3]**
> >
> > [Table 1:Results of federated fine-tuning under different initialization strategies for a two-layer CNN model]:
> >
> > | Initialization Strategy      | % Misaligned Filters | Avg. Test Accuracy (30 Rounds) |
> > |-------------------------------|-----------------------|--------------------------------|
> > | Random                       | 25.78                 | 73.36                          |
> > | Pre-trained on TinyImagenet   | 3.12                  | 75.42                          |
> > | Pre-trained on CIFAR-10       | 0.0                  | 80.30                          |
> >
> > As expected, pre-training on the same dataset yields the strongest improvement, with *0% misaligned filters* at initialization and the highest test accuracy. Importantly, pre-training on TinyImagenet, while not perfectly aligned, still substantially reduces the proportion of misaligned filters compared to random initialization (3.1% vs. 25.7%) and leads to better test accuracy than random initialization. These findings strongly support our theoretical insights: the reduction in misaligned filters at initialization directly correlates with improved FL performance, thereby reinforcing the connection between our theory and empirical results.
> >
> >
> > ## Re: Minor Points
> >
> > > I feel that the concept of aligned and misaligned filters need additional elaboration on the high level...I have the impression that such directional alignment might not be sufficient when we generalize to multi-class classification. Can the author elaborate more on this?
> >
> > At a high level, the notion of alignment captures whether a filter consistently responds to the underlying signal in the data. Specifically, once a filter is aligned, the sign of its inner product with the signal remains unchanged over time, i.e., if   $\langle w^{(t)}, \mu \rangle > 0$  then for all $t' \geq T$,   $\mathrm{sign}(\langle w^{(t')}, \mu \rangle) = \mathrm{sign}(\langle w, \mu \rangle)$ as established in Lemma 27. Importantly, this definition is not restricted to the binary setup. The key property - the preservation of sign consistency with respect to the signal - provides a principled way to quantify misalignment (see Equation 8 in our empirical measurements). While our current analysis formalizes this in the binary case, we believe the same notion can be naturally extended to multi-class classification. In particular, one could define alignment with respect to multiple class-specific signals, and then study whether filters consistently align with the relevant signal subspaces. We highlight this as an exciting direction for future work.
> >
> > > At multiple points, the authors mentioned that some assumptions (e.g., orthogonality) are meant to ease the technical exposition and relaxing them is not difficult. Can you please elaborate on high level how can these be easily relaxed?
> >
> > Thank you for pointing this out. We indeed mention two simplifying assumptions - orthogonality of vectors and equal client dataset sizes, that primarily serve to ease the exposition. Both can be relaxed without significantly affecting the analysis, as we elaborate below.
> >
> > 1. **Relaxing Orthogonality.**
> > This relaxation has already been addressed in prior work [a]. The key idea is that under Condition (2) on $d$, we can show that $\frac{\langle \xi, \mu \rangle}{\|\|\xi\|\|^2} = \mathcal{O}(1/d) \approx 0$ which implies that $\xi$ and $\mu$ are nearly orthogonal with high probability when $d$ is sufficiently large. Consequently, terms involving $\langle \xi, \mu \rangle$ can be treated as negligible noise, absorbed into existing noise terms, and do not significantly affect the analysis.
> >
> > 2. **Relaxing Equal-Sized Datasets.**
> > For clients with unequal dataset sizes, we can redefine the heterogeneity measure as: $h = \frac{1}{K} \sum_{k=1}^K \frac{\min\{|D_{+,k}|, |D_{-,k}|\}}{N_k}$, where $N_k$ is the dataset size at client $k$. Correspondingly, the local loss function at client $k$ would use $N_k$ (rather than the global $N$) in its denominator, as in Eq. (3). Additionally, we would impose a mild condition (Condition C3) ensuring that each client has a sufficient number of training samples. Beyond these modifications, we do not anticipate major changes to the analysis.
> >
> > In summary, both assumptions can be relaxed with relatively minor technical adjustments, and the main insights of our theoretical framework remain intact.

---

> > > ### Author Response · Authors · 2025-08-28
> > > **Response to Reviewer p1Zg [3/3]**
> > >
> > > ## Re: Requested Changes
> > >
> > > > 1. To connect better between experiment & theory, the authors can consider re-partition the 10-class CIFAR-10 dataset into a 2-class dataset, pre-train the CNN & use it as initialization for FedAvg. We expect to see that (1) the pre-trained CNN has good generalization and (2) using it as initialization, the FL performance improves significantly.
> > > > 2. Repeat the above experiment with the pre-trained CNN on a different dataset, e.g., TinyImageNet, and see if we also see a drastic improvement on the target dataset with FL.
> > >
> > > We have conducted these experiments suggested as summarized in our Response to Weakness 2. Specifically, we (a) re-partitioned CIFAR-10 into two classes (airplane and car), pre-trained a 2-layer CNN, and used it as initialization for FedAvg, and (b) repeated the setup with a CNN pre-trained on two semantically similar classes from TinyImageNet (albatross and racecar). The results are included in our Appendix under *Additional Experimental Details and Results*.
> > >
> > > >3. Elaborate on the alignment notion. It is currently mentioned in Lemma 3 and very briefly even though this is a central concept. Particularly, I would like to see discussion on how this notion can or can be generalized to accommodate multi-class classification.
> > >
> > > We have expanded the discussion in response to Minor Point 2, directly after Definition 1, to clarify what it means for a filter to be aligned and how this property extends beyond binary classification. In particular, we emphasize the implication that alignment stabilizes the sign of the filter response (see Lemma 27) and provides a foundation for generalization to multi-class settings.
> > >
> > > > 4. Elaborate on how the pre-trained model's generalizability comes into the picture.
> > >
> > > The role of pre-trained model generalizability is explicitly discussed in our responses to Weaknesses 1 and 3. In short, our analysis formalizes how the distance between the pre-training signal and the fine-tuning signal $\|\|\mu^{\text{(pre)}} - \mu\|\|$) governs whether filters remain aligned, thereby determining the effectiveness of pre-training.
> > >
> > > > 5. It'd also be good to discuss how the current model can be generalized to deep CNN settings (e.g., ResNet-18) with multiple CNN layers. There might be a difference between filter alignment at raw input level and alignment at higher feature level. In other words, it is unclear from the current theory whether filter alignment at input level alone is sufficient to characterize the entire benefit of pre-training on federated learning.
> > >
> > > Yes, for deeper CNNs, we anticipate that we will need a hierarchical feature model [b,c,d] to capture alignment in deeper layers. Intuitively, while deeper architectures provide the capacity to learn more complex hierarchical signals, aligning a filter with the signal will also progressively become harder across layers with the alignment of the $L$-th layer potentially depending on the alignment of the ($L$ - 1)-th layer. In non-IID FL, this suggests that feature learning deteriorates as we move from the base layer toward the final layer. This intuition is supported by empirical evidence (see experiment in Table 1 in [e]) and explains why pre-trained initialization offers greater benefits for deeper networks compared to shallower ones.
> > > We have added this discussion in a new concluding paragraph titled *Extension to Deeper CNNs and Transformer Architectures*.
> > >
> > > ___
> > >
> > > **References**
> > >
> > > [a] Yiwen Kou, et al. . Benign overfitting in two-layer relu convolutional neural networks, *ICML, 2023*
> > >
> > > [b] Binghui Li and Yuanzhi Li. Adversarial training can provably improve robustness: Theoretical analysis of feature learning process under structured data. *arXiv preprint arXiv:2410.08503, 2024.*
> > >
> > > [c] Zeyuan Allen-Zhu and Yuanzhi Li. Backward feature correction: How deep learning performs deep (hierarchical) learning. *COLT, 2023*
> > >
> > > [d] Peng Wang et al.. Understanding deep representation learning via layerwise feature compression and discrimination. *arXiv preprint arXiv:2311.02960,2023b*
> > >
> > > [e] Yaodong Yu, et al. TCT: Convexifying federated learning using bootstrapped neural tangent kernels. *NeurIPS, 2022*

---

> > > > ### Comment · Reviewer_p1Zg · 2025-09-17
> > > > **Follow-up**
> > > >
> > > > Dear Authors,
> > > >
> > > > Thank you for the detailed response. I apologize for the late follow-up. I was overwhelmed with multiple editorial work in the past weeks.
> > > >
> > > > --
> > > >
> > > > Overall, I am more convinced now that the developed results are sufficient to explain the advantage of FL with pre-trained CNNs over FL with random initialization of CNNs for binary classification tasks. With the added experiments, this point has been made more clear to me.
> > > >
> > > > I also want to elaborate that in the 3rd point, I meant to ask why do we believe || mu - mu_pre || can be sufficiently small? Naively thinking, mu_pre is the signal from very broad pre-training dataset while mu is the signal from the fine-tuning data which is assumed to follow/align with the structure of the 2-class data generation model.
> > > >
> > > > So, my point earlier was that unless the pre-trained model was built on data that align with such 2-class data generation model, it is not clear to me how || mu - mu_pre || will be small. The added experiment with CNN pre-trained on TinyImageNet and fine-tuned later with CIFAR-10 now provide additional empirical data to support this to a certain extent (though i would expect a lot more experiments across multiple 2-class settings around this point).
> > > >
> > > > --
> > > >
> > > > Given the above, my final view of this work is that it does establish several interesting ideas and the established model offers a plausible explanation for the FL advantage restricted to 2-class scenarios with low-complexity CNNs. I think this is a precise statement of the contribution here.
> > > >
> > > > While I am aware that the authors are eager to generalize this to generic multi-class cases with larger models such as ResNet, most of the existing discussions in the paper and in the rebuttal around this point are somewhat cursory rather than concrete. These could be good leads to expand this work but I suspect that the 2-class data generation model as well as the alignment concept would need significant overhaul. For multi-class cases, it will not simply be the sign but also the ranking structure that needs to be characterized.
> > > >
> > > > --
> > > >
> > > > Having said that, I think the established ideas so far are sufficiently interesting and can be accepted under the condition that the authors do not over-claim their theoretical contribution. My original concern was that the theoretical analysis was developed for 2-class scenarios with limited CNNs but was used as a characterization of the observed phenomenon with ResNet and multi-class scenarios. This is the reason why I asked for additional experiments on simpler scenarios that fit better with the theoretical settings. With the added experiment, it is now sufficiently clear what was explained by the theoretical model & though limited, that's still a fine contribution.
> > > >
> > > > I hope the authors will continue to expand this work in the rebuttal's envisioned directions to offer a more robust theoretical foundation for federated fine-tuning.

---

### Decision · Action_Editor_yD7t · 2025-09-17

**Recommendation:** Accept with minor revision

**Additional Comments:**

Required revisions for camera-ready:

- Scope and claims: Explicitly constrain theoretical claims to the analyzed setting (binary classification, two-layer CNN, FedAvg, stated assumptions). Rephrase broader statements to avoid implying guarantees for multi-class or deep architectures. Where multi-class/deep results are discussed, frame them as empirical observations or future directions.
- Clarify alignment: Add a higher-level explanation of “aligned” vs. “misaligned” filters early in the paper and discuss limits and potential generalizations (e.g., multi-class, deeper CNNs, non-convolutional architectures). Briefly outline how orthogonality and other simplifying assumptions might be relaxed at a high level.
- Experiment–theory connection: Add a brief discussion connecting the ResNet experiments to the theory, making clear that these results are qualitative/indicative. If feasible within minor-revision scope, include a small binary-class subset experiment to directly mirror the theoretical setting; otherwise, state this as a limitation and future work.

Additional comments:
- Any code release would be much appreciated by our community.

**Audience:**

Yes

**Audience Explanation:**

The paper offers new insight into why pre-trained initialization can help federated learning via the lens of filter alignment and heterogeneity effects under FedAvg. Even though the theoretical model is simplified, these findings are of interest to parts of the TMLR audience focused on theory, federated optimization, and understanding generalization in FL. The results provide learnable takeaways about when and why pre-training can mitigate heterogeneity; meeting TMLR’s criterion of interest without relying on claims of novelty or SOTA.

**Claims And Evidence:**

Yes

**Claims Explanation:**

The theoretical claims (training loss convergence and test-error bounds for FedAvg on a two-layer CNN in a binary setting, together with the role of aligned vs. misaligned filters and data heterogeneity) are supported by rigorous analysis and proof sketches, with fuller proofs in the appendix. Reviewers found the analysis technically sound and the rebuttal clarified earlier concerns. Empirical evidence on CIFAR-10 and TinyImageNet, including a misalignment measure applied to ResNet-18, is consistent with the theory’s qualitative predictions and was strengthened with small post-rebuttal edits (added proof sketches, an extra experiment). The original manuscript over-generalizes beyond the proved scope (e.g., extrapolating from 2-class, shallow CNN theory to multi-class and deeper networks); these seem to have been handled by toning down the claims + describing what is currently achieved and what is left for future work. Acceptance should be conditioned on such tightening claims to match the proven setting and clearly labeling empirical observations outside that scope as suggestive rather than theoretically supported.

---

> ### Author Response · Authors · 2025-10-17
> **Response to Action Editor**
>
> Thank you for the decision and for the constructive comments that helped us improve our paper. We have incorporated all requested revisions in the camera-ready version. Specifically:
> * We ensure that all theoretical claims are constrained to the analyzed setting (binary classification with a two-layer CNN under the stated assumptions).
> * Broader statements about generalization are now clearly framed as empirical observations or directions for future work.
> * We have added a paragraph after Definition 1 providing a high-level explanation of alignment vs. misalignment and briefly discussed potential extensions to multi-class settings.
> * In the Conclusion section, we have expanded the discussion on how our insights could generalize to deeper CNNs and Transformer-based architectures.
> * Following the orthogonality assumption, we added a short paragraph outlining how this assumption might be relaxed.
> * In the Experiments section, we now explicitly note that the ResNet results are qualitative and indicative. Additionally, we include a small binary-class subset experiment in the Appendix that directly mirrors our theoretical setting and reference it in the main text.
>
> Finally, we have added supplementary material containing the code for all experiments in Section 4.
>
> Thank you once again for your thoughtful feedback and to the reviewers for their valuable time and effort.